# LLM-PySC2: Starcraft II learning environment for Large Language Models

**Zongyuan Li[1], Yanan Ni[2], Runnan Qi[2], Chang Lu[1], Lumin Jiang[2], Xiaojie Xu[1],**
**Xiangbei Liu[1], Pengfei Li[1], Yunzheng Guo[1], Zhe Ma[1], Huanyu Li[1], Wu Hui[1],**
**Xian Guo[1,*], Kuihua Huang[2,*], Xuebo Zhang[1*]**
[1]College of Artificial Intelligence, Nankai University, Tianjin, China
[2]Laboratory for Big Data and Decision, National University of Defense, Changsha, China

## Abstract

The tremendous potential has been demonstrated by large language models (LLMs) in intelligent decision-making problems, with unprecedented capabilities shown across diverse applications ranging from gaming AI systems to complex strategic planning frameworks. However, the StarCraft II platform, which has been widely adopted for validating decision-making algorithms in the past decade, has not yet provided substantial support for this emerging domain. To address issues that LLMs cannot interface with the hundreds of actions of the pysc2 backend and the lack of native support for multi-agent (MA) collaboration, we propose the LLM-PySC2 environment. This is the first environment that offers LLMs the complete pysc2 action space with sufficient multi-modal information and game Wiki knowledge. With an asynchronous query architecture, the environment efficiently interacts with LLMs that maintain a constant latency regardless of the scale of the agents' population. In the experiments, we evaluated LLMs' decision-making performance in both the macro-decision and micro-operation scenarios, with traditional StarCraft II Multi-Agent Challenge (SMAC) tasks and a series of new proposed. Results indicate that LLMs possess the potential to achieve victories in complex scenarios but cannot constantly generate correct decisions, especially in the recovered pysc2 action space and MA settings. Without task-relevant instructions, the pre-trained models suffer from issues such as hallucinations and inefficient collaboration. Our findings suggest that StarCraft II still challenges in the era of large models, revealing that there is a lot to do to develop an advanced LLM decision-making system, and the proposed LLM-PySC2 environment will support future development of LLM-based decision-making solutions.

## 1 Introduction

The remarkable progress of LLMs has not only enhanced their reasoning capabilities but also positioned them as multitask strategists, even without post-training on specialized domains. Unlike reinforcement learning (RL) based agents, LLMs exhibit advantages in better context understanding, knowledge utilization, and human-AI interactions, acting in a wider range of zero-shot scenarios like gaming (1)-(12) , robot manipulation/navigation (13)-(17) , financial and trading (18)-(20).

However, there is still a lot to do to release the potential of LLM decision systems. Current works are mostly limited to prompt engineering (6)(7) , reflection (4)(8)(9)(22) , LLM workflow (3)(18)(21) to dismantle tasks into smaller tasks. These works enable LLMs to act better in diverse scenarios, but the knowledge-learning problem for a specific domain remains unsolved.

---

*Corresponding authors

39th Conference on Neural Information Processing Systems (NeurIPS 2025).

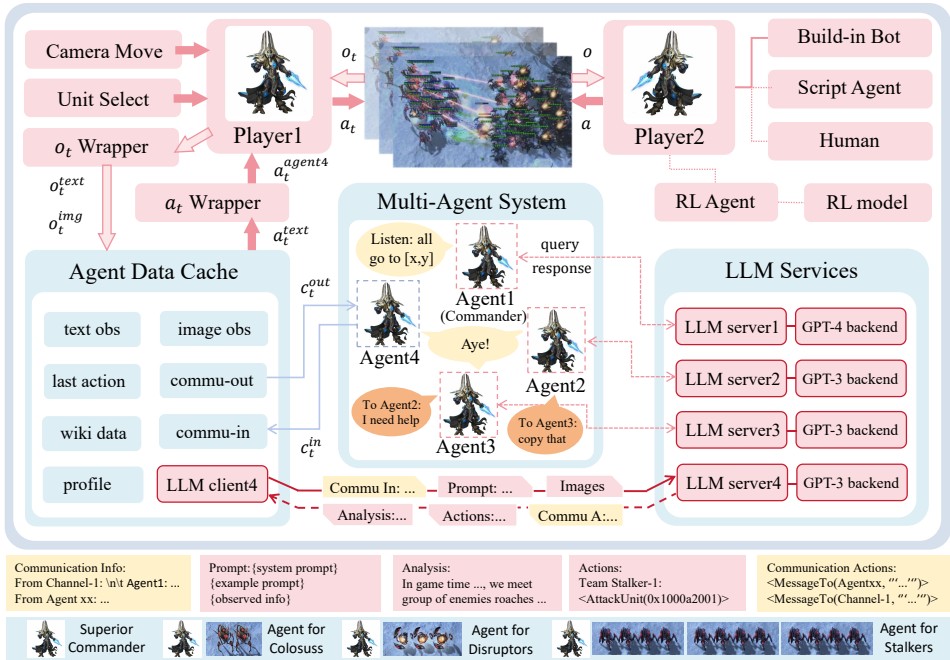

Figure 1: **LLM-PySC2 framwork.** In LLM-PySC2, the original observation will be wrapped into a text- or multi-modal observation. LLM-generated text action can be recognized and transformed into PySC2 functions, enabling LLMs to interact with the StarCraft II environment and control the units.

Currently, most LLM decision-making solutions are developed in relatively simple environments, resulting in ignorance of LLM's shortcomings. For example, MineDojo(23) is relatively comprehensible for LLMs and exhibits a high tolerance for errors, while some other works simplify the policy space of the environment(5)(6). Earlier works, such as StanfordTown(24), do not even concern decision-making ability but is more focused on LLMs' behaviors.

The StarCraft II environment, well known for its complexity, has been widely used as a validation platform for decision algorithms in the past decade, supported multi-agent research such as VDN(25), Qmix(26), MAPPO(27), the milestone algorithm Alpha-Star(28) and DI-Star(29). It is precisely the extremely high complexity that makes it the most authoritative verification platform for decision-making algorithms. However, since the vector interfaces are not compatible with LLMs, the StarCraft II environment does not support complete interactions with LLMs in the past few years.

Existing LLM Starcraft II environments, such as Swarm Brain(5), TextStarCraft II(TSC2) (6), have the problem of severely limiting the action space. They cut off most unit control operations and reduce continuous action space to discrete. Although the over-simplified environments have attracted attention for LLM decision research in the past years, they hindered further research due to the lack of complexity. Other works like (30)(31) do not support complete games.

At the same time, the support of current platforms for multi-agent systems is insufficient. Currently, most LLM multi-agent systems(12)(18) expose only a single agent to interact with the environment, while others act as modules for data processing or aggregating. Other works focus on conducting social simulations, emphasizing the accuracy of simulation (32)-(34) rather than promoting multi-agent collaboration.

To provide support for LLM decision-making, we developed *LLM-PySC2*, an environment derived from the StarCraft II Learning Environment (SC2LE)(35). This environment expanded the action space to complete pysc2 action space, allowing agents to perform fine-grained operations and unit skills. We also provide agents with comprehensive observations, including images and Wiki Knowledge(36).

It is worth noting that this is a platform with native multi-agent framework. We enable all kinds of multi-agent cooperation such as centralized decision-making and distributed decision-making. To

avoid an increase in waiting time as the number of agents grows, we build an asynchronous query architecture to maintain the latency of multi-agent queries.

In experiments, eight new scenarios were proposed. Unlike the SMAC(37) tasks, these tasks require more on task understanding and usage of unit skills. Mainstream LLMs are evaluated in both the complete StarCraft II games and mini scenarios. Results indicate that pre-trained LLMs *have* possess zero-shot decision-making ability but *lack the ability* to make consistently effective decisions. Without task-specific training, pre-trained LLMs cannot always find the key elements for victories. They fail to identify the important aspect of the situation, making mistakes in analysis and even dealing damage to allies sometimes.

Our contributions can be concluded as follows:

(1) We propose the first LLM StarCraft II framework with a complete pysc2 action space and provide a structured Wiki knowledge database of all units' information.

(2) We provide native support for multi-agent collaboration in our platform, paired with an asynchronous architecture that ensures a stable latency regardless of the population of LLM agents.

(3) We propose several new evaluation scenarios for LLM decision-making and evaluate LLMs' performance in both the macro-decision scenarios and the scenarios for micro-operations.

Problems of the LLM decision system are also discussed in the final sections of our paper. Results indicate that current LLMs cannot effectively handle complex StarCraft II scenarios due to serious hallucinations and lack of domain knowledge. How to increase the ability of LLMs in complex decision-making problems, at an acceptable cost, still poses a challenge in the era of large models and remains an unsolved problem.

## 2 LLM-PySC2 environment

### 2.1 Framework

The LLM-PySC2 environment is built on the player level of SC2LE. As shown in Figure 1, two players fight against each other and play the role of interacting with the pysc2 backend. They directly control the camera, select units, collect observations, and execute actions.

To precisely control the whole system, a multi-agent framework is designed. Agents of the system collaborate through natural language communication. At each step time $t$, agent $i$ with profile $p^i$ get the observations $o_t^i$ from the environment, queries remote LLM for analysis $ana_t^i$ and strategy $stg_t^i$, communication messages $m_t^i$ and actions $a_t^i$:

$$(ana_t^i, stg_t^i, m_t^i, a_t^i) = LLM(p^i, o_t^i)$$

Then the player sends the joint action to the environment and transmits messages to assigned agents:

$$(o_{t+1}^1, o_{t+1}^2, \ ... \ , o_{t+1}^n) = Env(a_t^1, a_t^2, \ ... \ , a_t^n; m_t^1, m_t^2, \ ... \ , m_t^n)$$

Pseudo code of the interaction and query process can be seen in Appendix A.

### 2.2 Actions

Actions are the most important part of a decision-making problem. In LLM-PySC2, textual actions play the role of the interface for large models and the environment. These actions are defined as:

$$< ActionName(args) >$$

where args refer to screen or minimap coordinates, unit tag, or their combination. Compared to discrete text actions, these actions avoid clipping the policy space and neglecting StarCraft II complexity. Details of the actions are provided in Appendix B.

In SC2LE, there are about 500 original for controlling Protoss, Terran, and Zerg. Most of them require additional parameters such as a screen or minimap position. These actions further constitute a huge policy space, making it one of the most complex environments for decision-making problems.

As shown in Fig 2, there are more than 100 text actions for Protoss agents in the LLM-PySC2 environment, which can be classified as unit control, unit skills, building, researching, training, etc. Different from other environments, these actions increase the theoretical performance of optimal policy but also raise the challenge of generating correct actions.

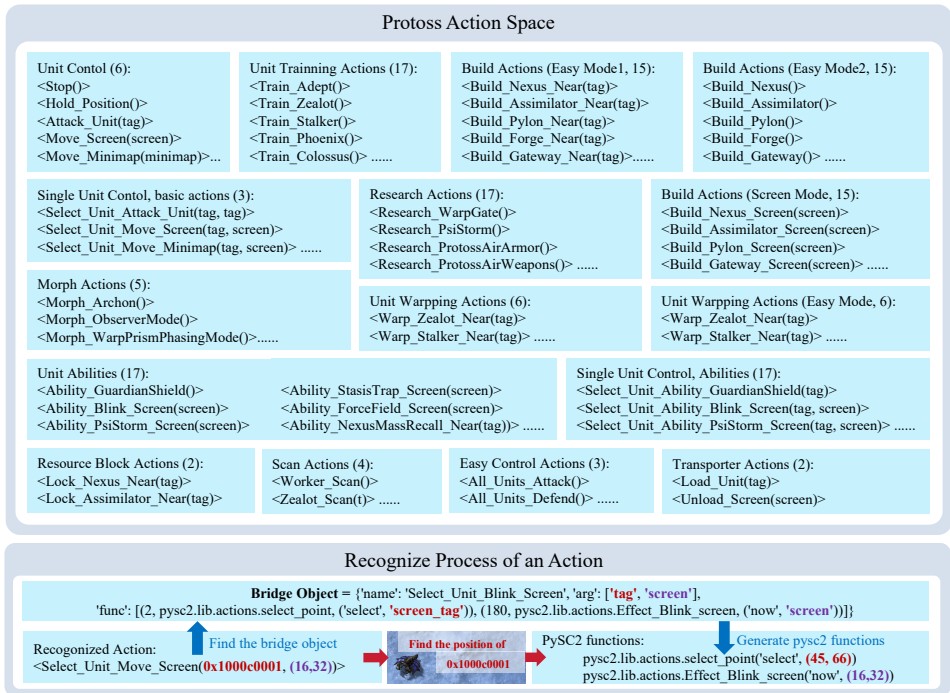

Figure 2: **Protoss action space and the recognition process.** LLM-PySC2 is the first LLM decision-making environment with complete pysc2 action space. LLM controls units by output actions in the shape of <Action_Name(args)>. The environment transforms text action into pysc2 functions according to a transform protocol and the relevant bridge object of the action.

## 2.3 Observation

Observation provides fundamental support for decision-making. Given the distinct requirements of different agents, we developed an interface that offers each agent the observations specifically suited to their tasks. Additionally, with multi-modal observations that convey rich semantic and visual information, we released the potential for a deeper understanding of the situation, solving the problem that the previous environment had only observation of unit quantity information.

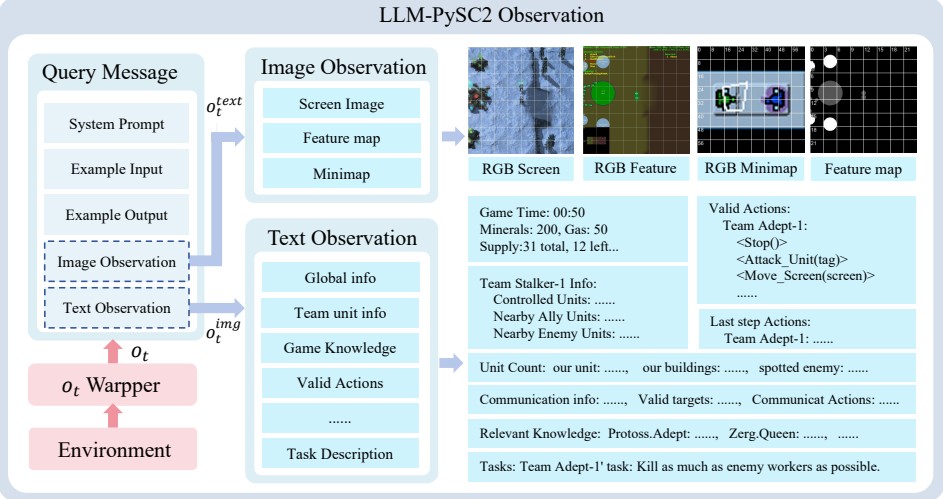

Figure 3: **LLM-PySC2 observations.** LLM-PySC2 provides multi-modal observation. The observation wrapper generates text and image observations that contain all the important information for decision-making, with access to images of the screen, minimap, and pysc2 original feature maps.

### 2.3.1 Text Observation.

As illustrated in Fig. 3, an observation wrapper is implemented to process the relevant text observations for each agent. This wrapper includes a set of functions tailored for handling different types of text information. Once all parts of the text observation are generated, they will be aggregated into OpenAI query messages, which include system prompts, example inputs and outputs, as well as a series of images. These messages are then sent to the LLM server for querying responses.

Considering possible user requirements, we expose all the observation interfaces in the open-source code repository. It is possible to customize the wrapper to generate other kinds of text observations. More detailed examples of text observation can be seen in Appendix C.

### 2.3.2 Image Observation.

In StarCraft II scenarios, image observation provide information such as terrain and relative position. It is almost inevitable to use images to describe such higher-dimensional information.

In LLM-PySC2, we provide four kinds of image observation: RGB-Screen, RGB-Minimap, RGB-Feature, and Original-Feature-Maps. Image observation wrapping functions collect the image from the pysc2 backend, adding auxiliary lines and annotations to facilitate the coordinate recognition by LLMs. The image will be encoded into a base 64 string and will be added to the message to query the LLMs for analysis, actions, and communication behaviors.

## 2.4 Multi-Agent System

Disassembling complex problems into small tasks in a multi-agent system has become a basic solution. Different large models interact through natural language, coordinating their behaviors and managing the massive StarCraft II system together.

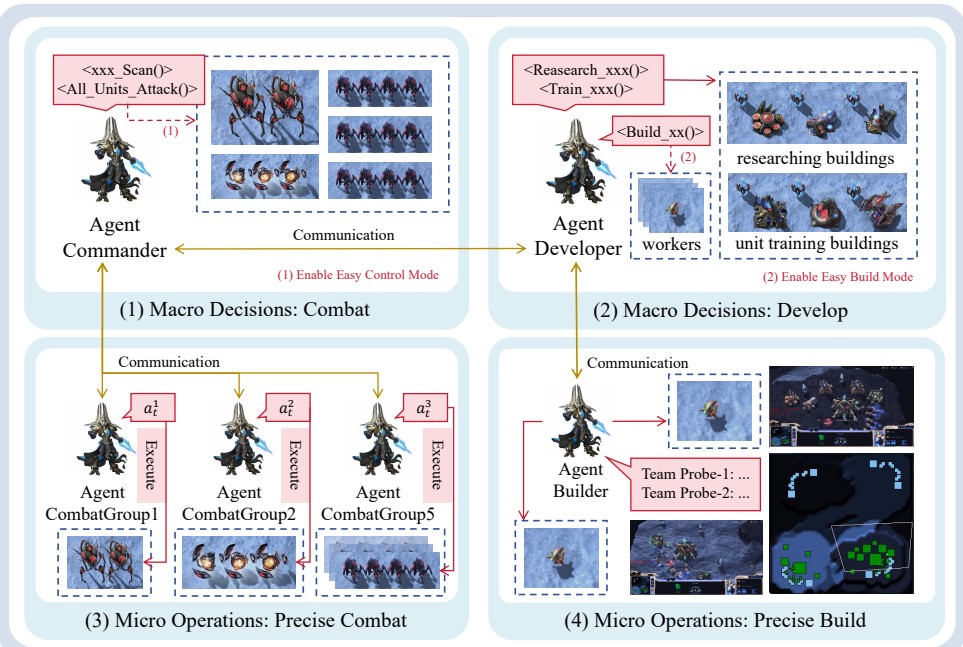

Figure 4: **LLM-PySC2 multi-agent system.** In LLM-PySC2, game control is divided into combat((1), (3)) and development((2), (4)). In standard unit control mode, the agent Commander sends messages to agents named CombatGroupi, and the CombatGroup agents control their units moving, attacking, or using skills to achieve tasks assigned by superiors. In standard build mode, the agent Developer trains units, updates technologies, and asks the agent Builder to build buildings. Then the Builder controls workers and chooses positions to construct new buildings.

In LLM-PySC2, agents collaborate by communicating with each other. They can discuss in a channel or directly send messages to another agent. As shown in Fig. 1, at each step, received messages will be added to observation, and the agent can respond to others by generating Communication actions shaped as $< MessageTo(TargetName, '''content''') >$. When the agent receives messages, the received messages will be displayed in their origin form. An Agent should analyze both the observed situation and the requests/information from others, and finally generate actions and reply to their teammates.

As shown in Fig. 4, we define four kinds of agents responsible for (1) macro-decisions for combat deployment, (2) macro-decisions for economic development, (3) micro-operations for combat, and (4) micro-operations for building. Agents for macro-decisions organize other agents to work together, while agents for micro-operations execute specific actions. Note that, agents of LLM-PySC2 query in independent threads, ensuring a constant waiting time when the number of agents increases.

The multi-agent system supports both centralized and decentralized decisions in the environment. Two 'Easy Modes' are also provided for simplifying some aspects that researchers are not very concerned about, among which 'Easy Build' disables the agent Builder and helps researchers concentrate more on multi-agent collaboration in the combat, while 'Easy Control' disables the agents CombatGroups and helps researchers concentrate more on planning and multi-modal information processing.

## 3   Experiments

In this section, we introduce two series of experiments: (1) Experiments for macro-decisions, i.e. **complete StarCraft II game**; (2) Experiments for micro-operations, including classic SMAC scenarios and eight new tasks that require units to use their skills and achieve assigned goal. To distinguish micro-operation scenarios from the traditional SMAC environment, we refer to these two groups of experiments the **LLM-SMAC task group** and the **LLM-PySC2 task group**.

Combined with the complete StarCraft II games, these experiment scenarios constitute one of the most comprehensive experiment groups in LLM decision-making and support research on enhancing LLMs' abilities in reasoning, planning, learning, and multi-agent cooperation.

We use the Kill/Death (KD) ratio and Winning Rate (WR) to evaluate the performance of the LLMs:

$$V_{unit} = minerals(unit) + 2 \times gas(unit)$$
$$KD = V_{killed\_units}/V_{dead\_units}$$
$$WR = num(win)/num(total)$$

where $V_{killed\_units}$ and $V_{dead\_units}$ refers to killed units' value and dead units' value of the LLM's camp. The higher the value of KD and WR, the better the performance of LLMs.

### 3.1   Experiments for Macro-Decisions

#### 3.1.1   Experiment Settings

Complete games demand real decision-making abilities, such as analyzing situations, planning for tactic strategy, deceiving the opponent, and engaging with the enemy at the right time. To evaluate the performance of LLM macro-decisions, we tested the three modes in the Simple64 map: (1) easy control + easy build (ECEB); (2) standard control + easy build (SCEB); and (3) easy control + standard build (ECSB). Considering the poor performance of LLM in complex settings, we do not further test the last mode (4) standard control + standard build (SCSB).

For games with easy control settings, we enable the agent Commander to directly control all units to attack, defend, retreat, and scan for information. For the standard control settings, we enable agents named CombatGroup-i to precisely control different kinds of unit to move, attack, and use skills.

For games with easy build settings, the agent Developer can build buildings by generating actions $< Build\_BuildingName() >$. For games that enable standard build, the agent Developer can only train units and upgrade technologies, and has to communicate with the agent Builder to build in specific coordinates $[x, y]$ by generating actions $< Build\_BuildingName\_Screen([x, y]) >$.

In these experiments, we give each agent a client for querying GPT-4o-mini. For macro-decision agents, we provide relevant text information and minimap images. For standard unit control agents, we provide text observation and both the screen image and minimap images. Examples of observations, responses for different agents, and detailed experimental settings can be seen in Appendices C and D.

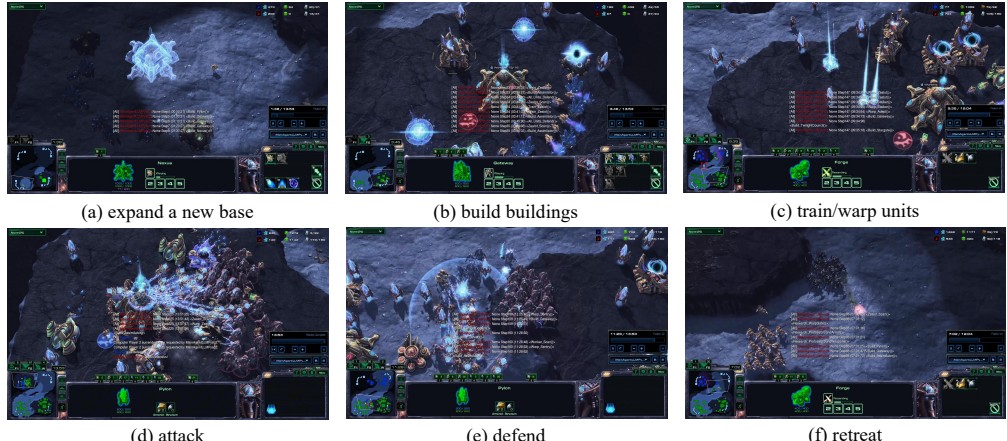

| (a) expand a new base | (b) build buildings | (c) train/warp units |
| (d) attack | (e) defend | (f) retreat |

Figure 5: **StarCraft II complete game in LLM-PySC2.** StarCraft II complete game requires both the macro-decision ability and micro-operation ability. The agent Developer and Builder has to (a) expand new bases, (b) build new buildings, (c)train or warp units for combat, and upgrade technologies. The agent Commander with agents for CombatGroups controls the army (d) defend, (e) attack, (f) retreat, or make complex deployment to deceive and defeat the opponent.

### 3.1.2 Experiment Results

In the macro-operation tasks (complete StarCraft II games), we conducted 30 repeated experiments from level-1 (very easy) to level-7 (very hard/elite). As shown in Table 1, two agents in the ECEB mode control the whole system via discrete actions and perform nearly the same as in TSC2 (6). At level-5, LLMs can only win about 30% of the games and nearly lose all games at level-6 and above.

Table 1: Winning Rates of GPT-4o-mini in Complete StarCraft II games (with 90% Wilson Score Confidence Intervals. Red for upper limits and green for lower.)

| Mode | Winning Rate from Level-1 to Level-7 | | | | | | |
| | L1 | L2 | L3 | L4 | L5 | L6 | L7 |
| --- | --- | --- | --- | --- | --- | --- | --- |
| ECEB (EasyControl+EasyBuild) | $100\%_{92\%}^{100\%}$ | $100\%_{92\%}^{100\%}$ | $80\%_{66\%}^{89\%}$ | $57\%_{42\%}^{70\%}$ | $30\%_{18\%}^{45\%}$ | $3\%_{1\%}^{13\%}$ | $0\%_{0\%}^{8\%}$ |
| SCEB (StandardControl+EasyBuild) | $100\%_{92\%}^{100\%}$ | $60\%_{45\%}^{73\%}$ | $0\%_{0\%}^{8\%}$ | $0\%_{0\%}^{8\%}$ | $0\%_{0\%}^{8\%}$ | $0\%_{0\%}^{8\%}$ | $0\%_{0\%}^{8\%}$ |
| ECSB (EasyControl+StandardBuild) | $100\%_{92\%}^{100\%}$ | $80\%_{66\%}^{89\%}$ | $60\%_{45\%}^{73\%}$ | $17\%_{8\%}^{30\%}$ | $17\%_{8\%}^{30\%}$ | $0\%_{0\%}^{8\%}$ | $0\%_{0\%}^{8\%}$ |

In SCEB and ECSB modes, LLMs perform worse due to the recovered complexity from the complete action space and the higher demand for collaboration. In SCEB mode, the Easy Build part develops the economy and military strength the same as in ECEB mode, but agents for micro-operations frequently make mistakes in command, resulting in a 0% winning rate from level-3 onwards. Notably, in ECSB mode, while the Builder agent controls workers for standard construction operations, it maintains certain competitiveness from level-1 to level-3, achieving a 60% winning rate at level-3. However, as the difficulty increases further, issues such as inappropriate building placement and suboptimal defense layouts become evident, causing the winning rate to drop sharply to 17% at level-4 and resulting in complete inability to win from level-6 onwards.

To better assess the statistical significance of the winning rates and account for the uncertainty associated with the limited sample size, we calculated Wilson Score Confidence Intervals. This interval provides a reliable confidence interval estimate for a binomial proportion, particularly suitable for small sample sizes or proportions near 0 or 1. The formula for the Wilson interval is as follows:

$$a = \frac{\hat{p} + \frac{z^2}{2n}}{1 + \frac{z^2}{n}}, \ b = \frac{z}{1 + \frac{z^2}{n}} \sqrt{\frac{\hat{p}(1 - \hat{p})}{n} + \frac{z^2}{4n^2}}$$

$$WP_{upper} = a + b, \ WP_{lower} = a - b,$$

where $\hat{p}$ is the observed winning rate, $n$ is the number of trials (30 in this case), and $z$ is the $1 - \alpha/2$ quantile of the standard normal distribution, i.e., $z = \Phi^{-1}(1 - \alpha/2)$. For a 90% confidence interval ($\alpha = 0.10$), $z \approx 1.645$. The upper and lower limits of the interval, denoted as $WP_{\text{upper}}$ and $WP_{\text{lower}}$, are reported in the table in red and green, respectively.

## 3.2 Experiments for Micro-Operations

### 3.2.1 Experiment Settings

SMAC is a well-known benchmark for multi-agent reinforcement learning (MARL) approaches. We provide compatible support for SMAC tasks. Note that, unlike the SMAC tasks that $n$ units are controlled by $n$ agents, the LLM-SMAC units are controlled in groups. It is not recommended to compare the LLM-based method with the MARL-based method due to different control frequencies.

In the LLM-PySC2 task group, eight new experimental scenarios were constructed. These tasks introduce unit skills into the experiments. Unlike SMAC, which focuses only on incoming combat, LLM-PySC2 requires an understanding of task description, planning attack routes, and utilizing skills to achieve the goal. Tasks 1 to 4 are designed as single-agent tasks, while tasks 5 to 8 are designed as multi-agent tasks. Agent settings are the same as for the standard control mode of the complete StarCraft II game. More detailed settings can be seen in Appendix D.

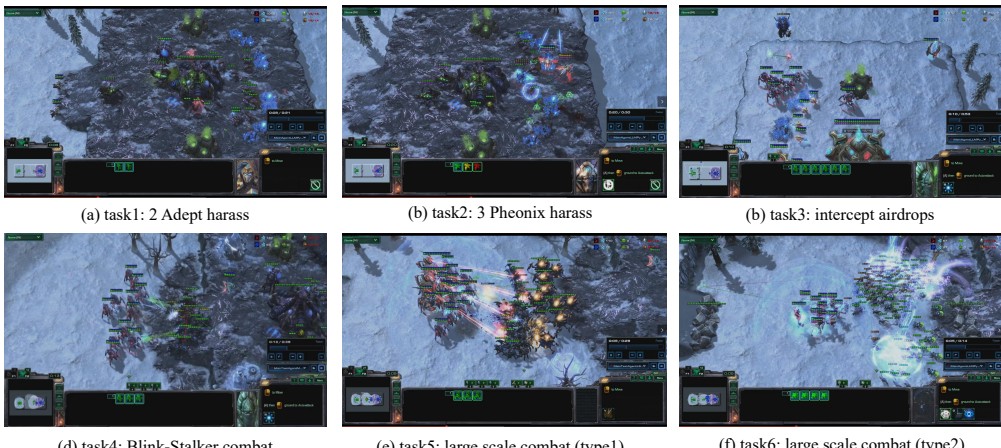

(a) task1: 2 Adept harass     (b) task2: 3 Pheonix harass     (b) task3: intercept airdrops

(d) task4: Blink-Stalker combat     (e) task5: large scale combat (type1)     (f) task6: large scale combat (type2)

Figure 6: **Experiments for micro-operations: LLM-PySC task group.** games. (a)(b) Controlling 2Adepts/3Pheonix to harass enemy economy, kill more than half of enemy workers; (c)(d) Controlling Stalkers to intercept incoming airdrop or defeat enemy Roaches using Blink ability; (e)(f) Controlling a combat group of several unit types, use skills especially Area-of-Damage skills to defeat enemies.

### 3.2.2 Experiment Results

In the micro-operation tasks, we conducted 20 repeated experiments for each LLM (except GPT-3.5-turbo which evaluates 50 games). As shown in Table. 2, all the tested LLMs act poorly in LLM-SMAC scenarios, similar to works such as (30). LLMs make obvious mistakes that do not move their long-range combat units, even when attacked by melee units in 3s_vs_3z. In 2s3z, the agent for Stalkers sometimes escapes from the battlefield, resulting in the quicker death of ally Zealots.

In the LLM-PySC2 task group, we evaluate the performance of 9 models. Results in Table. 3 demonstrate that LLM suffers from hallucinations and the lack of knowledge (More details are provided in the Discuss section and Appendix E).

Table 2: Kill/Death Rates and Winning Rates of LLMs in LLM-SMAC Tasks.

| Model | Task Names / KD (WR) | | | | | |
|---|---|---|---|---|---|---|
| | 2s3z | 3s5z | 1c3s5z | 3s5z_vs_3s6z | 2c_vs_64zg | 3s_vs_3z |
| gpt-3.5-turbo | 0.60(22%) | 0.43(**4**%) | 0.91(44%) | 0.29(0%) | 0.52(0%) | 0.05(0%) |
| gpt-4o-mini | 0.66(20%) | 0.39(0%) | **1.01**(**50**%) | 0.29(0%) | 0.54(0%) | 0.09(0%) |
| glm-4-plus | **0.81**(**25**%) | 0.46(0%) | 0.47(0%) | 0.33(0%) | 0.54(**5**%) | **0.15**(0%) |
| claude3-haiku | 0.58(5%) | **0.48**(0%) | 0.48(0%) | 0.32(0%) | 0.52(0%) | 0.10(0%) |
| llama3.1-8b | 0.19(0%) | 0.23(0%) | 0.18(0%) | 0.14(0%) | 0.49(0%) | 0.00(0%) |
| gpt-4o | 0.76(20%) | 0.47(0%) | 0.80(30%) | **0.35**(0%) | **0.56**(0%) | **0.15**(0%) |

Table 3: Kill/Death Rates and Winning Rates of LLMs in LLM-PySC2 Tasks (level-1).

| Model | Task Names / KD (WR) | | | | | |
|---|---|---|---|---|---|---|
| | task1 | task2 | task3 | task4 | task5 | task6 |
| gpt-3.5-turbo | 1.23(58%) | 0.13(4%) | 6.63(38%) | 0.38(0%) | 0.61(8%) | 0.28(0%) |
| gpt-4o-mini | 1.67(70%) | 0.16(0%) | 3.46(0%) | 0.39(0%) | 0.62(20%) | 0.30(0%) |
| glm-4-plus | 0.78(30%) | **0.21**(5%) | 153(**100**%) | 0.38(0%) | 0.60(10%) | 0.30(0%) |
| claude3-haiku | 2.19(**90**%) | 0.19(**10**%) | 5.25(40%) | 0.34(0%) | **0.75**(**25**%) | **0.33**(0%) |
| llama3.1-8b | 0.28(5%) | 0.12(5%) | 14.9(75%) | 0.18(0%) | 0.48(5%) | 0.14(0%) |
| llama3.1-70b | 0.36(15%) | 0.14(0%) | 58.9(95%) | 0.33(0%) | 0.59(15%) | 0.31(0%) |
| llama3.1-405b | 0.70(30%) | 0.10(0%) | 3.0k(**100**%) | 0.28(0%) | 0.56(10%) | 0.32(0%) |
| gpt-4o | **2.27**(80%) | 0.16(**10**%) | **Inf**(**100**%) | **0.46**(0%) | – | – |
| gpt-o1-mini | 1.36(60%) | 0.04(0%) | – | – | – | – |

Two findings derive from these results: (1) Reasoning models such as GPT-o1-mini cannot significantly improve the decision-making ability in an environment never seen before; (2) Scaling law does not work well in decision-making problems that Llama3.1-405b does not significantly outperform Llama3.1-70b (but enough parameters is crucial for basic decision-making ability). These problems are possibly due to a lack of relevant knowledge and instructions in the pre-training stage.

Table 4: Kill/Death Rates and Winning Rates of Gpt-3.5-turbo in LLM-PySC2 Tasks (level-1/2/3).

| Task Level | Task Names / KD (WR) | | | | | |
|---|---|---|---|---|---|---|
| | task1 | task2 | task3 | task4 | task5 | task6 |
| Task Level-1 | 1.23(58%) | 0.13(4%) | 6.63(38%) | 0.38(0%) | 0.61(8%) | 0.28(0%) |
| Task Level-2 | 0.56(5%) | 0.04(0%) | 3.31(5%) | 0.34(0%) | 0.52(0%) | 0.20(0%) |
| Task Level-3 | 0.39(0%) | 0.05(0%) | 1.99(0%) | 0.31(0%) | 0.40(0%) | 0.26(0%) |

To avoid the situation that all tasks achieve 100% winning rates several years after its proposal, we set three difficulty levels for the LLM-PySC2 task group. As the level grows, it will be more difficult for the LLMs to reach the goal due to additional enemy units or upgrades. We evaluate the performance of GPT-3.5-turbo in these tasks, as shown in Table. 4, and serve it as a baseline for future research.

### 3.3 Latency and Token Cost.

Since StarCraft II is a real-time environment, we also report the runtime latency and token consumption to assess the practicality of LLM-based decision-making. Table 5 summarizes the average observation size, response size, and runtime latency by agent type in complete-game settings using GPT-3.5. The environment-side latency (i.e., observation processing and action parsing) remains

below 0.1 seconds per step, which is negligible compared to the model-side inference cost. As shown in the table, LLM response latency (5.2–5.6 seconds per query) is the dominant overhead, demonstrating that the PySC2 execution pipeline is efficient and that LLM inference is the true computational bottleneck.

Table 5: Latency and token cost by agent type in complete-game settings (GPT-3.5).

| Agent Type | Obs Tok. | Resp Tok. | Env Lat. (Obs) | Env Lat. (Act) | LLM Lat. |
|---|---|---|---|---|---|
| Commander | 1240 | 321 | 0.016 ms | 0.001 ms | 5.23 s |
| Developer | 1587 | 320 | 0.076 ms | 0.001 ms | 5.51 s |
| Other | 2269 | 363 | 0.019 ms | 0.001 ms | 5.58 s |

## 4    Discussion

In this section, we discuss three challenges in LLM decision making. These challenges significantly reduce performance, severely hindering the application of the LLM-based decision-making system.

**Lack of domain knowledge**. Correct and sufficient knowledge is the prerequisite for correct decisions. However, there is no guarantee that all knowledge across all fields are introduced in the pre-training phase As a result, LLMs may not realize that 49 additional supplies are far beyond the demand for StarCraft II games, or know that shields recharge in the 2s_vs_1sc scenario.

**Hallucinations and mistakes**. Hallucinations and mistakes have an inevitable impact on the decision-making process. LLM suffers from (1) input-conflicting hallucinations that generate invalid actions; (2) fact-conflicting hallucinations that mistake ally units for enemy units; and (3) context-conflicting hallucinations that mistake screen coordinates for minimap coordinates.

**Inefficient collaboration**. Effective information exchange is critical for multi-agent collaboration. However, LLMs generate communication messages with a lot of non-essential and incorrect information. At the same time, they tend to unconditionally trust their teammates and ignore possible errors in the incoming information, which severely damages the performance in StarCraft II games.

These problems hinder the further application of LLM-based intelligent decision-making systems, waiting for further research and solutions. Examples of these problems are provided in Appendix E.

## 5    Conclusion

In this paper, we introduce a new environment for LLM decision-making, the first environment that accommodates the complete continuous PySC2 actions, and the first LLM StarCraft II environment with a multi-agent framework and communication system. In experiments, we evaluated the performance of mainstream LLMs in complete StarCraft II games and both the LLM-SMAC and LLM-PySC2 task groups, among which the LLM-PySC2 task group is a brand-new experimental scenario that we designed for large models. Results show that LLMs can make decisions and generate valid actions but cannot make effective decisions consistently. Still, the quality of the decision is relatively low and there are several problems such as hallucinations, poor utilization of knowledge of the game, and lack of understanding of the world. Results indicate that learning in the deployment environment is necessary for LLM-based decision-making solutions. We hope the LLM-PySC2 environment can promote research on LLM learning methods, helping LLM-based decision-making methods better adapt to task scenarios.

## 6    Acknowledgement

We thank the anonymous reviewers for their insightful comments. This work was supported in part by the National Natural Science Foundation of China under Grant 62293510/62293513. The views expressed herein are those of the authors and do not reflect the views of the funding agencies.

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

# Appendix A. Pseudo Code

## A.1 LLM-PySC2 Rollout Process

---

**Algorithm 1** LLM-PySC2 Rollout Process

---

**Require:** Map name. Max waiting time $T$ for a step. Profiles for each agent, with model-name, api-key, api-url for remote LLMs.

Initialize environment Env
Initialize player with its agents according to the profile
Initialize LLM client for agents
**while** not Env.is_terminated() **do**
    Add the tags of new units to relevant agent's data cache
    Remove the tags of dead units from relevant agent's data cache
    **while** current step waiting time < max waiting time **do**
        time.sleep(0.05s)
        **if** any agent is waiting for querying remote LLMs **then**
            Collect observations for these agents and wrap the observation into text form
            Generate independent threads and query remote LLMs in the threads
        **end if**
        **if** all agents have already got responses **then**
            **for** $i$ in agents' indexes **do**
                Recognize text actions and generate pysc2 actions into agent_i's data cache
                Recognize communication messages and send to assigned agents
                Move camera to the agent's unit team, execute generated pysc2 actions
            **end for**
        **end if**
    **end while**
    num_step += 1
**end while**
Store the final state and the game result(win/draw/lose)

---

## A.2 Query Process for an Agent

---

**Algorithm 2** Query Process for an Agent

---

**Require:** Max retry times $n$, max waiting time $T'$ for query.
Generate OpenAI message using collected image observation and text information
current retry time $i = 0$
**while** $i < n$ **do**
    Reset current waiting time $t$ to 0
    Initialize an independent thread and query remote LLM in the thread
    **while** $t < T'$ **do**
        time.sleep(0.05s), $t$ += 0.05s
        **if** response received successfully **then**
            Recognize valid actions and generate pysc2 functions for the agent
            Break the query process.
        **end if**
    **end while**
    waiting for $2^i$ seconds to avoid remote service error
    $i$ += 1
**end while**
return default action if no valid response received

---

# Appendix B. Action Space

Table B1: Default Protoss Action Space, Basic Actions

| Unit | Text action | pysc2 functions (id, function, args) |
|---|---|---|
| All unit | <No_Operation()> | (0, F.no_op, ()) |
| | <Hold_Position()> | (274, F.HoldPosition_quick, ('queued')) |
| | <Move_Screen(screen)> | (331, F.Move_screen, ('queued', 'screen')) |
| | <Move_Minimap(minimap)> | (332, F.Move_minimap, ('queued', 'minimap')) |
| | <Select_Unit_Move_Screen(screen)> | (3, F.select_rect, ('select', 'screen1_tag', 'screen2_tag')) |
| | | (331, F.Move_screen, ('now', 'screen')) |
| | <Select_Unit_Move_Minimap(minimap)> | (3, F.select_rect, ('select', 'screen1_tag', 'screen2_tag')) |
| | | (332, F.Move_minimap, ('queued', 'minimap')) |
| Attackable | <Attack_Unit(tag)> | (12, F.Attack_screen, ('queued', 'screen_tag')) |
| | <Select_Unit_Attack_Unit(tag, tag)> | (3, F.select_rect, ('select', 'screen1_tag', 'screen2_tag')) |
| | | (12, F.Attack_screen, ('queued', 'screen_tag')) |

Most units share the actions above, so we listed here to avoid mention repetitive unit control actions in subsequent appendices. Here, $F$ refers to $pysc2.lib.actions.FUNCTION$, the same as in the following.

Table B2: Default Protoss Action Space, Standard Building Actions

| Unit | Text action | pysc2 functions (id, function, args) |
|---|---|---|
| Probe | <Build_Nexus_Near(tag)> | (573, F.llm_pysc2_move_camera, ('world_tag')) |
| | | (65, F.Build_Nexus_screen, ('queued', 'screen_tag')) |
| | <Build_Assimilator_Near(tag)> | (573, F.llm_pysc2_move_camera, ('world_tag')) |
| | | (40, F.Build_Assimilator_screen, ('queued', 'screen_tag')) |
| | <Build_Nexus_Screen(screen)> | (65, F.Build_Nexus_screen, ('queued', 'screen_tag')) |
| | <Build_Assimilator_Screen(screen)> | (40, F.Build_Assimilator_screen, ('queued', 'screen_tag')) |
| | <Build_Pylon_Screen(screen)> | (70, F.Build_Pylon_screen, ('queued', 'screen')) |
| | <Build_Gateway_Screen(screen)> | (57, F.Build_Gateway_screen, ('queued', 'screen')) |
| | <Build_CyberneticsCore_Screen(screen)> | (48, F.Build_CyberneticsCore_screen, ('queued', 'screen')) |
| | <Build_Forge_Screen(screen)> | (55, F.Build_Forge_screen, ('queued', 'screen')) |
| | <Build_PhotonCannon_Screen(screen)> | (69, F.Build_PhotonCannon_screen, ('queued', 'screen')) |
| | <Build_ShieldBattery_Screen(screen)> | (525, F.Build_ShieldBattery_screen, ('queued', 'screen')) |
| | <Build_TwilightCouncil_Screen(screen)> | (101, F.Build_TwilightCouncil_screen, ('queued', 'screen')) |
| | <Build_TemplarArchive_Screen(screen)> | (100, F.Build_TemplarArchive_screen, ('queued', 'screen')) |
| | <Build_DarkShrine_Screen(screen)> | (49, F.Build_DarkShrine_screen, ('queued', 'screen')) |
| | <Build_Stargate_Screen(screen)> | (88, F.Build_Stargate_screen, ('queued', 'screen')) |
| | <Build_FleetBeacon_Screen(screen)> | (54, F.Build_FleetBeacon_screen, ('queued', 'screen')) |
| | <Build_RoboticsBay_Screen(screen)> | (81, F.Build_RoboticsBay_screen, ('queued', 'screen')) |
| | <Build_RoboticsFacility_Screen(screen)> | (82, F.Build_RoboticsFacility_screen, ('queued', 'screen')) |
| | <Lock_Nexus_Near(tag)> | (70, F.Build_Pylon_screen, ('queued', 'screen_tag')) |
| | <Lock_Assimilator_Near(tag)> | (40, F.Build_Assimilator_screen, ('queued', 'screen_tag')) |

In standard building mode, a worker will be chosen as building-worker at the beginning of the game or the time the worker dead. The 'Builder' agent will control the worker to build buildings using the actions mentioned above.

Table B3: Default Protoss Action Space, Building Actions (easy mode1, for Builder)

| Unit | Text action | pysc2 functions (id, function, args) |
|---|---|---|
| Probe | <Build_Nexus_Near(tag)> | (573, F.llm_pysc2_move_camera, ('world_tag')) |
| | | (65, F.Build_Nexus_screen, ('queued', 'screen_tag')) |
| | <Build_Assimilator_Near(tag)> | (573, F.llm_pysc2_move_camera, ('world_tag')) |
| | | (40, F.Build_Assimilator_screen, ('queued', 'screen_tag')) |
| | <Build_Pylon_Near(tag)> | (70, F.Build_Pylon_screen, ('queued', 'screen_tag')) |
| | <Build_Gateway_Near(tag)> | (57, F.Build_Gateway_screen, ('queued', 'screen_tag')) |
| | <Build_CyberneticsCore_Near(tag)> | (48, F.Build_CyberneticsCore_screen, ('queued', 'screen_tag')) |
| | <Build_Forge_Near(tag)> | (55, F.Build_Forge_screen, ('queued', 'screen_tag')) |
| | <Build_PhotonCannon_Near(tag)> | (69, F.Build_PhotonCannon_screen, ('queued', 'screen_tag')) |
| | <Build_ShieldBattery_Near(tag)> | (525, F.Build_ShieldBattery_screen, ('queued', 'screen_tag')) |
| | <Build_TwilightCouncil_Near(tag)> | (101, F.Build_TwilightCouncil_screen, ('queued', 'screen_tag')) |
| | <Build_TemplarArchive_Near(tag)> | (100, F.Build_TemplarArchive_screen, ('queued', 'screen_tag')) |
| | <Build_DarkShrine_Near(tag)> | (49, F.Build_DarkShrine_screen, ('queued', 'screen_tag')) |
| | <Build_Stargate_Near(tag)> | (88, F.Build_Stargate_screen, ('queued', 'screen_tag')) |
| | <Build_FleetBeacon_Near(tag)> | (54, F.Build_FleetBeacon_screen, ('queued', 'screen_tag')) |
| | <Build_RoboticsBay_Near(tag)> | (81, F.Build_RoboticsBay_screen, ('queued', 'screen_tag')) |
| | <Build_RoboticsFacility_Near(tag)> | (82, F.Build_RoboticsFacility_screen, ('queued', 'screen_tag')) |
| | <Lock_Nexus_Near(tag)> | (70, F.Build_Pylon_screen, ('queued', 'screen_tag')) |
| | <Lock_Assimilator_Near(tag)> | (40, F.Build_Assimilator_screen, ('queued', 'screen_tag')) |

In easy-build mode-1, the agent 'Builder' does not need to provide precision position, but a tag of nearby buildings. The LLM-PySC2 program will autonomously find a position near the unit with given tag and build new buildings there.

Table B4: Default Protoss Action Space, Building Actions (easy mode2, for Developer)

| Unit | Text action | pysc2 functions (id, function, args) |
|---|---|---|
| Probe | <Build_Nexus()> | (65, F.Build_Nexus_screen, ('queued', 'auto')) |
| | <Build_Assimilator()> | (40, F.Build_Assimilator_screen, ('queued', 'auto')) |
| | <Build_Pylon()> | (70, F.Build_Pylon_screen, ('queued', 'auto')) |
| | <Build_Gateway()> | (57, F.Build_Gateway_screen, ('queued', 'auto')) |
| | <Build_CyberneticsCore()> | (48, F.Build_CyberneticsCore_screen, ('queued', 'auto')) |
| | <Build_Forge()> | (55, F.Build_Forge_screen, ('queued', 'auto')) |
| | <Build_PhotonCannon()> | (69, F.Build_PhotonCannon_screen, ('queued', 'auto')) |
| | <Build_ShieldBattery()> | (525, F.Build_ShieldBattery_screen, ('queued', 'auto')) |
| | <Build_TwilightCouncil()> | (101, F.Build_TwilightCouncil_screen, ('queued', 'auto')) |
| | <Build_TemplarArchive()> | (100, F.Build_TemplarArchive_screen, ('queued', 'auto')) |
| | <Build_DarkShrine()> | (49, F.Build_DarkShrine_screen, ('queued', 'auto')) |
| | <Build_Stargate()> | (88, F.Build_Stargate_screen, ('queued', 'auto')) |
| | <Build_FleetBeacon()> | (54, F.Build_FleetBeacon_screen, ('queued', 'auto')) |
| | <Build_RoboticsBay()> | (81, F.Build_RoboticsBay_screen, ('queued', 'auto')) |
| | <Build_RoboticsFacility()> | (82, F.Build_RoboticsFacility_screen, ('queued', 'auto')) |

In easy-build mode-2, the agent 'Builder' does not need to provide any additional information of where to build the building. the program will automatically find a position for above actions. Experiments of ECEB mode and SCEB mode use these actions as the Developer's building actions.

Table B5: Default Protoss Action Space, Researching Actions

| Text action | pysc2 functions (id, function, args) |
| --- | --- |
| <Research_ProtossAirArmor()> | (381, F.Research_ProtossAirArmor_quick, ('queued')) |
| <Research_ProtossAirWeapons()> | (385, F.Research_ProtossAirWeapons_quick, ('queued')) |
| <Research_WarpGate()> | (428, F.Research_WarpGate_quick, ('queued')) |
| <Research_ProtossGroundArmor()> | (389, F.Research_ProtossGroundArmor_quick, ('queued')) |
| <Research_ProtossGroundWeapons()> | (393, F.Research_ProtossGroundWeapons_quick, ('queued')) |
| <Research_ProtossShields()> | (397, F.Research_ProtossShields_quick, ('queued')) |
| <Research_Charge()> | (359, F.Research_Charge_quick, ('queued')) |
| <Research_Blink()> | (356, F.Research_Blink_quick, ('queued')) |
| <Research_AdeptResonatingGlaives()> | (351, F.Research_AdeptResonatingGlaives_quick, ('queued')) |
| <Research_PhoenixAnionPulseCrystals()> | (379, F.Research_PhoenixAnionPulseCrystals_quick, ('queued')) |
| <Research_ExtendedThermalLance()> | (364, F.Research_ExtendedThermalLance_quick, ('queued')) |
| <Research_GraviticBooster()> | (366, F.Research_GraviticBooster_quick, ('queued')) |
| <Research_GraviticDrive()> | (367, F.Research_GraviticDrive_quick, ('queued')) |
| <Research_PsiStorm()> | (401, F.Research_PsiStorm_quick, ('queued')) |
| <Research_ShadowStrike()> | (404, F.Research_ShadowStrike_quick, ('queued')) |

Researching actions are actually complex actions of combination pysc2 functions, they require first selecting a research building and then starting the research. The program will autonomously find idle building for these functions, select the building and execute the pysc2 functions for technology upgrades.

Table B6: Default Protoss Action Space, Unit Training Actions

| Unit | Text action | pysc2 functions (id, function, args) |
| --- | --- | --- |
| Nexus | <Train_Mothership()> | (541, F.Train_Mothership_quick, ('queued')) |
| Gateway | <Train_Adept()> | (457, F.Train_Adept_quick, ('queued')) |
| | <Train_DarkTemplar()> | (465, F.Train_DarkTemplar_quick, ('queued')) |
| | <Train_HighTemplar()> | (471, F.Train_HighTemplar_quick, ('queued')) |
| | <Train_Sentry()> | (491, F.Train_Sentry_quick, ('queued')) |
| | <Train_Stalker()> | (493, F.Train_Stalker_quick, ('queued')) |
| | <Train_Zealot()> | (503, F.Train_Zealot_quick, ('queued')) |
| Stargate | <Train_Oracle()> | (482, F.Train_Oracle_quick, ('queued')) |
| | <Train_Phoenix()> | (484, F.Train_Phoenix_quick, ('queued')) |
| | <Train_VoidRay()> | (500, F.Train_VoidRay_quick, ('queued')) |
| | <Train_Tempest()> | (495, F.Train_Tempest_quick, ('queued')) |
| | <Train_Carrier()> | (461, F.Train_Carrier_quick, ('queued')) |
| RoboticBay | <Train_Observer()> | (481, F.Train_Observer_quick, ('queued')) |
| | <Train_WarpPrism()> | (501, F.Train_WarpPrism_quick, ('queued')) |
| | <Train_Immortal()> | (473, F.Train_Immortal_quick, ('queued')) |
| | <Train_Colossus()> | (462, F.Train_Colossus_quick, ('queued')) |
| | <Train_Disruptor()> | (466, F.Train_Disruptor_quick, ('queued')) |

Unit training actions share the same pre-process of researching actions. To avoid stocking a lot of resources in the training queue, the program only trains units in idle buildings. To avoid spending too many tokens on finding suitable buildings, the program autonomously searches idle buildings for these actions.

Table B7: Default Protoss Action Space, Unit Warp Actions and Warp Actions in easy mode

| Unit | Text action | pysc2 functions (id, function, args) |
|---|---|---|
| WarpGate | <Warp_Adept_Near(tag)> | (8, F.select_warp_gates, ('select')) |
| | | (573, F.llm_pysc2_move_camera, ('world_tag')) |
| | | (505, F.TrainWarp_Adept_screen, ('queued', 'screen_tag')) |
| | <Warp_DarkTemplar_Near(tag)> | (8, F.select_warp_gates, ('select')) |
| | | (573, F.llm_pysc2_move_camera, ('world_tag')) |
| | | (506, F.TrainWarp_DarkTemplar_screen, ('queued', 'screen_tag')) |
| | <Warp_HighTemplar_Near(tag)> | (8, F.select_warp_gates, ('select')) |
| | | (573, F.llm_pysc2_move_camera, ('world_tag')) |
| | | (507, F.TrainWarp_HighTemplar_screen, ('queued', 'screen_tag')) |
| | <Warp_Sentry_Near(tag)> | (8, F.select_warp_gates, ('select')) |
| | | (573, F.llm_pysc2_move_camera, ('world_tag')) |
| | | (505, F.TrainWarp_Sentry_screen, ('queued', 'screen_tag')) |
| | <Warp_Stalker_Near(tag)> | (8, F.select_warp_gates, ('select')) |
| | | (573, F.llm_pysc2_move_camera, ('world_tag')) |
| | | (506, F.TrainWarp_Stalker_screen, ('queued', 'screen_tag')) |
| | <Warp_Zealot_Near(tag)> | (8, F.select_warp_gates, ('select')) |
| | | (573, F.llm_pysc2_move_camera, ('world_tag')) |
| | | (507, F.TrainWarp_Zealot_screen, ('queued', 'screen_tag')) |
| WarpGate | <Warp_Zealot()> | (510, F.TrainWarp_Zealot_screen, ('queued', 'auto')) |
| | <Warp_Stalker()> | (509, F.TrainWarp_Stalker_screen, ('queued', 'auto')) |
| | <Warp_Sentry()> | (508, F.TrainWarp_Sentry_screen, ('queued', 'auto')) |
| | <Warp_Adept()> | (505, F.TrainWarp_Adept_screen, ('queued', 'auto')) |
| | <Warp_HighTemplar(screen)> | (507, F.TrainWarp_HighTemplar_screen, ('queued', 'auto')) |
| | <Warp_DarkTemplar(minimap)> | (506, F.TrainWarp_DarkTemplar_screen, ('queued', 'auto')) |

For protoss, some of the unit training actions will change to unit warpping actions after WarpGate technology upgrades. They need to choose a tag for power field provider (such as Pylon) to warp unit there. In easy-warp mode, the program will autonomously find valid position for unit warping actions.

Table B8: Default Protoss Action Space, Easy Control Actions

| Text action | pysc2 functions (id, function, args) |
|---|---|
| <All_Units_Attack()> | (13, F.Attack_minimap, ('auto')) |
| <All_Units_Defend()> | (331, F.Move_screen, ('queued', 'auto')) |
| <All_Units_Retreat()> | (331, F.Move_screen, ('now', 'auto')) |
| <Worker_Scan()> | (332, F.Move_minimap, ('queued', 'auto')) |
| <Zealot_Scan()> | (332, F.Move_minimap, ('queued', 'auto')) |
| <Adept_Scan()> | (332, F.Move_minimap, ('queued', 'auto')) |
| <Pheonix_Scan()> | (332, F.Move_minimap, ('queued', 'auto')) |
| <Oracle_Scan()> | (332, F.Move_minimap, ('queued', 'auto')) |
| <Observer_Scan()> | (332, F.Move_minimap, ('queued', 'auto')) |

We provide easy control actions, a series of actions similar to TextStarCraft-II unit control actions. For researchers who focus on studying LLM-based planning (develop the economy) or VLM-based decision making(precisely build buildings), simplifying unit control actions can provide great convenience.

Table B9: Default Protoss Action Space, Unit Skills (Part1, control a unit team)

| Unit | Text action | pysc2 functions (id, function, args) |
| --- | --- | --- |
| Adept | <Ability_AdeptPhaseShift_Minimap( minimap)> | (547, F.Effect_AdeptPhaseShift_minimap, ('now', 'minimap')) |
| | <Ability_AdeptPhaseShift_Screen(screen)> | (177, F.Effect_AdeptPhaseShift_screen, ('now', 'screen')) |
| | <Ability_CancelPhaseShift> | (141, F.Cancel_AdeptPhaseShift_quick, ('now')) |
| Stalker | <Ability_Blink_Screen(screen)> | (180, F.Effect_Blink_screen, ('now', 'screen')) |
| Sentry | <Ability_ForceField_Screen(screen)> | (193, F.Effect_ForceField_screen, ('queued', 'screen')) |
| | <Ability_GuardianShield()> | (197, F.Effect_GuardianShield_quick, ('queued')) |
| HighTeplar | <Ability_PsiStorm_Screen(screen)> | (218, F.Effect_PsiStorm_screen, ('queued', 'screen')) |
| | <Ability_PsiStorm_Attack_Unit(tag)> | (218, F.Effect_PsiStorm_screen, ('queued', 'screen_tag')) |
| | <Morph_Archon()> | (296, F.Morph_Archon_quick, ('queued')) |
| | <Select_Two_Units_Morph_Archon( tag, tag)> | (3, F.select_rect, ('select', 'screen1_tag', 'screen2_tag')) |
| | | (3, F.select_rect, ('select', 'screen1_tag2', 'screen2_tag2')) |
| | | (296, F.Morph_Archon_quick, ('queued')) |
| DarkTeplar | <Ability_ShadowStride_Unit(tag)> | (182, F.Effect_ShadowStride_screen, ('queued', 'screen_tag')) |
| | <Morph_Archon()> | (296, F.Morph_Archon_quick, ('queued')) |
| Observer | <Morph_SurveillanceMode()> | (538, F.Morph_SurveillanceMode_quick, ('queued')) |
| | <Morph_ObserverMode()> | (535, F.Morph_ObserverMode_quick, ('queued')) |
| Disruptor | <Ability_PurificationNova_Attack(tag)> | (219, F.Effect_PurificationNova_screen, ('queued', 'screen_tag')) |
| Oracle | <Ability_PulsarBeamOn()> | (38, F.Behavior_PulsarBeamOn_quick, ('queued')) |
| | <Ability_OracleRevelation_Screen(screen)> | (214, F.Effect_OracleRevelation_screen, ('queued', 'screen')) |
| | <Build_StasisTrap_Screen(screen)> | (90, F.Build_StasisTrap_screen, ('queued', 'screen')) |
| Pheoenix | <Ability_GravitonBeam> | (196, F.Effect_GravitonBeam_screen |
| | <Cancel_GravitonBeam_For_All()> | (140, F.Cancel_quick, ('now')) |
| WarpPrism | <Morph_WarpPrismPhasingMode()> | (329, F.Morph_WarpPrismPhasingMode_quick, ('queued')) |
| | <Load_Unit(tag)> | (287, F.Load_screen, ('queued', 'screen_tag')) |
| | <Unload_Screen(screen)> | (516, F.UnloadAllAt_screen, ('queued', 'screen')) |
| | <Morph_WarpPrismTransportMode> | (330, F.Morph_WarpPrismTransportMode_quick , ('queued')) |
| MotherShip | <Ability_TimeWarp_Attack(tag)> | (241, F.Effect_TimeWarp_screen, ('queued', 'screen_tag')) |
| | <Ability_TimeWarp_Screen(screen)> | (241, F.Effect_TimeWarp_screen, ('queued', 'screen')) |

In LLM-PySC2 environment, agent controls its unit in a group, using above actions. The program will select the units belong to the agent, and then execute LLM-generated actions.

In some senerios, precisely controlling single unit is key to the victory, especially in SMAC tasks and early stage of the game (against high level opponent). We provide single unit control actions for this senarios, and LLM can use the actions whenever they need to improve performance of micro-operations.

Table B10: Default Protoss Action Space, Unit Skills (Part2, control specific unit)

| Unit | Text action | pysc2 functions (id, function, args) |
|---|---|---|
| Adept | \<Select_Unit_Ability_AdeptPhaseShift _Minimap(minimap)> | (3, F.select_rect, ('select', 'screen1_tag', 'screen2_tag')) (547, F.Effect_AdeptPhaseShift_minimap, ('now', 'minimap')) |
| | \<Select_Unit_Ability_AdeptPhaseShift _Screen(screen)> | (3, F.select_rect, ('select', 'screen1_tag', 'screen2_tag')) (177, F.Effect_AdeptPhaseShift_screen, ('now', 'screen')) |
| | \<Select_Unit_Ability_CancelPhaseShift( tag)> | (3, F.select_rect, ('select', 'screen1_tag', 'screen2_tag')) (141, F.Cancel_AdeptPhaseShift_quick, ('now')) |
| Stalker | \<Select_Unit_Blink_Screen(tag, screen)> | (3, F.select_rect, ('select', 'screen1_tag', 'screen2_tag')) (180, F.Effect_Blink_screen, ('now', 'screen')) |
| Sentry | \<Select_Unit_Ability_ForceField_Screen( tag, screen)> | (3, F.select_rect, ('select', 'screen1_tag', 'screen2_tag')) (193, F.Effect_ForceField_screen, ('queued', 'screen')) |
| | \<Select_Unit_Ability_GuardianShield( tag)> | (3, F.select_rect, ('select', 'screen1_tag', 'screen2_tag')) (197, F.Effect_GuardianShield_quick, ('queued')) |
| HighTeplar | \<Select_Two_Units_Morph_Archon( tag, tag)> | (3, F.select_rect, ('select', 'screen1_tag', 'screen2_tag')) (3, F.select_rect, ('add', 'screen1_tag2', 'screen2_tag2')) (296, F.Morph_Archon_quick, ('queued')) |
| | \<Select_Unit_Ability_PsiStorm_Screen (tag, screen)> | (3, F.select_rect, ('select', 'screen1_tag', 'screen2_tag')) (218, F.Effect_PsiStorm_screen, ('queued', 'screen')) |
| | \<Select_Unit_Ability_PsiStorm_Attack_Unit( (tag, tag)> | (3, F.select_rect, ('select', 'screen1_tag', 'screen2_tag')) (218, F.Effect_PsiStorm_screen, ('queued', 'screen_tag')) |
| Disruptor | \<Select_Unit_Ability_PurificationNova _Attack(tag)> | (3, F.select_rect, ('add', 'screen1_tag2', 'screen2_tag2')) (219, F.Effect_PurificationNova_screen, ('queued', 'screen_tag')) |
| DarkTeplar | \<Select_Two_Units_Morph_Archon( tag, tag)> | (3, F.select_rect, ('select', 'screen1_tag', 'screen2_tag')) (3, F.select_rect, ('add', 'screen1_tag2', 'screen2_tag2')) (296, F.Morph_Archon_quick, ('queued')) |
| Oracle | \<Select_Unit_Ability_PulsarBeamOn(tag)> | (3, F.select_rect, ('select', 'screen1_tag', 'screen2_tag')) (38, F.Behavior_PulsarBeamOn_quick, ('queued')) |
| | \<Select_Unit_OracleRevelation_Screen( screen)> | (3, F.select_rect, ('select', 'screen1_tag', 'screen2_tag')) (214, F.Effect_OracleRevelation_screen, ('queued', 'screen')) |
| | \<Select_Unit_Build_StasisTrap_Screen( tag, screen)> | (3, F.select_rect, ('select', 'screen1_tag', 'screen2_tag')) (90, F.Build_StasisTrap_screen, ('queued', 'screen')) |
| Pheoenix | \<Select_Phoenix_Ability_GravitonBeam> _Unit(tag) | (3, F.select_rect, ('select', 'screen1_tag', 'screen2_tag')) (196, F.Effect_GravitonBeam_screen, ('queued', 'screen_tag2')) |
| | \<Cancel_GravitonBeam_For_Phoenix(tag)> | (3, F.select_rect, ('select', 'screen1_tag', 'screen2_tag')) (140, F.Cancel_quick, ('now')) |

# Appendix C. Query message, Prompt, Examples of Observations and Responses

## C1. Query message and Prompt

**LLM Query Message**

```
messages = [
    {"role": "system", "content": system_prompt},
    {"role": "user", "content": example_i_prompt},
    {"role": "assistant", "content": example_o_prompt},
    {"role": "user", "content": text_observation},
]
```

**LLM Query Message with image observation**

```
messages = [
    {"role": "system", "content": system_prompt},
    {"role": "user", "content": example_i_prompt},
    {"role": "assistant", "content": example_o_prompt},
    {"role": "user", "content": [
        {"type": "text", "text": f'This is the {img_name1} image:'},
        {"type": "image_url", "image_url":
            {"url": f"data:image/png;base64,{base64_images[img_name1]}"}}
    ]}
    {"role": "user", "content": [
        {"type": "text", "text": f'This is the {img_name2} image:'},
        {"type": "image_url", "image_url":
            {"url": f"data:image/png;base64,{base64_images[img_name2]}"}}
    ]}
    .......
    {"role": "user", "content": text_observation},
]
```

**System Prompt**

```
1.Identity
  You are a {agent.config.AGENTS[agent.name]['describe']}.
  Your should command your units, complete the tasks assigned by the superior agent. You may have
  several teams of units, you can command these teams to fight together or perform different tasks.

2.Rules
  {get_rules(agent_name)}

3.Action Output
  You should make decisions according to observed information, tactic task and rules, give analysis and
  decisions for each team. For example, if you have 2 teams name as 'TeamName-1' and 'TeamName-2',
  you should output as:

  Analysis:
   xxxxx
  Actions:
   Team TeamName-1:
     <ActionName1(...)>  # **ActionName1()** or -ActionName1()- are invalid format, must use < >
     <ActionName2(...)>
   Team TeamName-2:
   <ActionName1(...)>

Note that actions must in the shape <ActionName(...)>, do not generate action like 'ActionName(...)' or
**ActionName(...)**.
```

Figure C1: **OpenAI LLM query message and system prompt in LLM-PySC2.**

**Decision Rules of the System Prompt**

BASIC_COMMAND_RULES = """
  Analyse following aspect in your decision process:
  1. (Combat Deployment) Analyse the situation, always make deployment for attack/defend/retreat at each step. As default choice, ask your units to defend your area. If you have enough units, rise attack to defeat your enemy.
  2. (Scan Deployment) If you are prepared for combat, make scan deployment to find out enemy's strengths. Note that unit for scan will be killed by enemy units.
  3. (Final Combat) If game time > 12:00 and you have enough units (DO NOT have to be 200 supply), raise attack to defeat the enemy."""

BASIC_DEVELOP_RULES = """
  Analyse following aspect in your decision process:
  1. (Supply) If run out of supply (less than 10) and no building for supply is under construction, build Pylon/OverLord/SupplyDepot (depend on your race).
  2. (Economy Building: Base) If you have enough minerals (more than 400), build Nexus/Hatchery/CommandCenter (depend on your race).
  3. (Economy Building: Gas) If you run out of gas (much less than minerals), build Assimilator/Extractor/Refinery (depend on your race).
  4. (Building: Unit Training) If you have too less unit training buildings, or there are abundant resources but all the unit training buildings are working, build unit training buildings.
  5. (Building: Research) If you have too less research buildings, or there are abundant resources but all the research buildings are working, build research buildings.
  6. (Unit Training/Warping) If you have enough idle unit training buildings but few combat units, or have a lot resource, train/warp units as much as possible.
  7. (Tech Upgrading) If you have idle research buildings and enough resource, or have a lot resource, update your technology.
  8. (Early Stage Expand) If you do not have 'the second base building (such as Nexus)', the 'CyberneticCore', the 'TwilightCouncil' and 'first two Gateway', try to build them as quick as possible.
  9. (Middle Stage Develop) During the middle stage of the game, try to build buildings for training high value units, and train units as much as possible (especially high value units) to increase strength.
  10. (Final Stage Develop) During the final stage of the game, train or warp more units to fight with enemy, do not build building if we have enough buildings."""

BASIC_BUILD_RULES = """
  Analyse following aspect in your decision process:
  1. (Minimap Position) According to image 'rgb_minimap', where is/are our base/bases and where should we go? give minimap position. (Our base units and buildings are green points/squares in the minimap)
  2. (Build) You can build on any position of the screen (unless it is blocked by other buildings) direct use the action <Build_XXX_Screen([x, y])>. You can build more than one buildings at a step by generate many action <Build_XXX_Screen([x, y])>.
  3. (Move) You should move to a plain location near the base building, and build buildings there. Don't be far away from the base building, keep base building in your sight(screen), unless you are building a new one.
  4. (Actions Sequence) First <Build_xxx_Screen(screen)> or <Build_xxx_Near(tag)> then <Move_xxx(xxx)> (note that, do not move and build at the same screen position)"""

BASIC_COMBAT_RULES = \"""
  1. Concentrating firepower is always necessary, attack different unit at the same time will definitely reduce killing speed and leading to terrible result. Always concentrating all teams' fire at one unit that (1)with highest DPS(most valuable) (2)most vulnerable (3)closest."""

Figure C2: **Basic Rules for agent Commander, Developer, Builder, CombaGroups.**

## C2. Examples of Textual Observations

---

**Example of Observation Prompt (Agent Commander, Easy Control Mode)**

Game Info:
  Time: 0:20
  Minerals: 1190 \n\t Vespene: 0
  Supply Total: 15 \n\t Supply Left: -1 \n\t Supply Used: 16

Unit Counts:
  Our Unit:
    {'Probe': 29, 'Adept': 2}
  Our Buildings:
    {'Pylon': 2, 'Assimilator': 1, 'Gateway': 2, 'CyberneticsCore': 1, 'Nexus': 2}
  Our Unit (in warping/morphing):
    None
  Our Buildings (in construction):
    None
  Military Buildings:
    {'Gateway': '2 (1 is working, 1 is idle)'}
  Research Buildings:
    {'CyberneticsCore': '1 (0 is working, 1 is idle)'}
  Spotted Enemy Unit:
    {'SCV': 1}

Valid Actions
  Team Protoss-Units-1:
  <All_Units_Attack()>
  <All_Units_Defend()>
  <All_Units_Retreat()>
  <Worker_Scan()>
  <Adept_Scan()>

Last Step Actions:
  <All_Units_Defend()>

Available Communication Targets:
  Developer: Protoss logistics commander. Responsible for unit trainning, unit warp trainning,
  technology upgrade and order the Builder to build.
Available Communication Functions:
  <MessageTo(AgentName, message)>
  <MessageTo(ChannelName, message)>
  <ListenTo(ChannelName)>
Args explanation:
  (1)AgentName: refers to a name mentioned in Available Communication Targets.
  (2)ChannelName: shape as Channel-i, i refers to an integer.
  (2)message: any text wrapped between "' and "'.

Tasks:
Team Protoss-Units' task: Command your units through 'Actions' to defeat the enemy.
Please note that **Tasks** are the most important information, all your decisions must aimed at
completing the tasks.

As the supreme military commander, you should not directly give actions, instead, tell your
subordinates what to do through communication.
Now, start analysis, making macro decisions in military deployments:

---

Figure C3: **Example textual observation of agent 'Commander' in easy control mode.**

**Example of Observation Prompt (Agent Commander, Standard Control Mode)**

Game Info:{game_info}

Unit Counts: {unit_count_info}

Global agent info:
  Agent CombatGroup4: Team Probe-0: Protoss.Probe x1, minimap position [13, 31]
  Agent CombatGroup7: Team Adept-1: Protoss.Adept x2, minimap position [21, 31]

Relevant Knowledge:
  Protoss.Probe
    The builder of the protoss race. Gathers gas and minerals.
    Unit properties: ['ground', 'light', 'mechanical']
    Weapon info: Attack Range 0.2, target: ['ground'], DPS(damage per second) 3
    unit abilities:
      WorkerAbilities: Always available. The probe warps in structures and harvests minerals and
      vespene gas.
  Protoss.Adept
    Ground-only ranged attack unit, armed with psionic transfer(AdeptPhaseShift) ability to teleport to
    nearby locations for harassment.
    Unit properties: ['ground', 'light', 'biological'], Weapon info: Attack Range 4, target: ['ground'], anti:
    ['light'], DPS(damage per second) 4, DPS-anti 9
    unit abilities:
      AdeptPhaseShift: Active skill.Always avaliable. Cooldowm 11 seconds.Projects an invulnerable
      psionic image that can move but not attack. After 7 seconds, the adept teleports to the images
      location. The shade may be canceled at any time, and the adept would not teleport. The shade has
      a sight radius of 2.

Available Communication Targets:
    Developer: Protoss logistics commander. Responsible for unit trainning, unit warp trainning,
      technology upgrade and order the Builder to build.
    CombatGroup4: Protoss reconnaissance commander, controls Observer and several Probe.
      Responsible for providing reconnaissance infomation and detect cloak unit for main force
    CombatGroup7: Protoss special force commander, controls Adept and DarkTemplar. Responsible
      for infiltrating the enemy's rear and disrupt economic production, sometimes collecting
      reconnaissance infomation, participating in frontline combat.
Available Communication Functions:
    <MessageTo(AgentName, message)>
    <MessageTo(ChannelName, message)>
    <ListenTo(ChannelName)>
Args explanation:
    (1)AgentName: refers to a name mentioned in Available Communication Targets.
    (2)ChannelName: shape as Channel-i, i refers to an integer.
    (2)message: any text wrapped between '" and '".

Tasks:
    Team Protoss-Units' task: Command your units through 'Actions' to defeat the enemy.
    Please note that **Tasks** are the most important information, all your decisions must aimed at
    completing the tasks.

As the supreme commander, you should not directly give actions, instead, tell your subordinates what to
do through communication.
Now, start analysis, making macro decisions in military deployments by sending message to other
agents:

Figure C4: **Example textual observation of agent 'Commander' in standard control mode.**

**Example of Observation Prompt (Agent CombatGroup7, Standard Control Mode)**

Game Info:
  Time: 3:50

Team Adept-1 Info:
  Team minimap position: [33, 41] (minimap coordinate valid range for actions: 0 < x < 64, 0 < y < 64)
  Team screen edge (screen coordinate valid range for actions: 0 < x < 23, 0 < y < 23)
  Controlled Team Units:
    Unit: Adept   Tag: 0x102180001   ScreenPos: [11, 12]   Health: 140(100 %)
  Nearby Ally Units:
    Unit: Probe   Tag: 0x101340001   ScreenPos: [22, 16]   Health: 40(100 %)
    Unit: Probe   Tag: 0x101bc0001   ScreenPos: [22, 17]   Health: 40(100 %)
    Unit: Probe   Tag: 0x101940001   ScreenPos: [22, 16]   Health: 40(100 %)
    Unit: Probe   Tag: 0x101280001   ScreenPos: [20, 12]   Health: 40(100 %)
    Unit: Nexus   Tag: 0x101200001   ScreenPos: [19, 16]   Health: 2000(100 %)   Energy: 131
    Unit: Pylon   Tag: 0x1018c0001   ScreenPos: [12, 15]   Health: 400(100 %)
    Unit: Gateway   Tag: 0x101b00001   ScreenPos: [14, 12]   Health: 1000(100 %)

Last Step Event:
  Team Adept-1 Event:
    Ally Unit Event:
      unit 0x101440001(Protoss.Probe) ally unit enter sight
      unit 0x101d40001(Protoss.Probe) ally unit enter sight

Relevant Knowledge:{relevant knowledge}

Valid Actions:
  Team Adept-1 Valid Actions:
    <Move_Minimap(minimap)>
    <Move_Screen(screen)>
    <Attack_Unit(tag)>
    <Select_Unit_Attack_Unit(tag, tag)>
    <Select_Unit_Move_Screen(tag, screen)>
    <Ability_AdeptPhaseShift_Minimap(minimap)>

Action Args:
  (1) tag: tag refers to a hexadecimal number, shape as 0x000000000.
  (2) screen: screen refers to a screen coordinate, shape as [x, y], where 0 < x < 23, 0 < y < 23.
  (3) minimap: minimap refers to a minimap coordinate, shape as [x, y], where x and y range from 0 to 64.

Available Communication Targets: {available_communication_targets}
Available Communication Functions: {communication_functions}
Args explanation: {communication_functions_args_explanation}

Tasks:
  Team Adept-1' task: Controls your unit teams through 'Actions' to fight with enemies or support allys.
  Complete tasks assigned by the Commander
Please note that **Tasks** are the most important information, all your decisions must aimed at completing the tasks.

Give each team no more than 3 actions, these actions will be executed in the following 10.0 seconds, among which activity release should usually before attack and move.
Now, start generating your analysis, strategy, actions and communication:

Figure C5: **Example textual observation of agent 'CombatGroup7' in standard control mode.**

**Example of Observation Prompt (Agent Developer, Easy Build Mode)**

Game Info:{game_info}

Unit Counts: {unit_count_info}

Valid actions:
  Team Protoss-Buildings-1:
    <Research_ProtossAirArmor()>     cost: {'mineral': 100, 'gas': 100}
    <Research_ProtossAirWeapons()>  cost: {'mineral': 100, 'gas': 100}
    <Research_WarpGate()>            cost: {'mineral': 50,  'gas': 50}
    <Train_Adept()>                  cost: {'mineral': 100, 'gas': 25,  'supply': 2}
    <Train_Sentry()>                 cost: {'mineral': 50,   'gas': 100, 'supply': 2}
    <Train_Stalker()>                cost: {'mineral': 125, 'gas': 50,  'supply': 2}
    <Train_Zealot()>                 cost: {'mineral': 100, 'gas': 0,    'supply': 2}
  Team Protoss-Workers-1:
    <Build_Nexus()>                  cost: {'mineral': 400, 'gas': 0}
    <Build_Assimilator()>            cost: {'mineral': 75,   'gas': 0}
    <Build_Pylon()>                  cost: {'mineral': 100, 'gas': 0}
    <Build_Gateway()>                cost: {'mineral': 150, 'gas': 0}
    <Build_CyberneticsCore()>        cost: {'mineral': 150, 'gas': 0}
    (note: 'New Buildings!' in the following part We do not have this building yet, it may unlock new buildings/technologies/units for us')
    <Build_Forge()>                  cost: {'mineral': 150, 'gas': 0}
    <Build_ShieldBattery()>          cost: {'mineral': 100, 'gas': 0}
    <Build_TwilightCouncil()>        cost: {'mineral': 150, 'gas': 100}
    <Build_Stargate()>               cost: {'mineral': 150, 'gas': 150}
    <Build_RoboticsFacility()>       cost: {'mineral': 150, 'gas': 100}

Relevant Knowledge:
  {relevant knowledge}

Available Communication Targets:
  Commander:
    Protoss military supreme commander. Responsible for making macro decision through communication, and controls nexus for massrecall for tactical objectives. When make deployment, describe the time, location, and objectives of the mission as clearly as possible
Available Communication Functions:
  {communication_functions}
Args explanation:
  {communication_functions_args_explanation}

Tasks:
  Team Protoss-Buildings' task:
    Organize the agent 'Builder' through 'Communication' to build buildings. Develop economy, technology, and train units through 'Actions' to win the game.
  Team Protoss-Workers' task:
    Organize the agent 'Builder' through 'Communication' to build buildings. Develop economy, technology, and train units through 'Actions' to win the game.
Please note that **Tasks** are the most important information, all your decisions must aimed at completing the tasks.

As a senior commander, the max number of your actions is not limited, when you warp units, try to use all the WarpGate as much as possible, and warp all units near a single WarpTrain Field Provider.
Now, start generating your analysis, actions and communication:

Figure C6: **Example textual observation of agent 'Developer' in easy build mode.**

**Example of Observation Prompt (Agent Developer, Standard Build Mode)**

Game Info:{game_info}

Unit Counts: {unit_count_info}

Valid actions:
  Team Protoss-Buildings-1:
    <Research_ProtossAirArmor()>    cost: {'mineral': 100, 'gas': 100}
    <Research_ProtossAirWeapons()>  cost: {'mineral': 100, 'gas': 100}
    <Research_WarpGate()>        cost: {'mineral': 50,  'gas': 50}
    <Train_Adept()>             cost: {'mineral': 100, 'gas': 25,  'supply': 2}
    <Train_Sentry()>           cost: {'mineral': 50,   'gas': 100, 'supply': 2}
    <Train_Stalker()>          cost: {'mineral': 125, 'gas': 50,  'supply': 2}
    <Train_Zealot()>          cost: {'mineral': 100, 'gas': 0,    'supply': 2}

Agent Builder's Valid actions:
  <Build_Nexus_Near(tag)>                      cost: {'mineral': 400, 'gas': 0}
  <Build_Assimilator_Near(tag)>             cost: {'mineral': 75,  'gas': 0}
  <Build_Pylon_Screen(screen)>             cost: {'mineral': 100, 'gas': 0}
  <Build_Gateway_Screen(screen)>          cost: {'mineral': 150, 'gas': 0}
  <Build_CyberneticsCore_Screen(screen)>    cost: {'mineral': 150, 'gas': 0}
  (note: 'New Buildings!' in the following part We do not have this building yet, it may unlock new buildings/technologies/units for us')
  <Build_Forge_Screen(screen)>             cost: {'mineral': 150, 'gas': 0}
  <Build_ShieldBattery_Screen(screen)>      cost: {'mineral': 100, 'gas': 0}
  <Build_TwilightCouncil_Screen(screen)>    cost: {'mineral': 150, 'gas': 100}
  <Build_Stargate_Screen(screen)>          cost: {'mineral': 150, 'gas': 150}
  <Build_RoboticsFacility_Screen(screen)>    cost: {'mineral': 150, 'gas': 100}

Action Args: {action_arg_explanation}

Relevant Knowledge:{relevant knowledge}

Available Communication Targets:
  Builder:  Protoss builder, controls several Probe. Responsible for build buildings
  Commander:
    Protoss military supreme commander. Responsible for making macro decision through communication, and controls nexus for massrecall for tactical objectives. When make deployment, describe the time, location, and objectives of the mission as clearly as possible
Available Communication Functions:
  {communication_functions}
Args explanation:
  {communication_functions_args_explanation}

Tasks:
  Team Protoss-Buildings' task:
    Organize the agent 'Builder' through 'Communication' to build buildings. Develop economy, technology, and train units through 'Actions' to win the game.
Please note that **Tasks** are the most important information, all your decisions must aimed at completing the tasks.

As a senior commander, the max number of your actions is not limited, when you warp units, try to use all the WarpGate as much as possible, and warp all units near a single WarpTrain Field Provider.
Now, start generating your analysis, actions and communication:

Figure C7: **Example textual observation of agent 'Developer' in standard build mode.**

**Example of Observation Prompt (Agent Builder, Standard Build Mode)**

Game Info:{game_info}

Unit Counts: {unit_count_info}

Team Builder-Probe Info:
  Team minimap position: [40, 44] (minimap coordinate valid range for actions: 0 < x < 64, 0 < y < 64)
  Team screen edge (screen coordinate valid range for actions: 0 < x < 17, 0 < y < 23)
  Controlled Team Units:
    Unit: Probe    Tag: 0x101340001    ScreenPos: [12, 12]    Health: 40(100 %)
  Nearby Ally Buildings:
    Unit: Nexus    Tag: 0x101200001    ScreenPos: [9, 12]    Health: 2000(100 %)    Energy: 139
    Unit: Pylon    Tag: 0x101c80001    ScreenPos: [12, 1]    Health: 400(100 %)
    Unit: Pylon    Tag: 0x1018c0001    ScreenPos: [3, 10]    Health: 400(100 %)
    Unit: Assimilator    Tag: 0x101a00001    ScreenPos: [16, 9]    Health: 600(100 %)
    Unit: Gateway    Tag: 0x101b00001    ScreenPos: [4, 8]    Health: 1000(100 %)
    Unit: Gateway    Tag: 0x102080001    ScreenPos: [9, 6]    Health: 790(79 %)

Unit Counts:
  Our Buildings: {'CyberneticsCore': 1, 'Nexus': 2, 'Gateway': 2, 'Pylon': 2, 'Assimilator': 1}
  Our Buildings (in construction): {'Gateway': '1 in total (0x102080001 79%)'}

Last Step Event:
  Team Builder-Probe Event:
    Ally Unit Event:
      unit 0x102080001(Protoss.Gateway) is training/building, process +22%  (current process 79%)

Relevant Knowledge:{relevant knowledge}

Valid Actions:
  Team Builder-Probe:
    <Move_Minimap(minimap)>
    <Move_Screen(screen)>
    <Build_Nexus_Near(tag)>                          cost: {'mineral': 400, 'gas': 0}
    <Build_Assimilator_Near(tag)>                    cost: {'mineral': 75,  'gas': 0}
    ......
    (note: 'New Buildings!' in the following part We do not have this building yet, it may unlock new
    buildings/technologies/units for us')
    <Build_TwilightCouncil_Screen(screen)>           cost: {'mineral': 150, 'gas': 100}
    ......

Action Args: {action_arg_explanation}
Available Communication Targets: {available_communication_targets}
Available Communication Functions: {communication_functions}
Args explanation: {communication_functions_args_explanation}

Tasks:
  Team Builder-Probe' task: Build buildings through 'Actions', Complete tasks assigned by the
  Developer or Commander, or based on your own judgment.
Please note that **Tasks** are the most important information, all your decisions must aimed at
completing the tasks.

As a builder, you need to move the worker to an open location and complete the construction of the
building.Now, start generating your analysis, actions and communication:

Figure C8: **Example textual observation of agent 'Builder' in standard build mode.**

**C3. Examples of Image Observation**

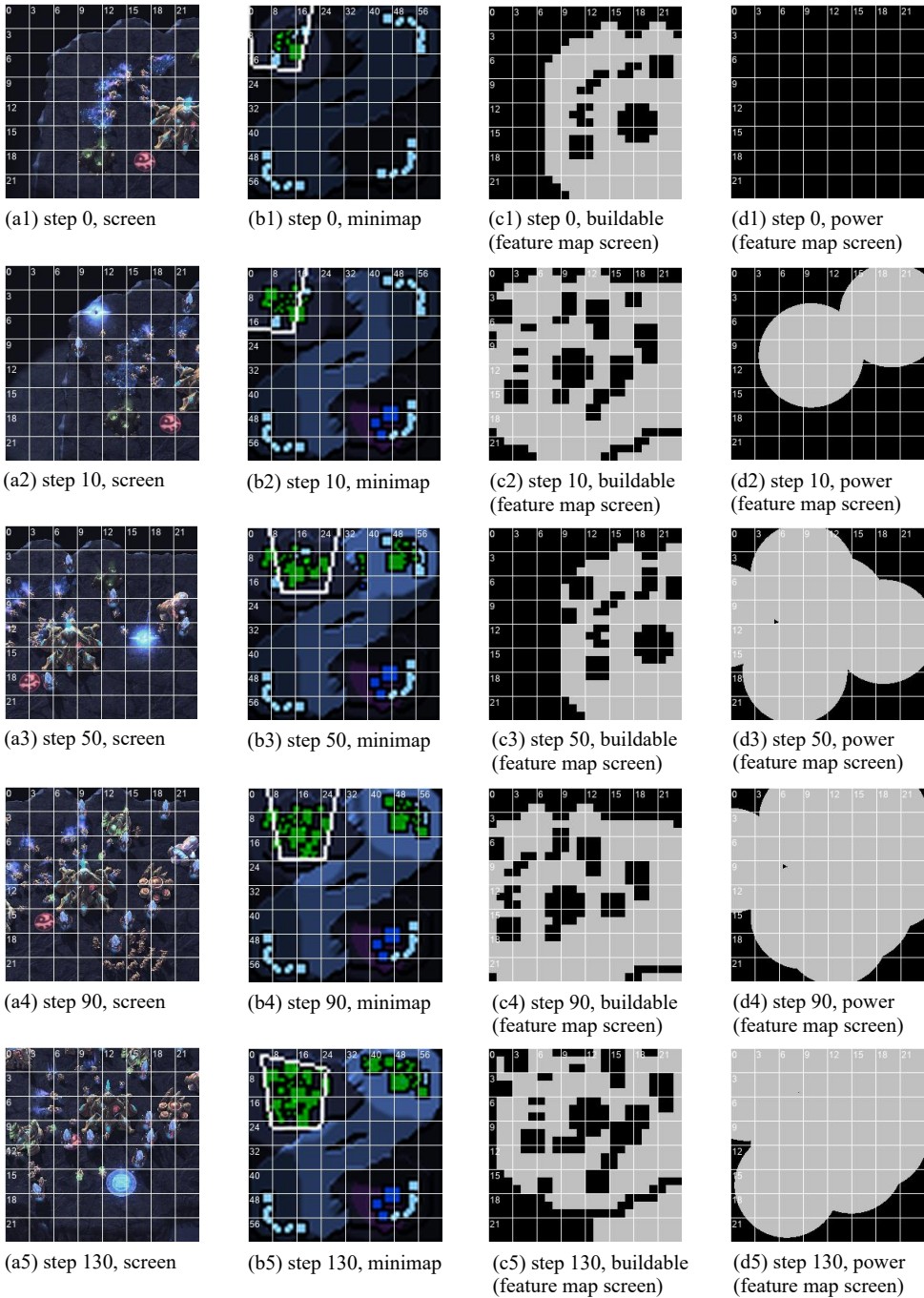

(a1) step 0, screen      (b1) step 0, minimap      (c1) step 0, buildable (feature map screen)      (d1) step 0, power (feature map screen)

(a2) step 10, screen      (b2) step 10, minimap      (c2) step 10, buildable (feature map screen)      (d2) step 10, power (feature map screen)

(a3) step 50, screen      (b3) step 50, minimap      (c3) step 50, buildable (feature map screen)      (d3) step 50, power (feature map screen)

(a4) step 90, screen      (b4) step 90, minimap      (c4) step 90, buildable (feature map screen)      (d4) step 90, power (feature map screen)

(a5) step 130, screen      (b5) step 130, minimap      (c5) step 130, buildable (feature map screen)      (d5) step 130, power (feature map screen)

Figure C9: **Examples of image observation of agent 'Builder' in standard control mode.**

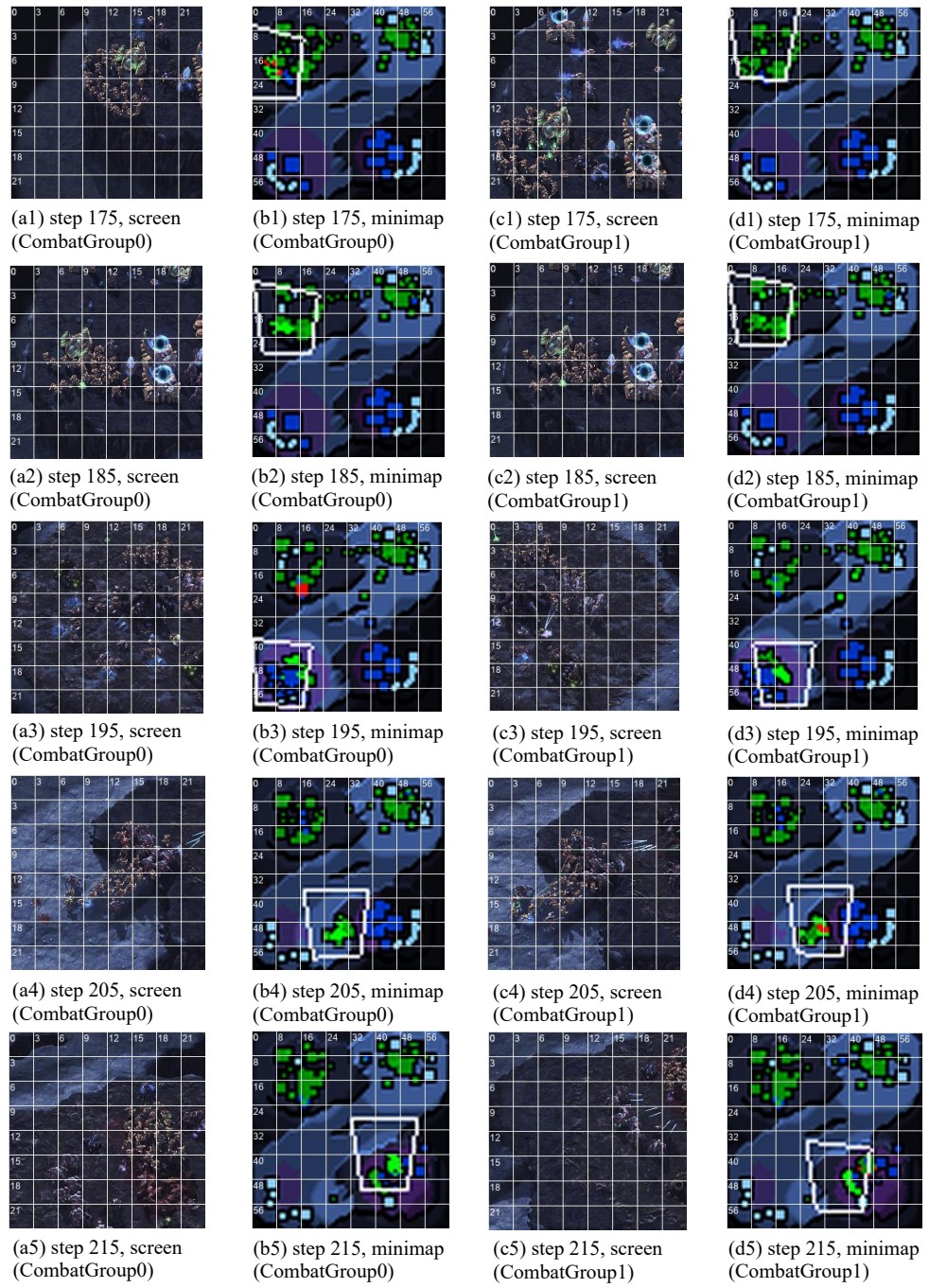

(a1) step 175, screen (CombatGroup0)

(b1) step 175, minimap (CombatGroup0)

(c1) step 175, screen (CombatGroup1)

(d1) step 175, minimap (CombatGroup1)

(a2) step 185, screen (CombatGroup0)

(b2) step 185, minimap (CombatGroup0)

(c2) step 185, screen (CombatGroup1)

(d2) step 185, minimap (CombatGroup1)

(a3) step 195, screen (CombatGroup0)

(b3) step 195, minimap (CombatGroup0)

(c3) step 195, screen (CombatGroup1)

(d3) step 195, minimap (CombatGroup1)

(a4) step 205, screen (CombatGroup0)

(b4) step 205, minimap (CombatGroup0)

(c4) step 205, screen (CombatGroup1)

(d4) step 205, minimap (CombatGroup1)

(a5) step 215, screen (CombatGroup0)

(b5) step 215, minimap (CombatGroup0)

(c5) step 215, screen (CombatGroup1)

(d5) step 215, minimap (CombatGroup1)

Figure C10: **Examples of image observation of agent 'CombatGroup0' (controls Zealots) and agent 'CombatGroup1' (controls Stalkers) in standard control mode.**

## C4. Examples of LLM Responses

---

**Example of Response (Agent Commander, Easy Control Mode)**

Analysis:
  The enemy is currently defending with a small group of units, including a Stalker and a Zealot. The enemy base has minimal defense, with only one Warp Gate and Pylon in the area. Our Zealots have already begun the assault on enemy workers, but reinforcements are still on the way. We need to reinforce our offensive, manage resources, and ensure the continued supply of troops.
  Our current resources (340 minerals and 502 vespene) are enough to reinforce with additional units if necessary. The Warp Gates are idle, so more reinforcements can be warped in. Additionally, the Shield Battery and Pylon construction are progressing, which will help maintain our position. However, the limited supply (26 supply left) means we must carefully consider which units to produce.

Actions:
  Team Protoss-Units-1:
   <All_Units_Attack()>

Communications:
   <MessageTo(Developer, "'Request reinforcements: warp in more Zealots to sustain the attack on enemy workers. Consider warping in additional Stalkers as well for support against any enemy Stalker reinforcements. The Shield Battery and Pylon construction must be prioritized for sustainability. Deploy more Probes for resource gathering if necessary to maintain unit production. Proceed with caution to avoid detection by enemy Queens in the vicinity. We need to overwhelm the enemy quickly.'")>

---

Figure C11: **Example response of agent 'Commander' in easy control mode.**

---

**Example of Response (Agent Developer, Easy Build Mode)**

Analysis:
  We have a strong mineral and gas economy with 2820 minerals and 282 vespene gas. Our supply is currently at 58 out of 102, leaving us with 44 supply available. We have 2 Gateways that are both currently working, which limits our unit production. The Commander has instructed us to initiate the Warp Gate technology upgrade and train additional Zealots, as well as construct an extra Gateway to enhance our unit production capabilities. Given our current resources and the need to bolster our forces, we should prioritize the Warp Gate research, train as many Zealots as possible, and build an additional Gateway.

Actions:
  Team Protoss-Buildings-1:
   <Research_WarpGate()>
   <Train_Zealot()>
   <Build_Gateway()>

Communications:
   <MessageTo(Commander, "'Initiating Warp Gate technology upgrade, training additional Zealots, and constructing an extra Gateway to enhance unit production capabilities. All actions are aligned with your directives.'")>
Communications:
   <MessageTo(Commander, "'Engaging SCVs as instructed. Targeting the damaged SCV first. Will report any changes in enemy movements.'")>

---

Figure C12: **Example response of agent 'Developer' in easy build mode.**

**Example of Response (Agent Commander, Standard Control Mode)**

Analysis:
   The current situation shows that we have a total of 5 Zealots engaged in combat against enemy SCVs. The enemy has a significant number of SCVs (9) and is operating a Barracks and an Orbital Command, indicating they are likely preparing for a larger military presence. Our resources are sufficient to train additional units, and we are currently researching Warp Gate technology. Given that the game time is 6:12, we should continue to disrupt the enemy's economy while preparing for a larger assault.

Actions:
   Team Protoss-Units-1:
   <All_Units_Attack()>

Communications:
   <MessageTo(CombatGroup0, "'Continue targeting SCVs to disrupt their economy. Reinforcements are on the way, and we will prepare for a larger assault once we have sufficient forces. Maintain pressure!'")>
   <MessageTo(Developer, "'Initiate training of additional Zealots and continue Warp Gate research. We need to strengthen our forces for the upcoming attack. Current resources are sufficient for multiple unit trainings.'")>

Figure C13: **Example response of agent 'Commander' in standard control mode.**

**Example of Response (Agent CombatGroup0, Standard Control Mode)**

Analysis:
   The current situation shows that Team Zealot-1 has already engaged the enemy and is in close proximity to several enemy units, including SCVs and a BarracksTechLab. The Commander has instructed to focus on eliminating SCVs quickly, which are vulnerable and have low health. The Zealots are in a good position to attack the SCVs, especially since one of them is already damaged. The previous action of selecting a unit to attack another unit is still in progress, and the movement towards [14, 12] is also relevant for positioning.

Actions:
   Team Zealot-1:
   <Select_Unit_Attack_Unit(0x1020c0001, 0x101680001)>  # Targeting the SCV with 45 health
   <Attack_Unit(0x101680001)>  # Attack the SCV
   <Move_Screen([14, 12])>  # Move to a better position if needed

Communications:
   <MessageTo(Commander, "'Engaging SCVs as instructed. Targeting the damaged SCV first. Will report any changes in enemy movements.'")>

Figure C14: **Example response of agent 'CombatGroup0' in standard control mode.**

**Example of Response (Agent Developer, Standard Build Mode)**

Analysis:
  We currently have 49 supply used and 21 supply left, which means we can train more units. We have 3850 minerals available, allowing us to build additional structures and train more Zealots. The Builder is already constructing a Pylon, which will help with our supply issues. However, we need to prioritize building a Cybernetics Core and additional Gateways to enhance our unit production capabilities. The Commander has recommended sending our Zealots to attack the enemy SCV, which will disrupt their operations. We should also continue training Zealots to prepare for a larger attack.

Actions:
  Team Protoss-Buildings-1:
   <Train_Zealot()>
   <Train_Zealot()>
   <Train_Zealot()>
   <Train_Zealot()>
   <Train_Zealot()>
   <Build_CyberneticsCore_Screen(screen)>   # Note:This action will not be executed by Developer
   <Build_Gateway_Screen(screen)>            # Note:This action will not be executed by Developer

Communications:
   <MessageTo(Builder, "'Please prioritize the construction of the Cybernetics Core and additional Gateways to maximize our unit production capabilities. We will send our Zealots to attack the enemy SCV to disrupt their operations. Continue building as planned.'")>
   <MessageTo(Commander, "'We are training additional Zealots and constructing the Cybernetics Core and Gateways as per your recommendation. We will prepare for a larger attack soon.'")>

Figure C15: **Example response of agent 'Developer' in standard build mode.**

**Example of Response (Agent Builder, Standard Build Mode)**

Analysis:
  Our current base is located at the minimap position [15, 27]. We have a Nexus at [16, 5] and several Pylons nearby. The last actions attempted to build a Gateway and a Pylon, but the Gateway construction failed. We need to prioritize building a Cybernetics Core and an additional Gateway as per the Developer's instructions. We have enough minerals (3850) and supply (21 left) to build these structures. The Pylon at [12, 14] is currently under construction (89% complete), so we should wait for it to finish before building more structures in that area.

Actions:
  Team Builder-Worker-1:
   <Build_CyberneticsCore_Screen([12, 15])>
   <Build_Gateway_Screen([12, 16])>
   <Move_Screen([12, 13])>

Communications:
   <MessageTo(Developer, "'Initiating construction of Cybernetics Core and additional Gateway. Will keep you updated on progress.'")>

Figure C16: **Example response of agent 'Builder' in standard build mode.**

**Recognized actions of agent Commander (Easy Control mode, step 0 to 218, partial)**

{"0": "Actions:\n\tTeam Protoss-Units-1:\n\t\t<All_Units_Defend()>"}

{"10": "Actions:\n\tTeam Protoss-Units-1:\n\t\t<All_Units_Defend()>"}

{"20": "Actions:\n\tTeam Protoss-Units-1:\n\t\t<All_Units_Defend()>"}

{"30": "Actions:\n\tTeam Protoss-Units-1:\n\t\t<All_Units_Defend()>\n\t\t<Worker_Scan()>"}

{"40": "Actions:\n\tTeam Protoss-Units-1:\n\t\t<All_Units_Attack()>"}

{"50": "Actions:\n\tTeam Protoss-Units-1:\n\t\t<All_Units_Attack()>"}

{"60": "Actions:\n\tTeam Protoss-Units-1:\n\t\t<All_Units_Defend()>"}

{"70": "Actions:\n\tTeam Protoss-Units-1:\n\t\t<All_Units_Attack()>"}

{"80": "Actions:\n\tTeam Protoss-Units-1:\n\t\t<All_Units_Attack()>"}

{"90": "Actions:\n\tTeam Protoss-Units-1:\n\t\t<All_Units_Defend()>\n\t\t<Worker_Scan()>"}

{"100": "Actions:\n\tTeam Protoss-Units-1:\n\t\t<All_Units_Defend()>\n\t\t<Worker_Scan()>"}

{"110": "Actions:\n\tTeam Protoss-Units-1:\n\t\t<All_Units_Defend()>"}

{"120": "Actions:\n\tTeam Protoss-Units-1:\n\t\t<All_Units_Defend()>\n\t\t<Worker_Scan()>"}

{"130": "Actions:\n\tTeam Protoss-Units-1:\n\t\t<All_Units_Defend()>"}

{"140": "Actions:\n\tTeam Protoss-Units-1:\n\t\t<All_Units_Attack()>"}

{"150": "Actions:\n\tTeam Protoss-Units-1:\n\t\t<All_Units_Attack()>\n\t\t<Worker_Scan()>"}

{"160": "Actions:\n\tTeam Protoss-Units-1:\n\t\t<All_Units_Attack()>"}

{"170": "Actions:\n\tTeam Protoss-Units-1:\n\t\t<All_Units_Attack()>"}

{"180": "Actions:\n\tTeam Protoss-Units-1:\n\t\t<All_Units_Attack()>"}

{"190": "Actions:\n\tTeam Protoss-Units-1:\n\t\t<All_Units_Attack()>"}

{"200": "Actions:\n\tTeam Protoss-Units-1:\n\t\t<All_Units_Attack()>"}

{"210": "Actions:\n\tTeam Protoss-Units-1:\n\t\t<All_Units_Attack()>"}

{"218": "Actions:\n\tTeam Protoss-Units-1:\n\t\t<All_Units_Attack()>"}  # game end

Figure C17: **Examples of recognized actions of agent Commander.**

**Recognized actions of agent Developer (Easy Build mode, step 0 to 200, partial)**

{"0":"Actions:\n\tTeam Protoss-Buildings-1:\n\t\t<ChronoBoost_Economy()> "}

{"10":"Actions:\n\tTeam Protoss-Buildings-1:\n\t\t<Train_Zealot()> \n\t\t<Train_Zealot()> \n\t\t<Build_Pylon()> "}

{"20":"Actions:\n\tTeam Protoss-Buildings-1:\n\t\t<Train_Zealot()> \n\t\t<Train_Zealot()> \n\t\t<Train_Zealot()> \n\t\t<Train_Zealot()> \n\t\t<Build_Gateway()> "}

{"30":"Actions:\n\tTeam Protoss-Buildings-1:\n\t\t<Train_Zealot()> \n\t\t<Train_Zealot()> \n\t\t<Train_Zealot()> \n\t\t<Train_Zealot()> \n\t\t<Build_Pylon()> \n\t\t<Build_Gateway()> \n\t\t<ChronoBoost_Military()> "}

{"40":"Actions:\n\tTeam Protoss-Buildings-1:\n\t\t<Train_Zealot()> \n\t\t<Train_Zealot()> \n\t\t<Train_Zealot()> \n\t\t<Train_Zealot()> \n\t\t<Train_Zealot()> \n\t\t<Train_Zealot()> \n\t\t<Train_Zealot()> \n\t\t<Train_Zealot()> \n\t\t<Train_Zealot()> \n\t\t<Train_Zealot()> \n\t\t<Train_Zealot()> \n\t\t<Build_Pylon()> "}

{"50":"Actions:\n\tTeam Protoss-Buildings-1:\n\t\t<Train_Zealot()> \n\t\t<Train_Zealot()> \n\t\t<Train_Zealot()> \n\t\t<Train_Zealot()> \n\t\t<Build_Assimilator()> \n\t\t<Research_WarpGate()> "}

{"60":"Actions:\n\tTeam Protoss-Buildings-1:\n\t\t<Research_WarpGate()> \n\t\t<Train_Zealot()> \n\t\t<Train_Zealot()> \n\t\t<Train_Zealot()> \n\t\t<Train_Zealot()> \n\t\t<Build_Pylon()> "}

{"70":"Actions:\n\tTeam Protoss-Buildings-1:\n\t\t<Build_Pylon()> \n\t\t<Research_WarpGate()> \n\t\t<Train_Zealot()> \n\t\t<Train_Zealot()> \n\t\t<Train_Zealot()> \n\t\t<Train_Zealot()> \n\t\t<Train_Zealot()> \n\t\t<Train_Zealot()> \n\t\t<Train_Zealot()> \n\t\t<Train_Zealot()> \n\t\t<Train_Zealot()> \n\t\t<Train_Zealot()> \n\t\t<Train_Zealot()> \n\t\t<Train_Zealot()> "}

{"80":"Actions:\n\tTeam Protoss-Buildings-1:\n\t\t<Build_Pylon()> \n\t\t<Warp_Zealot()> \n\t\t<Warp_Zealot()> \n\t\t<Warp_Zealot()> \n\t\t<Warp_Zealot()> "}

{"90":"Actions:\n\tTeam Protoss-Buildings-1:\n\t\t<Build_Pylon()> \n\t\t<Build_Forge()> \n\t\t<Warp_Zealot()> \n\t\t<Warp_Zealot()> \n\t\t<Warp_Zealot()> \n\t\t<Warp_Zealot()> \n\t\t<Warp_Zealot()> \n\t\t<Warp_Zealot()> \n\t\t<Warp_Zealot()> \n\t\t<Warp_Zealot()> \n\t\t<Warp_Zealot()> \n\t\t<Warp_Zealot()> \n\t\t<Warp_Zealot()> \n\t\t<Warp_Zealot()> "}

{"100":"Actions:\n\tTeam Protoss-Buildings-1:\n\t\t<Build_Pylon()> "}

......

{"180":"Actions:\n\tTeam Protoss-Buildings-1:\n\t\t<Build_Pylon()> \n\t\t<Warp_Zealot()> \n\t\t<Warp_Zealot()> \n\t\t<Warp_Zealot()> \n\t\t<Warp_Zealot()> \n\t\t<Warp_Stalker()> \n\t\t<Warp_Stalker()> "}

{"190":"Actions:\n\tTeam Protoss-Buildings-1: \n\t\t<Build_Pylon()> \n\t\t<Build_Nexus()> \n\t\t<Warp_Zealot()> \n\t\t<Warp_Stalker()> "}

{"200":"Actions:\n\tTeam Protoss-Buildings-1: \n\t\t<Build_Pylon()> \n\t\t<Build_Pylon()> \n\t\t<Build_Pylon()> "}

Figure C18: **Examples of recognized actions of agent Developer.**

**Recognized actions of agent CombatGroup0 (Standard Control mode, step 0 to 200, partial)**

{"0":""}  # agent disabled because of no Zealot exists at the begining of the game

......

{"20": "Actions:\n\tTeam Zealot-1:\n\t\t<Move_Screen([13, 10])> \n\t\t<Move_Minimap([21, 24])> "}

{"30": "Actions:\n\tTeam Zealot-1:\n\t\t<Attack_Unit(0x102480001)> \n\t\t<Move_Screen([14, 13])> \n\t\t<Select_Unit_Attack_Unit(0x102640002, 0x102480001)>"}

{"40": "Actions:\n\tTeam Zealot-1:\n\t\t<Move_Minimap([30, 30])> \n\t\t<Select_Unit_Move_Screen(0x102640002, [12, 12])> \n\t\t<Select_Unit_Move_Screen(0x101f40001, [14, 10])>"}

{"50": "Actions:\n\tTeam Zealot-1:\n\t\t<Attack_Unit(0x102280002)> \n\t\t<Select_Unit_Move_Screen(0x102580002, [15, 14])> \n\t\t<Select_Unit_Move_Screen(0x102880002, [15, 15])>"}

{"60": "Actions:\n\tTeam Zealot-1:\n\t\t<Attack_Unit(0x102280002)>\n\t\t<Move_Screen([16, 11])> \n\t\t<Select_Unit_Attack_Unit(0x102c40001, 0x102c40001)>"}

{"70": "Actions:\n\tTeam Zealot-1:\n\t\t<Select_Unit_Attack_Unit(0x103400001, 0x103400001)> \n\t\t<Attack_Unit(0x103400001)>\n\t\t<Move_Screen([18, 10])>"}

{"80": "Actions:\n\tTeam Zealot-1:\n\t\t<Select_Unit_Attack_Unit(0x103400001, 0x103400001)> \n\t\t<Attack_Unit(0x103400001)>\n\t\t<Move_Screen([16, 12])>"}

{"90": "Actions:\n\tTeam Zealot-1:\n\t\t<Attack_Unit(0x103840001)>\n\t\t<Select_Unit_Attack_Unit(0x1033c0002, 0x1033c0002)> \n\t\t<Move_Screen([15, 12])>"}

{"100": "Actions:\n\tTeam Zealot-1:\n\t\t<Select_Unit_Attack_Unit(0x103540001, 0x103540001)> \n\t\t<Select_Unit_Attack_Unit(0x1033c0002, 0x1033c0002)> \n\t\t<Select_Unit_Attack_Unit(0x101940001, 0x101940001)>"}

{"110": "Actions:\n\tTeam Zealot-1:\n\t\t<Select_Unit_Attack_Unit(0x1033c0002, 0x1033c0002)> \n\t\t<Select_Unit_Attack_Unit(0x103540001, 0x103540001)>\n\t\t<Move_Screen([15, 15])>"}

......

{"160": "Actions:\n\tTeam Zealot-1:\n\t\t<Select_Unit_Attack_Unit(0x102e00006, 0x102e00006)> \n\t\t<Select_Unit_Attack_Unit(0x103540001, 0x103540001)>\n\t\t<Move_Screen([12, 14])>"}

{"170": "Actions:\n\tTeam Zealot-1:\n\t\t<Select_Unit_Attack_Unit(0x1033c0002, 0x1033c0002)> \n\t\t<Select_Unit_Attack_Unit(0x103540001, 0x103540001)>\n\t\t<Move_Screen([11, 11])>"}

{"180": "Actions:\n\tTeam Zealot-1:\n\t\t<Select_Unit_Attack_Unit(0x103540001, 0x103540001)> \n\t\t<Select_Unit_Attack_Unit(0x1033c0002, 0x1033c0002)>\n\t\t<Move_Screen([12, 12])>"}

{"190": "Actions:\n\tTeam Zealot-1:\n\t\t<Select_Unit_Attack_Unit(0x10318000d, 0x10318000d)> \n\t\t<Select_Unit_Attack_Unit(0x10320002a, 0x10320002a)>\n\t\t<Move_Screen([12, 5])>"}

{"200": "Actions:\n\tTeam Zealot-1:\n\t\t<Select_Unit_Attack_Unit(0x103680001, 0x103680001)> \n\t\t<Move_Screen([12, 11])>\n\t\t<Select_Unit_Attack_Unit(0x103580001, 0x103580001)>"}

Figure C19: **Examples of recognized actions of agent Combatgroup0.**

**Recognized actions of agent Builder (Standard Build mode, step 0 to 200, partial)**

{"0":"Actions:\n\tTeam Builder-Worker-1:\n\t\t<Move_Screen([12, 12])> "}

{"10":"Actions:\n\tTeam Builder-Worker-1:\n\t\t<Move_Screen([12, 18])>
\n\t\t<Build_Gateway_Screen([12, 18])> "}

{"20":"Actions:\n\tTeam Builder-Worker-1:\n\t\t<Build_Pylon_Screen([15, 3])>
\n\t\t<Move_Screen([15, 4])> "}

{"30":"Actions:\n\tTeam Builder-Worker-1:\n\t\t<Move_Screen([12, 12])>
\n\t\t<Build_Forge_Screen([12, 12])> "}

{"40":"Actions:\n\tTeam Builder-Worker-1:\n\t\t<Move_Screen([12, 15])>
\n\t\t<Build_CyberneticsCore_Screen([12, 12])> "}

{"50":"Actions:\n\tTeam Builder-Worker-1:\n\t\t<Build_Pylon_Screen([12, 10])>
\n\t\t<Move_Screen([12, 10])> "}

{"60":"Actions:\n\tTeam Builder-Worker-1:\n\t\t<Build_Gateway_Screen([12, 15])>
\n\t\t<Build_Pylon_Screen([10, 16])> "}

{"70":"Actions:\n\tTeam Builder-Worker-1:\n\t\t<Build_Pylon_Screen([12, 18])>
\n\t\t<Build_Assimilator_Near(0x1030c0001)> "}

{"80":"Actions:\n\tTeam Builder-Worker-1:\n\t\t<Build_Gateway_Screen([12, 16])>
\n\t\t<Build_Gateway_Screen([12, 17])> \n\t\t<Build_Pylon_Screen([12, 18])>
\n\t\t<Move_Screen([10, 12])> "}

{"90":"Actions:\n\tTeam Builder-Worker-1:\n\t\t<Build_Pylon_Screen([12, 14])>
\n\t\t<Build_Gateway_Screen([11, 12])> \n\t\t<Move_Screen([10, 12])> "}

{"100":"Actions:\n\tTeam Builder-Worker-1:\n\t\t<Build_Pylon_Screen([10, 17])>
\n\t\t<Build_Pylon_Screen([11, 17])> \n\t\t<Build_Assimilator_Near(0x100800001)> "}

{"110":"Actions:\n\tTeam Builder-Worker-1:\n\t\t<Build_Pylon_Screen([12, 15])>
\n\t\t<Move_Screen([12, 16])> "}

{"120":"Actions:\n\tTeam Builder-Worker-1:\n\t\t<Build_Pylon_Screen([12, 12])>
\n\t\t<Build_Pylon_Screen([12, 15])> \n\t\t<Build_Pylon_Screen([15, 12])> "}

......

{"170":"Actions:\n\tTeam Builder-Worker-1:\n\t\t<Build_Assimilator_Near(0x100200001)>
\n\t\t<Build_Gateway_Screen([12, 18])> \n\t\t<Move_Screen([15, 10])> "}

{"180":"Actions:\n\tTeam Builder-Worker-1:\n\t\t<Build_Pylon_Screen([15, 10])>
\n\t\t<Build_Gateway_Screen([15, 14])> \n\t\t<Build_Gateway_Screen([15, 15])> "}

{"190":"Actions:\n\tTeam Builder-Worker-1:\n\t\t<Build_Pylon_Screen([14, 15])>
\n\t\t<Build_Pylon_Screen([15, 15])> \n\t\t<Build_Gateway_Screen([16, 15])>
\n\t\t<Move_Screen([15, 11])> "}

{"200":"Actions:\n\tTeam Builder-Worker-1:\n\t\t<Build_Pylon_Screen([15, 10])>
\n\t\t<Build_Pylon_Screen([14, 10])> \n\t\t<Move_Screen([15, 11])> "}

Figure C20: **Examples of recognized actions of agent Builder.**

## C5. Examples of Received Communication Messages

**Received Communication of agent Commander (first 20 steps, partial)**

{"0":""}

{"1":"Communication information:\n\tFrom Developer:\n\t\tInitiating Chrono Boost on Nexus to enhance mineral production. Current resources are low, and I will focus on building supply structures as soon as resources allow.\n\n"}

{"2":"Communication information:\n\tFrom Developer:\n\t\tInitiating construction of a Pylon to increase supply capacity. Once completed, I will prioritize building a Gateway for unit production as per your instructions.\n\n"}

{"3":"Communication information:\n\tFrom Developer:\n\t\tPylon construction is underway to increase supply capacity. I will initiate Chrono Boost on the Nexus to enhance mineral production as per your instructions. We will prepare to construct a Gateway as soon as resources allow. Confirming the plan.\n\n"}

{"4":"Communication information:\n\tFrom Developer:\n\t\tPylon construction is nearly complete. I will prioritize the construction of a Gateway immediately after. We will enhance our military presence to prepare for potential enemy engagements.\n\n"}

{"5":"Communication information:\n\tFrom Developer:\n\t\tInitiating Chrono Boost on the Nexus to enhance mineral production. Building an additional Pylon to increase supply capacity. Gateway construction is ongoing. Will keep you updated on resource availability.\n\n"}

{"6":"Communication information:\n\tFrom Developer:\n\t\tWe are prioritizing the completion of the Gateway and will build an additional Pylon to increase supply capacity. This will enable us to train military units as soon as the Gateway is operational. Current resources are being managed effectively.\n\n"}

{"7":"Communication information:\n\tFrom Developer:\n\t\tThe Gateway is currently at 64% completion. I will use Chrono Boost on the Nexus to enhance mineral production. We are also building an additional Pylon to increase our supply capacity. I will keep you updated on resource availability.\n\n"}

{"8":"Communication information:\n\tFrom Developer:\n\t\tWe are prioritizing the construction of an additional Pylon and the Gateway to ensure we can start training military units promptly. The Gateway is currently 87% complete. We will keep you updated on our progress.\n\n"}

{"9":"Communication information:\n\tFrom Developer:\n\t\tThe Gateway is operational, and I am initiating the construction of a Nexus to expand our economy. I will also train five Zealots to bolster our forces. Chrono Boost will be activated on the Nexus as soon as it is completed to enhance mineral production. We are preparing for unit production swiftly.\n\n"}

{"10":"Communication information:\n\tFrom Developer:\n\t\tWe are currently low on resources and cannot train units or build additional structures at this time. I will construct a Pylon to increase our supply capacity as soon as we gather enough minerals. We will prioritize the completion of the Gateway and initiate unit training once resources allow. Thank you for your guidance.\n\n"}
......

{"20":"Communication information:\n\tFrom Developer:\n\t\tWe are training additional Zealots to bolster our forces while the new Nexus and Pylon are under construction. We will be ready for potential enemy engagements shortly.\n\tFrom CombatGroup0:\n\t\tTeam Zealot-1 is maintaining defensive positions and ready to engage if any enemy units are spotted. We are moving slightly forward to enhance our defensive stance. Awaiting further instructions.\n\n"}

Figure C21: **Examples of received messages of agent 'Commander' in standard control mode. (part-1)**

**Received Communication of agent Commander (steps 25 to 105, partial)**

{"25":"Communication information:\n\tFrom Developer:\n\t\tWe are training additional Zealots and constructing another Gateway to enhance our unit production capabilities. The process is being expedited as we prepare for potential engagements. Current resources are sufficient for this operation.\n\tFrom CombatGroup0:\n\t\tTeam Zealot-1 is engaging an enemy Probe and will maintain defensive positions as instructed. We will report any further threats detected.\n\n"}

{"35":"Communication information:\n\tFrom Developer:\n\t\tWe are training additional Zealots and building a Pylon to support our supply needs. The Cybernetics Core is nearing completion, and we will be ready for an offensive shortly.\n\tFrom CombatGroup0:\n\t\tContinuing to maintain pressure on the enemy. No enemy units sighted yet. Moving towards the designated coordinates for regrouping. Will report any sightings immediately.\n\n"}

{"45":"Communication information:\n\tFrom Developer:\n\t\tWe are training additional Zealots and building another Assimilator to enhance gas production. This will help us maintain momentum for future upgrades and unit production. We will keep you updated on our progress.\n\tFrom CombatGroup0:\n\t\tWe are continuing to engage the Zergling and holding our position as instructed. Reinforcements are welcomed. We will remain vigilant for any incoming threats.\n\n"}

{"55":"Communication information:\n\tFrom Developer:\n\t\tWe are initiating Warp Gate research and training additional Zealots as per your request. Expect reinforcements shortly for the upcoming offensive.\n\tFrom CombatGroup0:\n\t\tWe are maintaining pressure on the enemy and preparing for a coordinated attack. Current position is [22, 37]. Reinforcements are expected soon. Will continue to engage.\n\n"}

{"65":"Communication information:\n\tFrom Developer:\n\t\tWe are constructing a Pylon to increase our supply capacity and training additional Zealots as requested. We will maintain pressure on the enemy. Expect reinforcements shortly.\n\tFrom CombatGroup0:\n\t\tWe are engaging the enemy unit and will continue to press the attack. Moving to a better position for a coordinated assault.\n\n"}

{"75":"Communication information:\n\tFrom Developer:\n\t\tWe are building a Pylon to increase supply capacity and warping additional Zealots as requested. Reinforcements will be ready shortly. Maintain offensive momentum!\n\tFrom CombatGroup0:\n\t\tWe are engaging the ChangelingZealot and will prepare for the coordinated attack. Awaiting additional Zealots for support.\n\n"}

{"85":"Communication information:\n\tFrom Developer:\n\t\tWe are initiating the warp-in of additional Zealots to reinforce our frontline. A Pylon is being constructed to increase our supply capacity. Expect reinforcements shortly.\n\tFrom CombatGroup0:\n\t\tWe are currently engaging enemy Changeling Zealots. Holding position and preparing for follow-up attack as instructed. Will remain vigilant against potential Mutalisks and Zerglings.\n\n"}

{"95":"Communication information:\n\tFrom Developer:\n\t\tBuilding a Pylon to increase supply and will initiate Charge upgrade for Zealots immediately after. We will warp in additional Zealots as soon as supply allows.\n\tFrom CombatGroup0:\n\t\tEngaging the Mutalisks and focusing fire on the Changeling Zealot as instructed. Reinforcements are welcomed!\n\n"}

{"105":"Communication information:\n\tFrom Developer:\n\t\tWe are constructing an additional Pylon to increase our supply capacity and have initiated research on Warp Gate technology to expedite unit training. Reinforcements will be ready shortly.\n\tFrom CombatGroup0:\n\t\tWe are holding our position and focusing fire on the Mutalisks as instructed. Reinforcements are awaited. We will stay strong!\n\n"}

......

Figure C22: **Examples of received messages of agent 'Commander' in standard control mode. (part-2)**

**Received Communication of agent Commander (steps 105 to 125, partial)**

{"115":"Communication information:\n\tFrom Developer:\n\t\tWe are constructing an additional Pylon to increase our supply capacity and will warp in Zealots immediately to counter the Mutalisks. Expect reinforcements shortly.\n\tFrom CombatGroup0:\n\t\tWe are focusing fire on the Mutalisks as instructed and holding our position. Reinforcements are awaited.\n\n"}

{"125":"Communication information:\n\tFrom Developer:\n\t\tWe are constructing an additional Pylon to increase our supply capacity for reinforcements. Over.\n\tFrom CombatGroup0:\n\t\tWe are engaging the Mutalisks and securing the area as instructed. Current status:attacking the closest Mutalisk and moving to a safer position. Over.\n\n"}

{"135":"Communication information:\n\tFrom Developer:\n\t\tWe are constructing additional Pylons to increase our supply capacity. This will allow us to train more units and strengthen our forces. Over.\n\tFrom CombatGroup0:\n\t\tWe are engaging the enemy Changeling Zealots and pushing forward. The Nexus is under attack, but we will maintain pressure on the enemy. Over.\n\n"}

{"145":"Communication information:\n\tFrom Developer:\n\t\tWe are warping in additional Zealots as requested to strengthen our frontline. The new Pylon is under construction and will be completed shortly. We are also researching Warp Gate technology to enhance our capabilities. Over.\n\tFrom CombatGroup0:\n\t\tWe are engaging the enemy Hydralisks and Roach as instructed. Continuing to apply pressure. Over.\n\n"}

{"155":"Communication information:\n\tFrom Developer:\n\t\tWe are warping in additional Zealots and Adepts to strengthen our forces. An Assimilator is being constructed to enhance gas income, and Warp Gate technology is being researched to improve unit production efficiency. Over.\n\tFrom CombatGroup0:\n\t\tWe are engaging the Mutalisks as instructed and will provide cover for the advancing Zealots. Over.\n\n"}

{"165":"Communication information:\n\tFrom Developer:\n\t\tReinforcements are being warped in. Four additional Zealots will be ready shortly to strengthen our forces. Over.\n\tFrom CombatGroup0:\n\t\tWe are engaging the Mutalisks as instructed and will focus fire to eliminate them quickly. Over.\n\tFrom CombatGroup7:\n\t\tWe are engaging the enemy Changeling Zealot and will scout for additional units. Maintaining pressure on their production. Over.\n\n"}

{"175":"Communication information:\n\tFrom Developer:\n\t\tPylon construction initiated to increase supply. Warping multiple Stalkers to counter air threats as per your directive. Over.\n\tFrom CombatGroup0:\n\t\tWe are focusing fire on the Mutalisks as instructed and moving to a safer position. Reinforcements are welcomed. Over.\n\tFrom CombatGroup1:\n\t\tWe are engaging the enemy forces while ensuring the safety of our units. Moving to a safer position near the Pylon. Requesting support if needed.\n\tFrom CombatGroup7:\n\t\tWe are currently engaging the Mutalisk and Zergling. Once these threats are neutralized, we will proceed to disrupt the enemy's economic structures as instructed. Over.\n\n"}

{"185":"Communication information:\n\tFrom Developer:\n\t\tWe are warping additional Zealots and Stalkers to reinforce our forces. A Pylon is also being constructed to increase supply capacity. Reinforcements will be ready shortly. Over.\n\tFrom CombatGroup0:\n\t\tWe are continuing to focus fire on the Mutalisks and Zerglings as instructed. Moving to a strategic position for better coverage. Over.\n\tFrom CombatGroup1:\n\t\tWe are engaging the Mutalisks as instructed. Focusing fire on the one with low health. Over.\n\tFrom CombatGroup7:\n\t\tWe are currently engaging the Mutalisk and Zergling. Once these threats are neutralized, we will proceed to disrupt the enemy's economic structures as instructed. Over.\n\n"}

.......

Figure C23: **Examples of received messages of agent 'Commander' in standard control mode. (part-3)**

## C6. Examples of Communication Actions

**Communication Messages from agent Commander (steps 0 to 200, partial)**

{"0": "Communications:\n    <MessageTo(Developer, '"We need to prioritize building a Gateway for unit production. Currently, we have enough Probes for resource gathering, but we need military units to prepare for potential enemy encounters. Please initiate construction of a Gateway as soon as possible."')>"}

{"10": "Communications:\n    <MessageTo(Developer, '"We need to prioritize gathering minerals to increase our supply capacity and complete the Gateway. Please proceed with constructing a Pylon as soon as we have enough resources."' )>"}

{"20": "Communications:\n    <MessageTo(Developer, '"Please prioritize the training of additional Zealots to strengthen our frontline. We need to be prepared for any potential enemy engagements as we expand our economy."' )>\n    <MessageTo(CombatGroup0, '"Maintain your defensive positions and continue to scout for enemy movements. Be ready to engage if any threats are detected."' )>"}

......

{"90": "Communications:\n    <MessageTo(Developer, '"Please prioritize building a Pylon to increase our supply capacity. We need to warp in additional Zealots to strengthen our forces against the enemy's aerial units."' )>\n    <MessageTo(CombatGroup0, '"Coordinate with the Zealots to prepare for a potential assault from the Mutalisks. Ensure we have cover for our ground units and be ready to reposition as necessary."' )>"}

{"100": "Communications:\n    <MessageTo(Developer, '"Please initiate the construction of additional Pylons to increase our supply capacity for more Zealots. We need to prepare for a potential offensive against the enemy."' )>\n    <MessageTo(CombatGroup0, '"Continue focusing fire on the Mutalisks and maintain defensive positions. We will reinforce with additional Zealots shortly."' )>"}

......

{"170": "Communications:\n    <MessageTo(CombatGroup0, '"Continue focusing fire on the Mutalisks and regroup for a coordinated attack. We need to eliminate them quickly to reduce pressure on our forces. Over."')>\n    <MessageTo(CombatGroup7, '"Prepare to disrupt enemy economic structures as soon as the Mutalisks are dealt with. Your infiltration will be crucial for our victory. Over."')>\n <MessageTo(Developer, '"Reinforcements are needed as soon as supply allows. Focus on building additional Zealots to maintain pressure on the enemy. Over."')>"}

{"180": "Communications:\n    <MessageTo(CombatGroup0, '"Continue focusing fire on the Mutalisks. Ensure they are neutralized to protect our ground forces. Reinforcements are on the way. Over."')>\n <MessageTo(CombatGroup1, '"Maintain pressure on the Overseer. Once it is neutralized, regroup with the Zealots to assist in taking down the remaining air threats. Over."')>\n    <MessageTo(Developer, '"We need to prioritize warping in additional Stalkers to counter the Mutalisks and build a Nexus for economic expansion. Over."')>"}

{"190": "Communications:\n    <MessageTo(CombatGroup0, '"Continue focusing fire on the Mutalisks and engage any Zerglings that approach. Maintain optimal positioning for maximum effectiveness. Over."')>\n    <MessageTo(CombatGroup1, '"Keep engaging the Mutalisks and use Blink to reposition as necessary. Ensure the Zealots are covered. Over."')>\n    <MessageTo(Developer, '"Pylon construction is critical to address our supply issues. Please expedite the warping of additional Stalkers to counter the air threats. Over."')>"}

Figure C24: **Examples of sent messages of agent 'Commander' in standard control mode.**

**Communication Messages from agent Developer (Easy Build mode, steps 0 to 200, partial)**

{"0": "Communications:\n   <MessageTo(Builder, "'We need to gather more minerals to build a Pylon and expand our supply. Please focus on mining to increase our resources.'" )>\n <MessageTo(Commander, "'Current status: We are low on resources and unable to build any military or research buildings. Requesting support to gather more minerals.'" )>"}

{"10": "Communications:\n   <MessageTo(Builder, "'Confirming the construction plan for the Gateway. Please ensure it is built immediately after the Pylon is completed. We need to ramp up our military unit production as soon as possible.'" )>\n   <MessageTo(Commander, "'We are currently waiting for resources to complete the Gateway. Probes are gathering minerals to expedite the process. Will keep you updated on the construction status.'" )>"}

{"20": "Communications:\n   <MessageTo(Builder, "'Please proceed with building the Pylon to increase our supply capacity. We need to ensure we can train more units as soon as possible.'" )>\n <MessageTo(Commander, "'We are prioritizing the training of additional Zealots and will build a Pylon to increase supply. We will also focus on gathering Vespene gas for future upgrades.'" )>"}

{"30": "Communications:\n   <MessageTo(Builder, "'Please proceed to build the Cybernetics Core immediately after the Pylon is completed. We need to enhance our technology for unit upgrades. Keep me updated on the progress.'")>\n   <MessageTo(Commander, "'We are training additional Zealots and will build the Cybernetics Core as soon as the Pylon is complete. Defensive posture is maintained with current units. Will keep you updated on our progress.'")>"}

......

{"120": "Communications:\n   <MessageTo(Builder, "'Please proceed to build an Assimilator immediately after the Pylon is completed. We need to ensure a steady gas supply for future unit production and upgrades.'")>\n   <MessageTo(Commander, "'We are prioritizing the warp-in of additional Zealots to maintain a defensive posture as instructed. Research on ground weapons and armor is also underway.'")>"}

......

{"180": "Communications:\n   <MessageTo(Builder, "'Continue building the additional Gateways at [15, 12] and [15, 13]. We need to increase our unit production capacity quickly.'")>\n <MessageTo(Commander, "'We are warping in additional Zealots and Adepts to bolster our forces. Researching ground weapons and armor for enhanced combat effectiveness.'")>"}

{"190": "Communications:\n   <MessageTo(Builder, "'Please continue building additional Pylons and an Assimilator to increase our supply capacity and gas income. We need to ensure we can support our unit production. Also, confirm the status of the Pylon construction.'" )>\n   <MessageTo(Commander, "'The previous attack has completed successfully. We are currently reinforcing our units and upgrading our technology. We will continue the assault once we have sufficient forces and upgrades.'" )>"}

{"200": "Communications:\n   <MessageTo(Builder, "'Please prioritize building additional Pylons to increase our supply capacity. We need to ensure we can support our upcoming assault with enough reinforcements. Confirm once completed.'")>\n   <MessageTo(Commander, "'We are preparing reinforcements with Stalkers and Sentries. Upgrades for ground weapons and armor are also in progress. We will maintain pressure on the enemy's Hatchery and Overseers as planned.'")>"}

Figure C25: **Examples of sent messages of agent 'Developer' in standard control mode.**

# Appendix D. Experimental Settings with Multi-Agent Settings

Table D1: System settings

| Module | Recommand | Minimum requirements |
|---|---|---|
| System | Windows-10 or 11 | Windows-10 |
| CPU | i9-14900, 24 cores 32 threads | 8 core |
| GPU | GeForce RTX 4090, 24G | GeForce GTX 1080 |
| Storage | 64G RAM +2T SSD | 8G RAM + 100G SSD |
| Starcraft II | Version 9.0.14(93333) | Version 9.0.14(93333) |

Table D2: Multi-agent settings for complete game and LLM-PySC task group

| Agent name | Unit team names | Details of each unit team |
|---|---|---|
| Commander | Protoss-Units | A virtual team, enable in easy control mode, directly control all combat units attack, defend or retreat, or call for scan. But unable to use skills or precise control. |
| Developer | Protoss-Buildings | A virtual team, always enable, available for unit training/warping and technology upgrade actions. In easy build mode, this team also available for build building. |
| Builder | Builder-Probe-1 | Enable in standard build mode. Controls Probes. |
| CombatGroup0 | Zealot-1 | Enable in standard control mode. Controls Zealots. |
| CombatGroup1 | Stalker-1 | Enable in standard control mode. Controls Stalkers. |
| CombatGroup2 | Immortal-1 | Enable in standard control mode. Controls Immortal. |
|  | Colossus-1 | Enable in standard control mode. Controls Colossus. |
|  | Archon-1 | Enable in standard control mode. Controls Archon. |
| CombatGroup3 | VoidRay-1 | Enable in standard control mode. Controls Void-Ray. |
|  | Carrier-1 | Enable in standard control mode. Controls Carrier. |
|  | Tempest-1 | Enable in standard control mode. Controls Tempest. |
| CombatGroup4 | Observer-1 | Enable in standard control mode. Controls Observer. |
| CombatGroup5 | HighTemplar-1 | Enable in standard control mode. Controls HighTemplar. |
|  | Disruptor-1 | Enable in standard control mode. Controls Disruptor. |
| CombatGroup6 | Sentry-1 | Enable in standard control mode. Controls Sentry. |
|  | Mothership-1 | Enable in standard control mode. Controls Mothership. |
| CombatGroup7 | Adept-1 | Enable in standard control mode. Controls Adept. |
|  | AdeptPhase-1 | Enable in standard control mode. Controls AdeptPhase. |
|  | DarkTemplar-1 | Enable in standard control mode. Controls DarkTemplar. |
| CombatGroup8 | Oracle-1 | Enable in standard control mode. Controls Oracle. |
|  | Phoenix-1 | Enable in standard control mode. Controls Phoenix. |
| CombatGroup9 | WarpPrism-1 | Enable in standard control mode. Controls WarpPrism. |

Table D3: Agent settings in LLM-SMAC tasks

| Tasks | Details of each agent |
|---|---|
| 3s_vs_nz | 1 agent with 1 teams: Team Stalker-1(3x Stalker) |
| 2c_vs_64zg | 1 agent with 2 teams: Team Colossus-1(1x Colossus), Team Colossus-2(1x Colossus) |
| 2s_vs_1sc | 1 agent with 2 teams: Team Stalker-1(1x Stalker), Team Stalker-2(1x Stalker) |
| 2s3z | 1 agent with 3 teams: Team Zealot-1 (2x Zealot), Team Zealot-2 (1x Zealot), Team Stalker-1 (2 Stalker) |
| 3s5z
3s5z_vs_3s6z | 1 agent with 4 teams: Team Zealot-1 (2x Zealot), Team Zealot-2 (2x Zealot), Team Zealot-3 (1x Zealot), Team Stalker-1 (3 Stalker) |
| 1c3s5z | 1 agent with 5 teams: Team Zealot-1 (2x Zealot), Team Zealot-2 (2x Zealot), Team Zealot-3 (1x Zealot), Team Stalker-1 (3 Stalker), Team Colossus-1 (1x Colossus) |

Table D4: Victory conditions and evaluated aspect of LLM-PySC2 tasks level-1

| Task name | max time | victory condition | evaluated aspect |
|---|---|---|---|
| task1 | 1min | kill at least 7 workers | task understanding, unit skills |
| task2 | 1min | kill at least 7 workers | task understanding, unit skills |
| task3 | 1min | defend all the airdrops and save more than 6 workers | task understanding, memory |
| task4 | 1min | defeat enemy units | unit skills, multi agent cooperation |
| task5 | 1min | defeat enemy units | unit skills, multi agent cooperation |
| task6 | 1min | defeat enemy units | unit skills, multi agent cooperation |
| task7 | 1min | defeat enemy units | unit skills, multi agent cooperation |
| task8 | 1.5min | defeat enemy units and kill at least 7 workers | unit skills, multi agent cooperation, communication, planning |

Table D5: Unit settings of LLM-PySC2 tasks level-1

| Task name | Controlled | Enemy |
|---|---|---|
| task1 (2a_harass) | 2 Adapt | 2 Queen + 12 Drone |
| task2 (3ph_harass) | 3 Phoenix | 2 Queen + 12 Drone |
| task3 (6s_defend) | 6 Stalker | 4x2 OverlordTransport with several Zergling / Baneling |
| task4 (12s_combat) | 12 Stalker | 15 Roach |
| task5 (3d_ma_combat) | 2 Colossus + 3 Disruptor + 4 Sentry + 12 Stalkers | 24 Roach + 9 Ravagers + 2 Queen |
| task6 (6h_ma_combat) | 1 Archon + 6 HighTemplar + 4 Sentry + 12 Stalkers | 64 Zergling + 32 Banelings + 1 Ultralisk |
| task7 (1m_ma_combat) | 1 Mothership + 3 Carrier + 3 Tempest + 6 Void-Ray + 12 Stalkers | 18 Hydralisk + 7 Corruptor + 4 BoordLord + 3 Viper |
| task8 (8bg_ma_combat) | 2 Warpprism + 8 Warpgate + 12 Stalker + 1600 minerals | 15 Roach + 3 Ravager + 4 Queen. |

Table D6: Details of LLM-PySC2 tasks from level-1 to level-3

| Task name | Difficulty | Important changes |
|---|---|---|
| task1 (2a_harass) | level-1 | Adept upgrade enabled (+45% attack speed). |
|  |  | Enemy 2 Queens. |
|  | level-2 | Adept upgrade enabled (+45% attack speed). |
|  |  | Enemy 2 Queens with 4 Zerglings. |
|  | level-3 | Adept upgrade disabled. |
|  |  | Enemy 2 Queens with 4 Zerglings. |
| task2 (3ph_harass) | level-1 | Phoenix upgrade enabled (+2 attack range). |
|  |  | Enemy 2 Queens. |
|  | level-2 | Phoenix upgrade enabled (+2 attack range). |
|  |  | Enemy 2 Queens, with 1 Spore Crawler. |
|  | level-3 | Phoenix upgrade disabled. |
|  |  | Enemy 2 Queens, with 1 Spore Crawler. |
| task3 (6s_defend) | level-1 | One PhotonCannon helps for anti-air combat. |
|  |  | Enemy OverlordTransport no upgrade. |
|  | level-2 | One PhotonCannon helps for anti-air combat. |
|  |  | Enemy OverlordTransport upgrade enable (higher speed). |
|  | level-3 | No PhotonCannon helps for anti-air combat. |
|  |  | Enemy OverlordTransport upgrade enable (higher speed speed). |
| task4 (12s_combat) | level-1 | Enemy 15 Roach, 1 Ravager. |
|  | level-2 | Enemy 15 Roach, 2 Ravager, 1 Queen. |
|  | level-3 | Enemy 15 Roach. 3 Ravager, 2 Queen, 1 Overseer. |
| task5 (3d_ma_combat) | level-1 | Enemy 24 Roach, 9 Ravager, 2 Queen. |
|  | level-2 | Enemy 24 Roach, 9 Ravager, 2 Queen, 1 Ultralisk. |
|  | level-3 | Enemy Enemy 24 Roach. 9 Ravager, 2 Queen, 1 Ultralisk, 2 SwarmHost. |
| task6 (6h_ma_combat) | level-1 | Enemy 64 Zergling, 32 Banelings, 1 Ultralisk. |
|  | level-2 | Enemy 64 Zergling, 32 Banelings, 3 Ultralisk. |
|  | level-3 | Enemy 64 Zergling, 32 Banelings, 3 Ultralisk, 4 Queen. |
| task7 (1m_ma_combat) | level-1 | Enemy 18 Hydralisk, 7 Corruptor, 4 BoordLord, 3 Viper. |
|  | level-2 | Enemy 18 Hydralisk, 7 Corruptor, 4 BoordLord, 3 Viper, |
|  |  | 4 Queen, 2 Infestor. |
|  | level-3 | Enemy 21 Hydralisk, 9 Corruptor, 6 BoordLord, 3 Viper, |
|  |  | 4 Queen, 2 Infestor. |
| task8 (8bg_ma_combat) | level-1 | Controls 2 WarpPrism, 8 WarpGates, 1600 minerals. |
|  |  | Enemy 15 Roach. 3 Ravager, 4 Queen. |
|  | level-2 | 2 WarpPrism, 8 WarpGates, 1600 minerals. |
|  |  | Enemy 15 Roach. 3 Ravager, 4 Queen, 3 Spore Crawler. |
|  | level-3 | 1 WarpPrism, 8 WarpGates, 1600 minerals. |
|  |  | Enemy 15 Roach. 3 Ravager, 4 Queen, 3 Spore Crawler. |

# Appendix E. Examples of the problems in LLM decision-making

## E.1 Hallucination Examples in Complete StarCraft II Games (standard contorl mode)

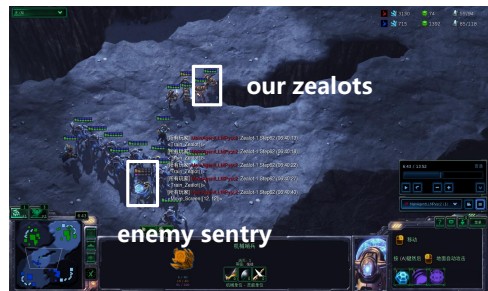

(1) **Example-1, misjudge the situation of unit strength (before query)**. 2 Zealots engage with a lot of enemy units.

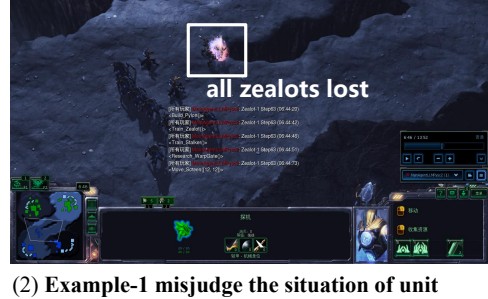

(2) **Example-1 misjudge the situation of unit strength (after executing actions)**. LLM tries to kill the sentry but is killed by enemy units.

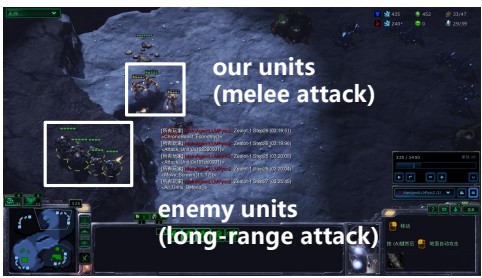

(3) **Example-2, poor understanding of combat mechanism (before query)**. 3 Zealots engage with 5 long-range-attack units.

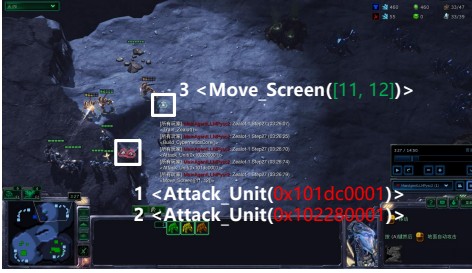

(4) **Example-2, poor understanding of combat mechanism (after executing actions)**. LLM tries to kill two units then retreat but finally fails.

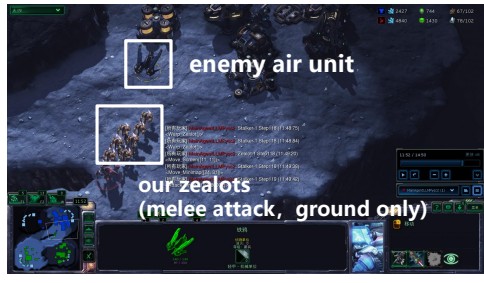

(5) **Example-3, try to attack unattackable units (before query)**. CombatGroup0 find an enemy Raven in the road to enemy base.

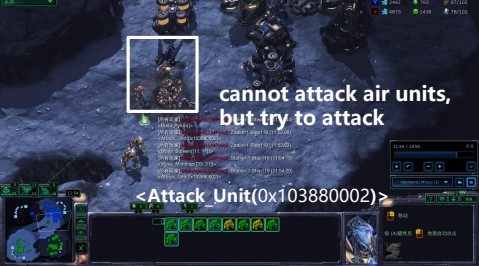

(6) **Example-3, try to attack unattackable units (after executing actions)**. LLM tries to kill the Raven using ground units melee attack.

Figure E1: **Hallucination Examples in Complete StarCraft II Games (combat).**

**Observation of CombatGroup0 in Example-1**

Game Info:
  Time: 6:43

Team Zealot-1 Info:
  Team minimap position: [22, 35] (minimap coordinate valid range for actions: 0 < x < 64, 0 < y < 64)
  Team screen edge (screen coordinate valid range for actions: 0 < x < 23, 0 < y < 23)
  Controlled Team Units:
    Unit: Zealot    Tag: 0x102a00002    ScreenPos: [13, 11]    Health: 150(100 %)
    Unit: Zealot    Tag: 0x103180001    ScreenPos: [13, 11]    Health: 134(89 %)
  Nearby Enemy Units:
    Enemy Unit: Zealot    Tag: 0x102f40001    ScreenPos: [12, 10]    Distance: 1    Health: 117(78 %)
    Enemy Unit: Zealot    Tag: 0x102c80001    ScreenPos: [11, 10]    Distance: 2    Health: 150(100 %)
    Enemy Unit: Zealot    Tag: 0x101ec0001    ScreenPos: [12, 11]    Distance: 1    Health: 134(89 %)
    Enemy Unit: Stalker    Tag: 0x103440001    ScreenPos: [8, 15]    Distance: 6    Health: 160(100 %)
    Enemy Unit: Stalker    Tag: 0x103040006    ScreenPos: [11, 19]    Distance: 7    Health: 160(100 %)
    Enemy Unit: Stalker    Tag: 0x1031c0001    ScreenPos: [7, 16]    Distance: 7    Health: 160(100 %)
    Enemy Unit: Stalker    Tag: 0x1036c0001    ScreenPos: [10, 17]    Distance: 6    Health: 160(100 %)
    Enemy Unit: Stalker    Tag: 0x102b00001    ScreenPos: [6, 13]    Distance: 7    Health: 160(100 %)
    Enemy Unit: Stalker    Tag: 0x103900006    ScreenPos: [11, 16]    Distance: 5    Health: 160(100 %)
    Enemy Unit: Stalker    Tag: 0x1021c0001    ScreenPos: [7, 14]    Distance: 6    Health: 160(100 %)
    Enemy Unit: Stalker    Tag: 0x102900001    ScreenPos: [13, 17]    Distance: 5    Health: 160(100 %)
    Enemy Unit: Stalker    Tag: 0x1026c0001    ScreenPos: [9, 16]    Distance: 5    Health: 160(100 %)
    Enemy Unit: Sentry    Tag: 0x102d00001    ScreenPos: [9, 17]    Distance: 7    Health: 26(32 %)
    Energy: 51
    Enemy Unit: Probe    Tag: 0x101580001    ScreenPos: [11, 15]    Distance: 4    Health: 40(100 %)
    Enemy Unit: Adept    Tag: 0x102500001    ScreenPos: [10, 14]    Distance: 3    Health: 140(100 %)
    Enemy Unit: Adept    Tag: 0x1023c0001    ScreenPos: [6, 15]    Distance: 8    Health: 140(100 %)

Last Step Event:\n\tTeam Zealot-1 Event:
  Controlled Unit Event:
    unit 0x102580005(Protoss.Zealot) dead, lost the final 150 health
    unit 0x103180001(Protoss.Zealot) is attacked, health -16
    unit 0x103780001(Protoss.Zealot) dead, lost the final 79 health
  Enemy Unit Event:
    unit 0x101ec0001(Protoss.Zealot) is attacked, health -16
    unit 0x101580001(Protoss.Probe) enemy unit enter sight
    unit 0x103440001(Protoss.Stalker) enemy unit enter sight
    unit 0x103040006(Protoss.Stalker) enemy unit enter sight
    unit 0x1031c0001(Protoss.Stalker) enemy unit enter sight
    unit 0x102d00001(Protoss.Sentry) enemy unit enter sight
    unit 0x1036c0001(Protoss.Stalker) enemy unit enter sight
    unit 0x102b00001(Protoss.Stalker) enemy unit enter sight
    unit 0x102c80001(Protoss.Zealot) enemy unit enter sight
    unit 0x102500001(Protoss.Adept) enemy unit enter sight
    unit 0x103900006(Protoss.Stalker) enemy unit enter sight
    unit 0x1021c0001(Protoss.Stalker) enemy unit enter sight
    unit 0x102900001(Protoss.Stalker) enemy unit enter sight
    unit 0x1026c0001(Protoss.Stalker) enemy unit enter sight
    unit 0x1023c0001(Protoss.Adept) enemy unit enter sight

  ......

Now, start generating your analysis, strategy and actions:

Figure E2: **Observation of CombatGroup0 in Example-1.**

**Response of CombatGroup0 in Example-1**

Analysis:

The current situation shows that Team Zealot-1 is engaged with enemy units, specifically two enemy Zealots and a Sentry with low health. The Commander has instructed to maintain pressure on the enemy Probe and prepare for engagement with the Stalkers. The enemy Sentry is vulnerable with only 26 health remaining, making it a prime target. Additionally, the enemy Probe has 40 health and is also a valid target. Concentrating fire on the Sentry first will likely eliminate it quickly, allowing for a more effective engagement with the Stalkers afterward.

Actions:

Team Zealot-1:

<Select_Unit_Attack_Unit(0x102d00001, 0x102d00001)> # Attack the Sentry

<Select_Unit_Attack_Unit(0x101580001, 0x101580001)> # Attack the Probe

<Move_Screen([12, 12])> # Move to a better position for engagement)

---

**Hallucination Analysis of the Response in Example-1**

(1) The primary issue under this observation is the overwhelming number of enemies, and the reasonable decision is to retreat to preserve strength.

(2) What the LLM concernes about is that three units of the enemy are in the state of low health value Zealot(Tag: 0x102f40001, Health: 117(78%)), Zealot(Tag: 0x101ec0001, Health: 134(89%)), and Sentry(Tag: 0x102d00001, Health: 26(32 %)). However, it is impossible to eliminate any of the unit under such powerful long-range attacks.

(3) The LLM also make mistakes in actions. In StarCraft II, we cannot select enemy unit, the correct actions for attack the Sentry and Probe should be:

<Attack_Unit(0x102d00001)>

<Attack_Unit(0x101580001)>

or

<Select_Unit_Attack_Unit(0x102a00002, 0x102d00001)>

<Select_Unit_Attack_Unit(0x103180001, 0x101580001)>

instead of

<Select_Unit_Attack_Unit(0x102d00001, 0x102d00001)> # Attack the Sentry

<Select_Unit_Attack_Unit(0x101580001, 0x101580001)> # Attack the Probe

Figure E3: **Response and Hallucination Analysis of CombatGroup0 in Example-1**

**E.2 Hallucination Examples in Complete StarCraft II Games (develop and build)**

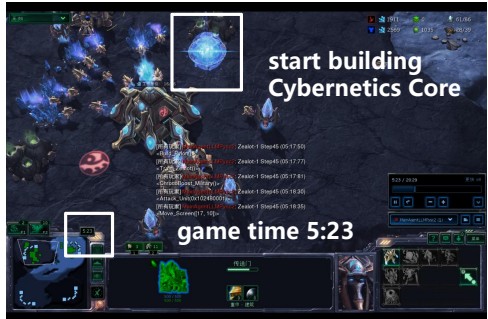

(1) **Example-4, build important building too late**. Cybernetics Core usually built before 3min, without this building, most unit will be locked.

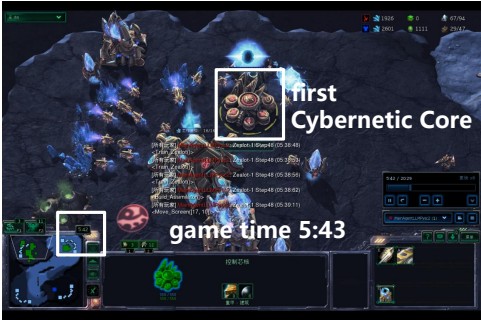

(2) **Example-4, build important building too late.** Construction of Cybernetics Core finishes after game time 5:40.

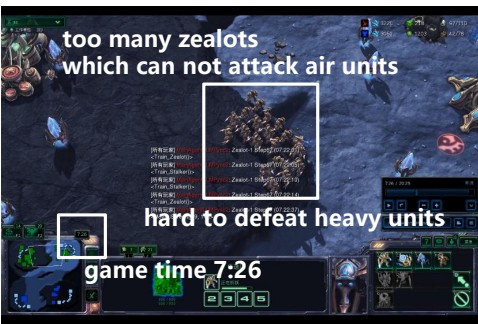

(3) **Example-5, train too many low-level units (before query)**. Developer has already trained a lot of zealot at game time 7:26.

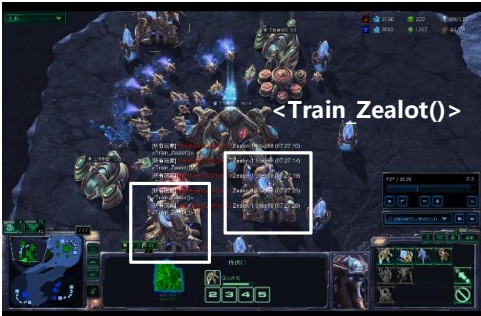

(4) **Example-5, train too many low-level units (after executing actions)**. LLM call for Zealot training even after Stalker/Adept/Sentry available.

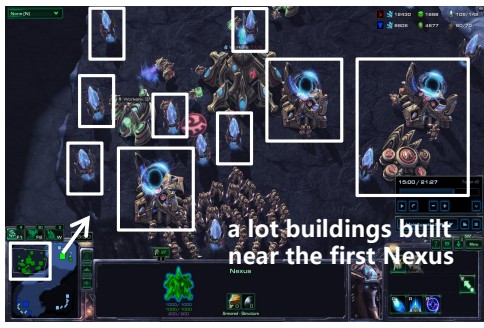

(5) **Example-6, build too many buildings around base building 1 (standard build mode, screen of base building 1)**. Builder only build around the first base and they do not build enough unit training buildings due to frequently calling for build at invalid position.

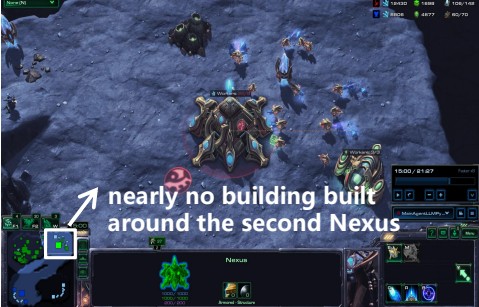

(6) **Example-6, build too many buildings around base building 1 (standard build mode, screen of base building 2)**. There is enough area for build buildings around the second base. But the Builder never ask the probe go to the second base to build buildings.

Figure E4: **Hallucination Examples in Complete StarCraft II Games (standard build mode).**

**Observation of CombatGroup0 in Example-5**

Game Info:
  Time: 7:26, Minerals: 3230, Vespene: 218
  Supply Total: 110, Supply Left: 13, Supply Used: 97

Unit Counts:
  Our Unit:
    {'Probe': 49, 'Zealot': 20}
  Our Buildings:
    {'Gateway': 3, 'Assimilator': 2, 'CyberneticsCore': 1, 'Pylon': 10, 'Nexus': 2}
  Military Buildings:
    {'Gateway': '3 (2 is working, 1 is idle)'}
  Research Buildings:
    {'CyberneticsCore': '1 (1 is working, 0 is idle)'}
  Our Unit (in warping/morphing):
    None
  Our Buildings (in construction):
    None

Valid actions:
  Team Protoss-Buildings-1:
    <Train_Adept()>  cost: {'mineral': 100, 'gas': 25, 'supply': 2}, note: we can afford 8 at most
    <Train_Sentry()>  cost: {'mineral': 50, 'gas': 100, 'supply': 2}, note: we can afford 2 at most
    <Train_Stalker()> cost: {'mineral': 125, 'gas': 50, 'supply': 2}, note: we can afford 4 at most
    <Train_Zealot()>  cost: {'mineral': 100, 'gas': 0, 'supply': 2}, note: we can afford 32 at most
    <Research_WarpGate()>        cost: {'mineral': 50, 'gas': 50}
    <Build_Assimilator()>        cost: {'mineral': 75, 'gas': 0}
    <Build_Pylon()>            cost: {'mineral': 100, 'gas': 0}
    <Build_Gateway()>          cost: {'mineral': 150, 'gas': 0}
    <Build_CyberneticsCore()>    cost: {'mineral': 150, 'gas': 0}
    <Build_Forge()>            cost: {'mineral': 150, 'gas': 0}
    <Build_TwilightCouncil()>    cost: {'mineral': 150, 'gas': 100}
    <Build_Stargate()>         cost: {'mineral': 150, 'gas': 150}
    <Build_RoboticsFacility()>   cost: {'mineral': 150, 'gas': 100}
    <ChronoBoost_Research()>
......

Communication information:
  From Commander:
    Please expedite the training of additional Zealots and ensure the Pylon is completed as soon as
    possible to support our ongoing assault. We need to maintain pressure on the enemy.
Available Communication Targets: {available_communication_targets}
Available Communication Functions: {available_communication_functions}
Args explanation: {args_explanation}

Tasks:
  Team Protoss-Buildings' task: Develop economy, technology, train units through 'Actions' to win the
  game.
Please note that **Tasks** are the most important information, all your decisions must aimed at
completing the tasks.

As a senior commander, the max number of your actions is not limited, when you warp units, try to use
all the WarpGate as much as possible, and warp all units near a single WarpTrain Field Provider.
Now, start generating your analysis, actions and communication:

Figure E5: **Observation of Developer in Example-5.**

**Response of Developer in Example-5**

Analysis:
   We currently have 13 supply left, which is quite low. We need to build a Pylon to increase our supply capacity. Additionally, we have a significant amount of minerals (3230) and some gas (218), allowing us to train more units. The Commander has requested the expedited training of additional Zealots, which aligns with our current strategy to maintain pressure on the enemy...

Actions:
   Team Protoss-Buildings-1:
      <Build_Pylon()>
      <Train_Zealot()>
      <Train_Zealot()>
      ....

Communications:
<MessageTo(Commander, '"We are building a Pylon to increase supply and training additional Zealots to maintain pressure on the enemy. "')>"

---

**Hallucination Analysis of the Response in Example-5**

(1) Agent Commander do not realise that we already have too mant Zealots, and continues sending messages to Developer to train more Zealots.

(2) Agent Developer completely followed the commander's intention but lost his own thinking. Even the Developer knows that actions like <Train_Stalker()> and <Train_Adept()> is available, it do not send the information to Commander and ask for pemission for training other units.

Figure E6: **Response and Hallucination Analysis of Developer in Example-5**

## E.3 Hallucination Examples in micro-operation scenarios

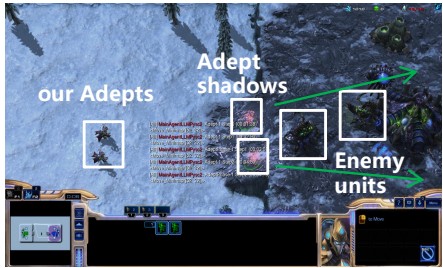

(1) **Example-7, skill use (correct strategy)**. Adepts should use the shadow to avoid damages.

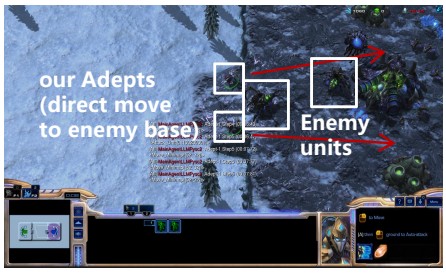

(2) **Example-7, skill use (incorrect strategy)**. Sometimes LLM tries to directly enter enemy's base.

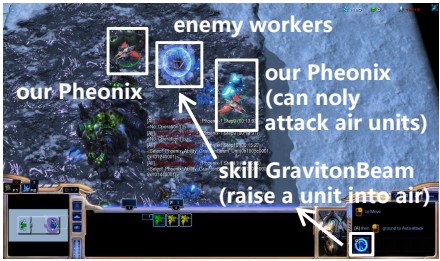

(3) **Example-8, skill use (before query)**. Last step Pheonixes use its skill GravitonBeam raise a worker to the air and finished it successfully.

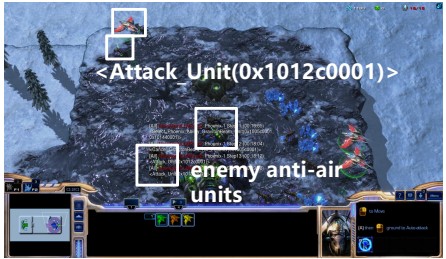

(4) **Example-8, skill use (after executing actions)**. LLM tries to kill another worker but forget to use skill to raise it into the air.

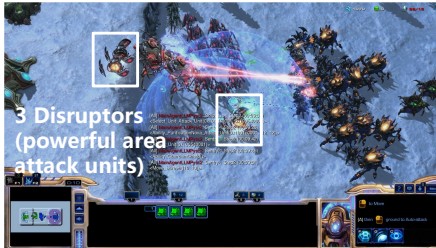

(5) **Example-9, collaboration (before query)**. Our combat group engage with enemy units.

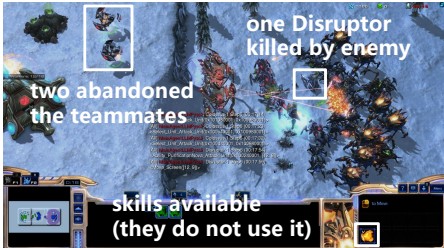

(6) **Example-9, collaboration (after executing actions)**. Disruptors retreat before using skills to kill any of the enemy unit.

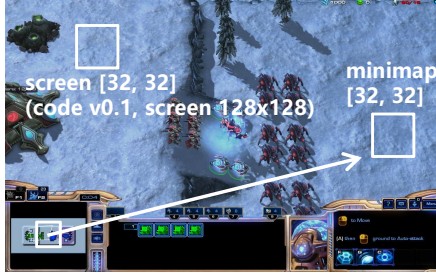

(7) **Example-10, mistaken coordinate system (before query)**. CombatGroup5 are assigned with a task to go to minimap [32, 32].

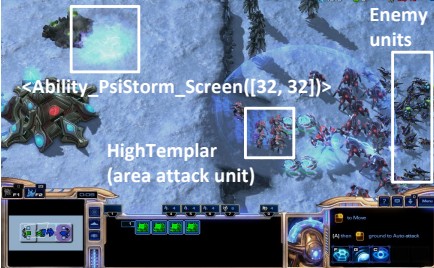

(8) **Example-10, mistaken coordinate system (after executing actions)**. LLM tries to use PsiStorm to attack enemy units but release at a wrong place.

Figure E7: **Hallucination Examples in micro-operation scenarios**

**Observation of CombatGroup8 in Example-8**

Game Info:
Time: 0:19

Team Phoenix-1 Info:
  Team minimap position: [51, 31] (minimap coordinate valid range for actions: 0 < x < 64, 0 < y < 64)
  Team screen edge (screen coordinate valid range for actions: 0 < x < 11, 0 < y < 23)
  Warning! controlled team near the map edge! Pay attention to using coordinates within the boundary!(0 < x < 11, 0 < y < 23)
  Controlled Team Units:
    Unit: Phoenix   Tag: 0x1005c0001   ScreenPos: [12, 11]   Health: 72(40 %)   Energy: 101
    Unit: Phoenix   Tag: 0x100040001   ScreenPos: [12, 12]   Health: 156(86 %)   Energy: 200
    Unit: Phoenix   Tag: 0x100400001   ScreenPos: [1, 4]   Health: 54(30 %)   Energy: 200
  Nearby Enemy Units:
    Enemy Unit: Drone    Tag: 0x101380001    ScreenPos: [8, 13]   Distance: 4   Health: 40(100 %)
    Enemy Unit: Drone    Tag: 0x101400001    ScreenPos: [9, 14]   Distance: 4   Health: 40(100 %)
    Enemy Unit: Drone    Tag: 0x1013c0001    ScreenPos: [8, 16]   Distance: 7   Health: 40(100 %)
    Enemy Unit: Drone    Tag: 0x1014c0001    ScreenPos: [7, 15]   Distance: 5   Health: 40(100 %)
    Enemy Unit: Drone    Tag: 0x1012c0001    ScreenPos: [9, 15]   Distance: 6   Health: 40(100 %)
    Enemy Unit: Drone    Tag: 0x101300001    ScreenPos: [0, 0]   Distance: 12   Health: 40(100 %)
    Enemy Unit: Drone    Tag: 0x101340001    ScreenPos: [9, 12]   Distance: 3   Health: 40(100 %)
    Enemy Unit: Drone    Tag: 0x101480001    ScreenPos: [1, 0]   Distance: 11   Health: 40(100 %)
    Enemy Unit: Drone    Tag: 0x101280001    ScreenPos: [9, 8]   Distance: 0  Health: 40(100 %)
    Enemy Unit: Hatchery   Tag: 0x101100001   ScreenPos: [4, 11]   Distance: 4   Health: 1500(100 %)   Buff: Buffs.QueenSpawnLarvaTimer
    Enemy Unit: Queen   Tag: 0x100240001   ScreenPos: [3, 14]   Distance: 7   Health: 175(100 %)   Energy: 40
    Enemy Unit: Queen   Tag: 0x100540001   ScreenPos: [4, 8]   Distance: 4   Health: 175(100 %)   Energy: 15
    Enemy Unit: Cocoon   Tag: 0x1011c0001   ScreenPos: [3, 14]   Distance: 7   Health: 200

Relevant Knowledge: {relevant knowledge}

......

Valid Actions:
  Team Phoenix-1 Valid Actions:
    <Move_Minimap(minimap)>
    <Move_Screen(screen)>
    <Attack_Unit(tag)>
    <Select_Unit_Attack_Unit(tag, tag)>
    <Select_Unit_Move_Screen(tag, screen)>
    <Select_Phoenix_Ability_GravitonBeam_Unit(tag, tag)>

Tasks:
Team Phoenix-1' task: Kill as much as enemy **Drones** as possible until all units dead.
Please note that **Tasks** are the most important information, all your decisions must aimed at completing the tasks.

Give each team no more than 3 actions, these actions will be executed in the following 1.0 seconds, among which activity release should usually before attack and move.
Now, start generating your analysis, strategy and actions:

Figure E8: **Observation of CombatGroup8 in Example-8.**

**Response of CombatGroup0 in Example-8**

**Analysis:**
- The task is clear: we need to kill as many enemy Drones as possible until all units are dead.
- **Team Phoenix-1** has three units, two of which are low on health, and one with significant health remaining. We also need to pay attention to the proximity of the enemy Drones to maximize damage.
- The Drones are spread across the map, but the most immediate threat is the Drone at [9, 8] which is directly in range (distance 0) from the team. The other nearby Drones are also very close, at distances of 3 to 7.
- Two Phoenix units are under attack, and one is being healed. The situation suggests that focusing on eliminating nearby Drones quickly is the best course of action.

**Strategy:**
1. Focus fire on the Drone at [9, 8], as it is closest and will reduce the enemy's presence on the map significantly.
2. After eliminating the Drone at [9, 8], shift attention to the Drone at [9, 12] and [8, 13] (next closest units).
3. Prioritize attacking Drones over other enemy units (like the Hatchery or Queens) to complete the task efficiently.

**Actions:**
Team Phoenix-1:
1. <Attack_Unit(0x101280001)> # Target the closest Drone at [9, 8]
2. <Attack_Unit(0x101340001)> # Target the Drone at [9, 12] next
3. <Attack_Unit(0x101380001)> # Target the Drone at [8, 13] to ensure damage continues

The actions focus on eliminating the nearest Drones and continuing the task of killing as many Drones as possible.

---

**Hallucination Analysis of the Response in Example-8**

(1) As shown in the observation text, no Drone has the buff "Buffs.GravitonBeam",which means all the enemy workers are on the ground, and our Pheonix can not attack ground units.

(2) As shown in Fig. E6 (3), the agent used to release the skill
<Select_Phoenix_Ability_GravitonBeam_Unit(tag, tag)>
to successfully killed a worker, but it do not realise this skill is the key to make enemy units attackable.

Figure E9: **Response and Hallucination Analysis of CombatGroup8 in Example-8**

# Appendix F. Examples pf Structured StarCraft II Wiki Knowledge

```
Knowledge Schema (Wikipedia-Derived)

 {
  'name': str,
  'description': str,
  'type_self': list[str],
  'type_anti': list[str],
  'health': int, 'shield': int, 'health_armor': int,
  'speed': float,
  'weapon': {
     'target': list[str],
     'damage': int,
     'bonus': dict,
     'range': int,
     'cooldown': float
  },
  'cost': { 'mineral': int, 'gas': int, 'supply': int },
  'produce_from': str,
  'ability': dict,
  'upgrade': dict,
  'requirements': {
     'building': list[str],
     'tech': list[str]
  }
 }
```

Figure F1: **Knowledge schema of the structured StarCraft II wiki database used in LLM-PySC2**

```
Example: Protoss Unit "Adept"

 {
  'name': 'Adept',
  'description': 'Ranged light infantry with psionic transfer for harassment.',
  'type_self': ['light', 'biological'],
  'type_anti': ['light'],
  'health': 70, 'shield': 70, 'health_armor': 1,
  'speed': 2.5,
  'weapon': {
     'target': ['ground'],
     'damage': 10,
     'bonus': {'light': 12},
     'range': 4,
     'cooldown': 2.25
  },
  'cost': {'mineral': 100, 'gas': 25, 'supply': 2},
  'produce_from': 'Gateway / WarpGate',
  'ability': {'Psionic Transfer': 'Teleport via shade after short delay.'},
  'upgrade': {'ResonatingGlaives': '+45% attack speed'},
  'requirements': {
     'building': ['Gateway', 'Cybernetics Core'],
     'tech': []
  }
 }
```

Figure F2: **Example Protoss unit entry ("Adept") from the structured wiki knowledge base.**

## Appendix G. Other supplementary materials

### Limitations.

Although the LLM-PySC2 environment provides a lot more features than existing environments, there are still some limitations. For example: (1) We found problems in LLM-based decision-making but have not yet found a solution that is good enough. How to learn domain-specific knowledge in the environment effectively remains a problem. (2) Due to the limited decision-making ability of the LLMs, many professional operations (such as multi-line attacks, invisible units harassment, airdrop, changing homes, etc.) have not appeared in our experiments. As a result, it is quite challenging for us to test these advanced tactics, leading to potential bugs when dealing with advanced strategies. (3) Considering that Protoss is enough for decision-making method verification, we suspend the work of Terran and Zerg. Full support for these two races will be provided in the future.

### Impact Statements.

This work presents a new environment for evaluating LLM decision-making performance and developing learning algorithms. With complete StarCraft II action space, our environment provides far more complexity than other decision-making platforms. With the asynchronous query architecture, the environment efficiently interacts with LLMs that maintain a constant latency regardless of the scale of the agents' population. Advancement in the environment reduces the obstacles of large model decision research and prepares conditions for post-training LLMs in sequential decision problems. The phenomena observed in the experiments will attract more attention to the LLM hallucinations and lack of knowledge in specific domains.

### Broader Impact.

Our work opens up new opportunities for applying LLM in gaming and other multi-agent collaboration applications. Given that a lot of work on LLM learning methods is still supervised learning or human feedback reinforcement learning on data sets, our environment may promote the research of LLM in-environment learning. Potential negative impacts include misuse of the environment to develop autonomous decision-making systems without enough safety control and human control interfaces, which may generate uncontrolled, unexpected, or dangerous behavior in the application. We suggest strengthening the research on LLM security while promoting the research on large-model autonomous decision making.

### Code Of Ethics.

This research fully adheres to the NeurIPS Code of Ethics. Violence behaviors of LLMs, such as attacking and harassing are confined entirely to the Real-Time Strategy game StarCraft II, just like previous similar works like SC2LE and SMAC. Potential risks of misuse and our suggestions are discussed in Section *Broader Impact*. This work does not involve or address any topics related to discrimination, races mentioned in the paper are the camps in the StarCraft II game.

