# OpenReview forum: "LLM-PySC2: Starcraft II learning environment for Large Language Models"
_NeurIPS.cc/2025/Conference — NeurIPS 2025 poster_

### Official Review · Reviewer_k26i · 2025-06-04

**Clarity:** 4
**Significance:** 3
**Originality:** 3
**Rating:** 4
**Confidence:** 4

**Summary:**

This paper transforms StarCraft II into an environment manageable by multimodal large language models. The author set up various in-game tasks and evaluated different types of large language models within this environment.

**Questions:**

See weaknesses.

**Ethical Concerns:**

["NO or VERY MINOR ethics concerns only"]

**Final Justification:**

I will retain my score (4)

**Limitations:**

Yes.

**Paper Formatting Concerns:**

Page 5, fourth line from the bottom, typo

**Quality:**

3

**Strengths And Weaknesses:**

Strengths
- Game AI itself is an interesting topic, and the multi-modal approach aligns with the current development direction of large models. The StarCraft II environment is also quite challenging, making it suitable for evaluation.
- The designed method is detailed and reasonable, and experiments have proven that large models can exhibit a certain level of capability.

Weaknesses
- Lack of intuitive description of the final capabilities. For example, what human-level equivalent was achieved? How do previously proposed agents perform on these tasks?
- Ablation studies for the method design are lacking. For example: The proposal that LLMs should return reasons for decisions—does this improve decision-making ability? By how much?
- The paper suggests in the experimental section that reasoning models similar to GPT-o1 perform poorly in unseen scenarios. This seems to contradict other similar works. For example, Gemini recently used its multi-modal reasoning capabilities to complete the game Pokémon Blue. Perhaps this is because the models used in this paper are not new enough; considering GPT-o3/o4-mini or Gemini-2.5-Pro might be worthwhile.
- The other models used are generally not new enough; further supplementation could be considered.

---

> ### Author Rebuttal · Authors · 2025-07-29
>
> Dear reviewer,
>
> We sincerely thank you for your time, thoughtful comments, and constructive suggestions. We greatly appreciate your efforts in reviewing our work and providing feedback that will help us improve the quality and clarity of the paper. Below, we provide point-by-point responses to each of the raised concerns.
>
> ---
>
> Question1: [Lack of intuitive description of the final capabilities. For example, what human-level equivalent was achieved? How do previously proposed agents perform on these tasks?]
>
> Response1:[Thank you for the insightful comment. We agree that a clearer contextualization of agent capabilities is valuable.
>
> (1)About human-level equivalent:
>
> Our current system has not yet been connected to Blizzard's official ranked matchmaking platform, which limits our ability to directly benchmark LLM performance against human players in a standardized environment. However, we provide evaluation results against the built-in bots as a proxy for capability.
>
> As shown in Table 1 of our paper, without any specialized prompt design, fine-tuning, or RL training, LLM-based agent is capable of defeating the Level-6 (Very Hard) built-in bot. This difficulty level is known to surpass entry-level human players and is generally considered to match the skill level of amateur participants in community tournaments. This demonstrates the model’s ability to generalize strategic behaviors directly from text observations and instructions.
>
> (2)About previously proposed agents:
>
> Regarding previously proposed agents, most existing StarCraft research relies on small models trained with reinforcement learning. These agents have achieved impressive results over the years but suffer from practical limitations such as poor generalization, lack of interpretability, limited support for human interaction, and susceptibility to reward hacking to an early rush strategy. In contrast, our platform is the first designed specifically for LLMs and enables a broader range of tasks including natural language interaction and long-horizon planning. To our knowledge, no prior work has deployed LLM agents in such full-action-space StarCraft environments, making our platform a unique and foundational step in this direction.]
>
> ------
>
> Question2: [Ablation studies for the method design are lacking. For example: The proposal that LLMs should return reasons for decisions—does this improve decision-making ability? By how much?]
>
> Response2:[Thank you for this valuable suggestion. The primary focus of this work is to construct a StarCraft II learning environment tailored for large language models (LLM-PySC2), and to conduct preliminary investigation into LLMs’ limitations. So, we have not conducted in-depth ablation studies on specific elements for the LLM performance, we leave these questions to future researches and hope the platform promotes research on LLM decision-making system.
>
> We agree that such studies would be highly insightful and essential for understanding the contribution of each component in the overall framework. Considering the intense timeline, we cannot guarantee such ablation experiments can be completed before the final submit deadline. We appreciate the reviewer’s recommendation and will try to including such ablation experiments in the final submittion or future arxiv versions of our work.]
>
> ---
>
> Question3: [The paper suggests in the experimental section that reasoning models similar to GPT-o1 perform poorly in unseen scenarios. This seems to contradict other similar works. For example, Gemini recently used its multi-modal reasoning capabilities to complete the game Pokémon Blue. Perhaps this is because the models used in this paper are not new enough; considering GPT-o3/o4-mini or Gemini-2.5-Pro might be worthwhile.The other models used are generally not new enough; further supplementation could be considered.]
>
> Response3: [Thank you for raising this important point. There are several reason that we have not test models new enough.
>
> (1) Price.
>
> Run a LLM Starcraft-II game requires much more interactions than traditional query-answer problem, especially for reasoning model. For Instance, for 10 games of a senario, o1-mini receive 2000 prompt tokens and generate about 3000 tokens per step, and a complete game consist of about 600 steps interactions, with several agents (5 for average, as an instance) work simutaniously.
>
> For o1-mini, experiment on the single scenario cost about 10x2000x600x5 = 60million input tokens, with 10x3000x600x5 = 90million response tokens. Considering the price that ‘1.10dollars/m tokens’ for input tokens and ‘4.40dollars/m tokens’ for response tokens (all the o-mini models share this price, as of 2025/7/29), it is expensive and also unfriendly for carbon emission for experiment even on a single scenario.
>
> (2) Rapid emergence of new models.
> Run complete experiments of a model cost a lot of times, especially those with strict concurrent limit or generating thousands of reasoning tokens. In the past half year before NeurIPS submittion deadline, GPT-O3、DeepSeek-V3/R1、Qwen3、llama4、Claude3.7、Gemini-2.5-Pro have been proposed. We found that our experimental speed couldn't even keep up with the speed at which the new model appeared, so we gave up following the models that is too new.
>
> We fully agree that including a wider range of open-source models would strengthen the evaluation of our environment. We may include additional results of other models (those do not cost too much) if time permitting before the final submission deadline. If we cannot finish the additional evaluations before the deadline, we will consider release these extended results in the future version of arXiv paper.]
>
> ---
>
> Once again, we thank you for your valuable feedback and hope that our clarifications and updates address your concerns.
>
> Sincerely,
>
> The Authors

---

> > ### Comment · Reviewer_k26i · 2025-08-03
> >
> > Thanks for the author's response.
> >
> > My primary concern remains the timeliness of the evaluated models. I acknowledge the practical challenges mentioned by the authors. However, I do believe that without evaluating the latest models, many of the paper's insights risk being incomplete or outdated, significantly diminishing its value.
> >
> > For this reason, I will maintain my current score.

---

> > > ### Author Response · Authors · 2025-08-07
> > >
> > > We sincerely thank you for your time, effort, and constructive feedback in the review and comments. Your insights have helped us identify important aspects for clarification and improvement, and we greatly appreciate the thoughtful suggestions that contributed to refining the quality of our work.
> > >
> > > We will try to add more evaluation results for latest models in the final submission or in a future arxiv version of the paper.
> > >
> > > Once again, thank you for your efforts in reviewing the paper.

---

### Official Review · Reviewer_fD6D · 2025-06-30

**Clarity:** 2
**Significance:** 2
**Originality:** 2
**Rating:** 4
**Confidence:** 4

**Summary:**

The research introduces LLM-PySC2, a novel StarCraft II learning environment designed for LLMs, addressing the current lack of support for LLMs in complex real-time strategy games. Main features of this environment include complete PySC2 action space, multi-modal information, game knowledge from a Wiki and an asynchronous query architecture to manage interactions with multiple LLM agents. Experiments show that while LLMs can make decisions, their performance can be inconsistent, exhibiting issues like hallucinations and inefficient collaboration due to a lack of domain-specific knowledge and instruction. The authors hope LLM-PySC2 will foster advancements in LLM-based decision-making.

**Questions:**

__Question on asynchronous query architecture__

Can the authors explain whether the asynchronous query architecture refers solely to calling the LLM APIs concurrently either by multi-threading or asyncio? If so, I would consider this a basic implementation necessity rather than a major feature contribution.

__Questions on wiki knowledge__

The author mentions that LLM-PySC2 offers game wiki knowledge. I don’t find much details on this other than the information offered as part of the prompt. Can the authors clarify whether this is what they mean by wiki knowledge?

**Ethical Concerns:**

["NO or VERY MINOR ethics concerns only"]

**Final Justification:**

The authors have addressed my comments.

**Limitations:**

Yes

**Quality:**

1

**Strengths And Weaknesses:**

__Strengths__

Compared to the existing benchmark that evaluates LLM policy on SC2, the environment introduced in this paper provides support for complete action space.
Also provide support for multi-modal observations such as RGB screen and RGB feature which would be very valuable for benchmarking multi-modal LLMs.

__Weaknesses__

It’s hard to assess the novelty/impact of this paper due to the limited evaluation presented in this paper. For example, the author mentioned the main contribution of this work as the support for complete action spaces. While this makes the environment more challenging, it’s difficult to understand immediately what kind of failure modes this immediately exposes that future work can research on.

__Limited evaluation__

The evaluations in this paper seem restricted to comparison between different LLM backbones. This offers limited insights on how LLM-backed agents compare with existing state-of-the-art RL agents (e.g. AlphaStar etc). Without comparison with more “traditional” RL agents, it is difficult to understand the strengths and weaknesses of LLM agents which I think is crucial for future work.

Nevertheless, the authors did mention that “It is not recommended to compare the LLM-based method with the MARL-based method due to different control frequencies.” I am not really sure about what the authors mean. Why can’t we run the LLM and MARL -based agents at their respective control frequency and evaluate the results?

I would like to see how the LLM-based agents perform compared to existing MARL approaches in order for me to raise my score.

__Nit__
* Rather than putting the illustration for all hallucination examples together. It would be more useful to present the illustration followed by each failure analysis separately. That would make the readers’ life easier when trying to understand the examples.
For the tables on evaluation performance. Do you run the evaluation with multiple seeds? If not I would consider doing it since SC2 are highly complex environments where the variance may be huge. If so, it would be great if you can include the confidence intervals.

---

> ### Author Rebuttal · Authors · 2025-07-29
>
> Dear reviewer,
>
> We sincerely thank you for your time, thoughtful comments, and constructive suggestions. We greatly appreciate your efforts in reviewing our work and providing feedback that will help us improve the quality and clarity of the paper. Below, we provide point-by-point responses to each of the raised concerns.
>
> ---
>
> Question1: [Question on asynchronous query architecture. Can the authors explain whether the asynchronous query architecture refers solely to calling the LLM APIs concurrently either by multi-threading or asyncio? If so, I would consider this a basic implementation necessity rather than a major feature contribution.]
>
> Response1: [We appreciate your question. Our asynchronous query architecture goes significantly beyond simply invoking LLM APIs concurrently via multi-threading or asyncio. Due to the complex interaction patterns required in a PySC2-based environment—especially in managing camera-based observation acquisition and multi-agent coordination—the design and implementation of our asynchronous system is far from trivial.
>
> Specifically, our architecture includes:
>
> (1) Thread Resource Scheduling:
>
> The system must first schedule the camera to collect pysc2 observation object ‘obs’ for each agent. A dedicated thread is launched for each agent to process both textual and visual observations. Within this thread, an additional llm-client thread is spawned to handle LLM communication, monitor responses, and post-process actions.
>
> (2) Error Handling:
>
> Our system incorporates extensive safeguards to detect and recover from various error types—such as network disconnections, timeout issues, and corner-case exceptions arising from obs/action module crashes (consider the size of obs/action space, this contains a lot of works). Since a failure in any single thread can risk bringing down the entire program, we implemented comprehensive error isolation and recovery mechanisms within the asynchronous framework to maintain system stability.
>
> Prior environments for LLM-based sequential decision-making typically involve single-agent or limited multi-agent settings. In contrast, our LLM-PySC2 framework scales up to 13 or more LLM agents concurrently. This necessitated a carefully designed asynchronous architecture tailored for safety, robustness, and concurrency in large-scale multi-agent settings.]
>
> ---
>
> Question2: [Questions on wiki knowledge. The author mentions that LLM-PySC2 offers game wiki knowledge. we don’t find much details on this other than the information offered as part of the prompt. Can the authors clarify whether this is what they mean by wiki knowledge?]
>
> Response2: [Thank you for pointing this out. We would like to clarify that “wiki knowledge” we are referring to is a structured and comprehensive knowledge base manually constructed from the official StarCraft II wiki pages. Specifically, we extracted detailed information for all units across the three races—amounting to approximately 120 units—by manually transcribing relevant data into a JSON-formatted Python dictionary. Since all information is faithfully extracted from the official StarCraft-II wiki webpage, we refer to this component as “wiki knowledge.”
>
> This structured wiki knowledge base is included in the provided anonymous GitHub package, located at <project_path>/llm_pysc2/lib/knowledge/<race>.py. It contains detailed attributes for each unit, including:
>
> - Name, description
> - Attack, shields, health
> - Description and data for Skills
> - Prerequisite units and technologies
> - Resource cost and training time
> - ......
>
> Altogether, this wiki database spans over 6,000 lines of structured content, faithfully representing the official StarCraft II unit specifications.
> In the default LLM-PySC2 usage, we incorporate a knowledge-query module that retrieves wiki entries relevant to the agent's current game situation as part of the prompt. To further address your concern, we will include illustrative examples of the wiki knowledge and its usage in the appendix of the revised paper.]
>
> ---
>
> Additionally, we noticed that you pointed out several issues in the Weaknesses section that were not explicitly raised as formal questions in the Questions to Authors section. We would like to briefly address these points here, in the hope that our responses help clarify your concerns.
>
> ---
> Weakness 1: [It’ s hard to assess the novelty/impact of this paper due to the limited evaluation presented in this paper. For example, the author mentioned the main contribution of this work as the support for complete action spaces. While this makes the environment more challenging, it’ s difficult to understand immediately what kind of failure modes this immediately exposes that future work can research on.]
>
> Response: [We appreciate your concern regarding the evaluation and the clarity of potential failure modes introduced by supporting the complete action space. As our primary contribution lies in building a scalable and LLM-friendly StarCraft II environment, we do not focus on exhaustively analyzing the failure modes of various LLMs. Nevertheless, we have included preliminary observations of such failure cases in the Discussion section and detailed further examples in Appendix E. We hope these can serve as a starting point for future investigations.
>
> Constructing an LLM StarCraft-II environment with complete SC2 action space is a non-trivial task. Even three years after GPT3’s emergence, there is no other works that have managed to achieve this. This speaks to both the technical challenge and the novelty/impact of our contribution. Our aim is for the environment to play a foundational role in the LLM decision-making research community, much like how SMAC catalyzed progress in MARL. With an environment in place, we open the door for future research to systematically investigate in-depth failure modes and develop effective decision-making strategies.]
>
> ---
>
> Weakness 2: [Limited evaluation, especially how LLM-based agents compare with existing state-of-the-art RL agents (e.g. AlphaStar etc). Why can’ t we run the LLM and MARL -based agents at their respective control frequency and evaluate the results?]
>
> Response: [We appreciate your comment. While we fully agree that comparing LLM-based agents to MARL baselines such as AlphaStar is an important long-term direction, we would like to clarify why such comparisons are currently non-trivial.
>
> First, from a performance perspective, the RL- agents with small model have already achieve near-100% winning rates across all PySC2 scenarios. Given these strong results, it is highly unlikely that current LLM agents can directly outperform the RL agents in these tasks, especially considering without any in-environment training. That said, our core contribution is not about outperforming AlphaStar or existing RL baselines, but rather about establishing a new platform to explore the emerging capabilities of LLMs in sequential decision-making.
>
> We outline three key reasons why we do not recommend direct comparisons between LLM-based and traditional MARL agents in this work:
>
> (1)Agent Configuration Differences:
>
> Traditional MARL environment (e.g., SMAC) typically assign one agent per unit, whereas LLM-PySC2 uses one LLM agent to control a squad of units. This design was intentional to reduce communication overhead and computational cost for LLMs.
>
> (2)Control Frequency Mismatch:
>
> SMAC environments typically support high-frequency per-unit control. In contrast, LLM-PySC2 supports a maximum of 2 steps per second and controls not so micro like Traditional RL agent in SMAC.
>
> (3)Action Space Discrepancies:
>
> SMAC uses discrete action spaces (e.g., move directions and attack commands), while LLM-PySC2 allows for rich, continuous choices including arbitrary movement coordinates and target selection via unique unit tags. This results in significantly different control granularity and decision complexity.
>
> Moreover, we note that the LLM agents in our platform are not trained in the environment—they rely solely on the knowledge and ability from pretraining. In contrast, MARL agents finished extensive task-specific training (e.g., 1e6+ steps in SMAC). We believe that future work may explore RL fine-tuning of LLMs within LLM-PySC2 and conduct direct comparisons under more aligned assumptions.
>
> In summary, our goal in this work is to build a platform rather than claim performance dominance. We hope this clarifies our design decisions and encourages further exploration of LLM capabilities using the platform we provide.]
>
> ---
>
> Weakness 3: [Nit. (1) About writing in hallucination examples. (2) About experiment seeds (3) add confidence intervals. ]
>
> Response: [
> (1) We sincerely thank you for your suggestions regarding the writing of the paper. We will revise the description of the examples in the appendix section of the final version to improve the clarity and readability of the paper.
>
> (2)By default, the game runs with a randomly assigned seed—we did not manually set a fixed seed. Additionally, due to the use of large model API services, the inherent randomness in LLM sampling cannot be controlled via user-seeding. Therefore, all experiments were conducted under stochastic conditions.
>
> (3) We agree that including confidence intervals can provide a more rigorous representation of the experimental results. While most classical RL papers and recent LLM-based works in gaming scenarios do not typically report confidence intervals, we will still make an effort to increase the number of experimental runs before the final submission and include the calculated confidence intervals in the final version of the paper.]
>
> ---
>
> Once again, we thank you for your valuable feedback and hope that our clarifications and updates address your concerns.
>
> Sincerely,
>
> The Authors

---

> > ### Comment · Reviewer_fD6D · 2025-08-03
> > **Response to author rebuttal**
> >
> > > structured wiki knowledge base
> >
> > Thanks for the clarification. I can find the related details in the repo. Just to mention, currently it's implemented for terran and zerg but not the others right?
> > Also, in the knowledge you have included code comments about the individual fields but I don't think they are included as part of the prompt. In my opinion, some of the comments are fairly useful for the LLMs to infer the meaning of the fields so it would be really useful to include them in the prompt in future work.
> >
> > >  That said, our core contribution is not about outperforming AlphaStar or existing RL baselines, but rather about establishing a new platform to explore the emerging capabilities of LLMs in sequential decision-making.
> >
> > I agree that the main motivation is to develop a platform, not to outperform existing RL baselines. Still, I believe both a quantitative and qualitative comparison with RL baselines would be very valuable addition to this work as it can highlight where the LLM agents fall short.
> >
> > > Control Frequency Mismatch
> >
> > I still disagree that this would mean that a comparison is discouraged. Since the LLM agents are controlling at a more macro level compared to the more traiditional RL baselines, you also lose what granularity that the other agents can provide. It would be interesting to see what that entails in practice if you have results comparing with those agents.
> >
> > **Overall**
> > Based on the author's rebuttal, I will consider raising my score to borderline accept. Authors, please consider incorporating some of the details mentioned in the rebutall as part of your main text. Based on comments from other reviewers, a lot of the technical design details appear missing in the paper and it would be nice if you include those in the main paper (if space allows) or the appendix.
> >
> > Reasons for not raising my score further:
> >
> > This paper overall looks more like a dataset/benchmark paper. However, the implementation is incomplete and I have browsed through the code and it would still need some polish and cleanup for it to be useful to the broader community. I trust the authors would continue working on this and the borderline acceptance rating is based on such consideration.

---

> ### Author Response · Authors · 2025-08-05
>
> Thank you very much for your thoughtful comments and for considering raising your score to borderline accept. We truly appreciate the time and care you have taken in reviewing both our paper and codebase.
>
> The structured wiki knowledge base currently covers all three races, Protoss, Terran, and Zerg. We also agree that including the code comments in the prompt is a valuable suggestion, and we will explore this in future work.
>
> We acknowledge that adding quantitative comparisons with RL baselines would strengthen the work. We appreciate your point about control frequency mismatch, and we will try to realize the comparisons with the traditional RL environment/method.
>
> We also take your feedback on details of the code in the paper. We will continue to polish and improve the implementation for broader community use.
>
> Thank you again for your constructive feedback and support.

---

### Official Review · Reviewer_M1d6 · 2025-07-03

**Clarity:** 2
**Significance:** 3
**Originality:** 3
**Rating:** 4
**Confidence:** 4

**Summary:**

The main contribution of this paper is to propose the LLM-PySC2 environment, which is the first environment to provide a complete PySC2 action space for LLMs, and supports multi-agent collaboration and communication.The article introduces the environment framework, action space, observation space and multi-agent system in detail, and evaluates the performance of LLMs in macro decision-making and micro operation scenarios through experiments.

**Questions:**

1. A key issue needs to be clarified: the efficiency performance of the action recognition module in handling large amounts of text and complex actions, as well as how to deal with action texts generated by LLMS that do not meet expectations, such as through what mechanisms to correct or ignore incorrect actions.

2. To enhance the observation and conversion efficiency, whether optimization measures have been taken, such as how to reduce data transmission volume and lower processing latency, as well as the impact of these optimizations on the overall system performance.

3. It is necessary to explain how the action recognition and observation conversion module ADAPTS to different game versions, maps and task types, as well as how to conveniently add or modify the types and contents of actions and observations to support new research requirements.

4. The paper does not have specific experimental verifications for independent modules such as action recognition and observation conversion. Although these modules are key components of the environment, no specialized tests have been conducted on their performance, accuracy, and efficiency. Instead, the focus has been placed on the LLMs's performance in the overall decision-making task.

**Ethical Concerns:**

["NO or VERY MINOR ethics concerns only"]

**Limitations:**

yes

**Quality:**

3

**Strengths And Weaknesses:**

**Strengths**

1. This paper proposed the first LLM decision environment supporting a complete PySC2 action space, providing a new research platform for the application of LLMs in complex strategy games.

2. Through experiments in the complete StarCraft II game and the newly designed LLM-PySC2 task group, the potential and limitations of LLMs in complex decision problems, particularly in the areas of hallucinations and insufficient knowledge are revealed.

**Weaknesses**

1. While comparisons have been made with some existing methods, systematic comparisons are lacking for other potentially applicable baseline methods, such as different LLM architectures or traditional reinforcement learning methods.

2. Although the article mentioned the universality of the environment, it did not specifically explain or demonstrate how to support the operations of both the human and Zerg races. The focus of the article is to demonstrate the functions and potential of the LLM-PySC2 environment, mainly illustrating the issue through the case of the Protoss.

3. Although the article provides the framework of the environment and experimental results, the detailed description of how to specifically implement key parts such as action recognition and observation transformation is not in-depth enough. For example, how to map text actions to specific PySC2 function calls? How do regular expressions precisely match various possible action formats, and how to handle situations with format errors or incompleteness.

4. This paper introduces the efficiency performance of the action recognition module in handling large amounts of text and complex actions, as well as how to deal with action texts generated by LLMS that do not meet expectations, such as through what mechanisms to correct or ignore incorrect actions.

5. The paper does not conduct ablation experiments on each component of the LLM-PySC2 environment (such as multimodal observation, asynchronous query architecture, different action space Settings, etc.) to evaluate the contribution of each component to the overall system performance.

6. The main contribution of the paper is to propose an environment rather than a method, which seems more suitable for Benchmark Track rather than Main Track.

---

> ### Author Rebuttal · Authors · 2025-07-29
>
> Dear reviewer,
>
> We sincerely thank you for your time, thoughtful comments, and constructive suggestions. We greatly appreciate your efforts in reviewing our work and providing feedback that will help us improve the quality and clarity of the paper. Below, we provide point-by-point responses to each of the raised concerns.
>
> ---
>
> Question1: [A key issue needs to be clarified: the efficiency performance of the action recognition module in handling large amounts of text and complex actions, as well as how to deal with action texts generated by LLMS that do not meet expectations, such as through what mechanisms to correct or ignore incorrect actions.]
>
> Response1:[Thank you for highlighting these points. The 8-page limitation kept details of efficiency and error handling out of the main paper. Below are some details:
>
> (1) Efficiency of Action Recognition:
>
> Our action recognition functions are implemented using traditional logic-driven code, ensuring fast execution suitable for real-time LLM decision-making. We test the action recognition latency that is about only 1ms. The delay is negligible compared to the LLM inference latency (typically ~10 seconds), environment latency does not constitute a performance bottleneck in the interactions.
>
> Considering that reporting efficiency is a very meaningful suggestion, we will include more details in the final version of the paper, either in the main text or in the appendix.
>
> (2) Error Handling of Invalid Actions:
>
> To prevent unexpected behavior or crashes in the LLM-PySC2 engine, we implement several safeguards at the environment interface level:
>
> a. Clear the text: When the action recognition module receives response, it delete symbols such as ‘-’, ‘*’, ‘ ’. Then, it identifies text actions line by line starting from the identifier 'Actions:' (Regardless of capitalization). For each line, it searches for string shape as <ActionName(xxx, xxx)>, and split into action name, parameters for future use. This step clears the text to avoid possible interference of undesirable characters.
>
> b. Action validity checks: The LLM receives the current set of valid actions, ensuring only executable actions are generated. Malformed or unrecognized actions are replaced with <No_Operation()>, which has no effect on the environment.
>
> c. Parameter validation: Each action’s parameters are checked for range and validity (e.g., building positions must be within screen bounds, in a pathable and buildable place, and in power field for Protoss). Actions with invalid parameters will be replaces by <No_Operation()>.
>
> Notably, we do not auto-correct semantically poor decisions (e.g., attack an inappropriate enemy unit or mistimed ability usage). This is by design, allowing the platform to assess and evaluate the LLM’s decision-making and instruction-following abilities in complex SC2 scenarios.]
>
> ---
>
> Question2: [To enhance the observation and conversion efficiency, whether optimization measures have been taken, such as how to reduce data transmission volume and lower processing latency, as well as the impact of these optimizations on the overall system performance.]
>
> Response2: [Thank you for the thoughtful question regarding optimization of observation and conversion efficiency.
> In our platform, the dominant performance bottleneck is the inference latency of the LLM, typically ranging from several to tens of seconds per step. In contrast, observation processing and data transmission are lightweight, operating at the millisecond level.
>
> Given this disparity, we did not prioritize further optimization of the observation module. Nonetheless, baseline profiling shows that observation collection, text conversion, and transmission together take about only 20 ms per step, which is negligible in the full interaction loop.]
>
> ---
>
> Question3: [It is necessary to explain how the action recognition and observation conversion module ADAPTS to different game versions, maps and task types, as well as how to conveniently add or modify the types and contents of actions and observations to support new research requirements.]
>
> Response3: [We agree that adaptability across different game settings and research tasks is essential, and our platform has been built with this flexibility.
>
> (1) Support for Different Game Versions and Maps:
>
> Our platform uses the pysc2 module (i.e. the core module of SC2LE proposed by DeepMind), which natively supports various StarCraft II versions and maps. The environment abstracts game state into a structured obs object (a named list), with valid actions dynamically extracted from it. This ensures the observation and actions remain valid across all versions and maps.
>
> (2) Custom Task Support and Flexibility:
>
> Researchers can define tasks via a customizable interface function. This function accesses the current obs and returns a task description for the LLM. Tasks can dynamically evolve based on game state, enabling experiments with multi-stage objectives.
>
> (3) Extensibility in adding or modifying the types of actions and observations:
>
> For actions, users can add “bridge objects”(as shown in Figure.2) that specifying an action’s textual description and pysc2 functions. For observations, the prompt is modular, with a lot of functions generating specific info blocks (e.g., global state, unit details). Users can modify or add functions to customize observed game information.
>
> These design allows seamless adaptation—researchers can create tasks, swap maps, extend actions/observations, and switch game versions while maintaining stable LLM integration.]
>
> ---
>
> Question4: [The paper does not have specific experimental verifications for independent modules such as action recognition and observation conversion. Although these modules are key components of the environment, no specialized tests have been conducted on their performance, accuracy, and efficiency. Instead, the focus has been placed on the LLMs's performance in the overall decision-making task.]
>
> Response4: [Thank you for pointing out the importance of evaluating core environment modules. We appreciate the opportunity to clarify the design and rationale behind our current evaluation strategy.
>
> Our action recognition and observation conversion modules use deterministic programmatic logic, ensuring high accuracy and reliability without requiring learning-based validation. These modules follow strict procedural rules and avoid computationally intensive operations, eliminating uncertainty.
>
> In terms of performance, these modules operate with minimal latency (a few milliseconds per step), which is negligible compared to LLM inference delays (seconds to tens of seconds). Since LLM processing dominates total interaction time, further optimization or benchmarking of these modules would yield low marginal gains. This approach aligns with standard practices in LLM-based interactive environments, where module-specific effeciency metrics are rarely reported—system validity is typically assessed via end-to-end task success.
>
> In our case, testing LLMs in actual gameplay inherently validates the environment’ s stability. If action recognition or observation parsing were flawed, LLMs would fail even in the easiest tasks. Thus, successful gameplay indirectly confirms module correctness.
>
> We recognize the value of quantitative insights. In the final submission, we will include performance statistics (e.g., average observation processing time and action parsing latency) to further clarify system efficiency.]
>
> ---
>
> Additionally, we noticed that you pointed out several issues in the Weaknesses section that were not explicitly raised as formal questions in the Questions to Authors section. We would like to briefly address these points here, in the hope that our responses help clarify your concerns.
>
> ---
>
> Weakness 1: [Systematic comparisons are lacking for other potentially applicable baseline methods, such as different LLM architectures or traditional reinforcement learning methods. ]
>
> Response: [While we fully agree that comparing LLM-based agents to traditional RL baselines such as AlphaStar is an important long-term direction, we would like to clarify why such comparisons are currently non-trivial.
>
> First, the RL- agents with small model have already achieve near-100% winning rates across all PySC2 scenarios. Given these strong results, it is highly unlikely that current LLM agents can directly outperform the RL agents in these tasks, especially considering without any in-environment training. Besides, there are three key reasons why we did not comparisons between LLM-based and traditional MARL agents:
>
> (1) Agent Configuration Differences: Traditional MARL environment (e.g., SMAC) typically assign one agent per unit, whereas LLM-PySC2 uses one LLM agent to control a squad of units.
>
> (2)Control Frequency Mismatch: SMAC typically support high-frequency control due to lightweight decision models and per-unit control. In contrast, LLM-PySC2 supports a maximum of 2 steps per second.
>
> (3)Action Space Discrepancies: SMAC uses discrete action spaces, while LLM-PySC2 allows for rich, continuous choices including arbitrary movement coordinates and target selection via unique unit tags.
>
> Moreover, we note that the LLM agents in our platform are not trained in the environment—they rely solely on the knowledge and ability from pretraining. In contrast, MARL agents finished extensive task-specific training (e.g., 1e6+ steps in SMAC). Due to the reasons mentioned above, we did not compare the performance of LLMs with that of traditional RL methods.]
>
> ---
>
> Considering the word limit for rebuttal, we can only provide brief response to some of these comments. We would be happy to discuss other weakness you mentioned in Reviewer Author Discussions stages if you are interested.
>
> Once again, we thank you for your valuable feedback and hope that our clarifications and updates address your concerns.
>
> Sincerely,
>
> The Authors

---

> > ### Comment · Reviewer_M1d6 · 2025-08-05
> >
> > Thanks to the author for the reply. Several key technical details remain insufficiently addressed:
> >
> > 1. The authors have not provided concrete examples of action space adaptation for the Zerg and Terran races, nor have they clarified whether these designs have been empirically tested.
> >
> > 2.  The implementation details regarding action recognition and observation transformation are inadequately described. In particular, the author does not explain how ambiguous or underspecified actions—such as "attack the enemy" without a specific target—are handled. Such ambiguities are common in complex real-time strategy games, and the absence of a clear disambiguation mechanism or target resolution strategy may significantly affect agent behavior consistency. The authors should clarify whether any such mechanisms are in place.
> >
> > 3. Independent evaluation of the contribution of the core components of the system is necessary.
> >
> >
> > 4.  the primary contribution of the work appears to be the development of an environment or benchmark platform rather than the proposal of a novel method. The emphasis on environment design, race-specific action space modeling, and interface standardization aligns more closely with the scope of a Benchmark Track paper than a Main Track submission.

---

> ### Author Response · Authors · 2025-08-05
>
> Thank you for your continued engagement and thoughtful feedback. We appreciate your follow-up and the opportunity to further clarify the technical aspects of our work:
>
> ---
>
> Comment1:[The authors have not provided concrete examples of action space adaptation for the Zerg and Terran races, nor have they clarified whether these designs have been empirically tested.]
>
> Response:[Thank you for the valuable comments. Considering our priority is to build an environment for LLM StarCraft-II research instead of gameplay, we believe the current Protoss race is already enough for a wide range of research. So we do not introduce complete Zerg and Terran support at this early version, which involves significant engineering complexity and effort in error handling code for all the actions.
>
> Actually, our framework supports quickly adding actions for other races due to its race-independent design. For example, we can easily define a ‘bridge object’(just like the instance in figure.2) for Zerg’s building action as: {‘name’: ‘Build_SpawningPool_Screen’, ‘arg’: ['screen'], 'func': [(85, F.Build_SpawningPool_screen, ('queued', 'screen'))]}. This object can be directly recognized by the environment, and the action will be added to the action space.
>
> In the future, we will add complete support for Zerg and Terran, but the priority may still remain in optimizing the core architecture of the platform rather than adding content for other races. We hope that the core architecture can bring maximum convenience to future developers, which serves as a strong foundation for future research.]
>
> ---
>
> Comment2:[The implementation details regarding action recognition and observation transformation are inadequately described. In particular, the author does not explain how ambiguous or underspecified actions—such as "attack the enemy" without a specific target—are handled. ... The authors should clarify whether any such mechanisms are in place.]
>
> Response:[Thank you for pointing this out. In our implementation, ambiguous or underspecified actions are fundamentally treated as invalid actions.
>
> Due to the explicit action definitions built into the environment (see Appendix B), all valid actions are required to be specific, such as <AttackScreen(screen)>, <AttackMinimap(minimap)>, or <AttackUnit(tag)>. Therefore, vague actions like "attack the enemy" without a valid target tag or position are invalid in our system.
>
> To ensure robustness, the action recognition pipeline enforces validity checks at 3 levels:
>
> (1) Format Validity: The LLMs are required to return actions in a specific format <ActionName(arg1, ...)>. If the response does not follow this structure, or the action is not in the bridge object list, the platform will automatically replace it with <No_Operation()>, which has no effect on the environment.
>
> (2) Action Validity: When the env detects an action, it searches for the pysc2 functions, and the env will automatically check function validity according to the valid action list provided by pysc2. Invalid actions will be replaced by <No_Operation()>.
>
> (3) Argument Validity: We provided a lot of argument validity check functions. For example, build actions should be in a pathable, buildable place, with enough resources and pre-techniques. If an action is paired with invalid arguments, it will be replaced by <No_Operation()>.
>
> These three levels of checks collectively ensure that invalid, ambiguous, or underspecified actions do not impact environment execution.]
>
> ---
>
> Comment3:[Independent evaluation of the contribution of the core components of the system is necessary.]
>
> Response:[Since the core components(obs generation, action recognition) are interface modules, it is unable to test the contribution of these components of the system -- removing the observation generation or action recognition modules would leave the LLM without interfaces to interact with the environment.
>
> Instead, we evaluated the efficiency performance of these modules. They cost only several milliseconds to complete the task, much lower than the latency of LLMs' response:
>
> | Agent Type | Obs/Response Size (Tokens, GPT-3.5) | Env Latency on Obs Processing (s) | Env Latency on Action Recognition (s) | LLM Response Latency (s, GPT-3.5) |
> |----|----|----|----|----|
> | Commander  | 1240/321  | 0.016 | 0.001 | 5.23 |
> | Developer  | 1587/320  | 0.076 | 0.001 | 5.51 |
> | Other      | 2269/363  | 0.019 | 0.001 | 5.58 |
> ]
>
> ---
>
> Comment4:[the primary contribution of the work appears ... aligns more closely with the scope of a Benchmark Track paper than a Main Track submission.]
>
> Response:[Considering our research footprint follows the TextCtarcraft2 paper (cite [6]) from NIPS2024 to supplement their overlooked action space, and research on the problems and performance in LLM decision-making. Therefore, like them, we chose to send it to the Main Track.]
>
> ---
>
> We hope the above response addresses your concerns. We sincerely thank you for your efforts in reviewing our paper.

---

### Official Review · Reviewer_s46V · 2025-07-03

**Clarity:** 3
**Significance:** 3
**Originality:** 2
**Rating:** 4
**Confidence:** 2

**Summary:**

This work introduces LLM-PySC2--the first StarCraft II environment that offers large language models the full pysc2 action space. The environment also provides multimodal observations and supports native multi‑agent communication through an asynchronous query architecture. Although LLMs demonstrate the potential to win in complicated contexts, it is still hard for them to constantly make correct actions. These could due to the common issues of LLMs such as hallucinations and inefficient collaboration. Most importantly, LLM-PySC2 provides a platform that encourages further research and development of LLM-based StarCraft II agents.

**Questions:**

1. Please consider adding Terran and Zerg.
2. Please provide quantitative analyses for hallucination
3. Please evaluate more open-source LLMs e.g., Qwen, Mistral, etc.

**Ethical Concerns:**

["NO or VERY MINOR ethics concerns only"]

**Final Justification:**

Will keep the original score, since some of the experiments are still missing, e.g., Terran and Zerg.

**Limitations:**

yes

**Paper Formatting Concerns:**

No major formatting issues

**Quality:**

3

**Strengths And Weaknesses:**

**Strengths**
1. LLM-PySC2 is the first StarCraft II framework that expose a complete pysc2 action space to LLMs, along with a structured Wiki knowledge database.

2. The environmental setup is clear and reasonable.

3. This work considers representative experimental scenarios including macro-decisions experiments, micro-operations experiments, and evaluates over a board range of LLMs.

4. LLM-PySC2 provides a platform that promotes the development of LLM-based decision-making approaches, especially on StarCraft II.


**Weaknesses**
1. Only the Protoss is considered. Terran and Zerg are not included.
2. This work does not test the LLMs that are fine-tuned using RL techiniques like PPO, GRPO.
3. Latency and token cost are not reported.
4. Hallucination analysis is a bit qualitative, with no quantitative evidences.
5. Minor issues: Spacebar is not properly used in Table 2.

---

> ### Author Rebuttal · Authors · 2025-07-29
>
> Dear reviewer,
>
> We sincerely thank you for your time, thoughtful comments, and constructive suggestions. We greatly appreciate your efforts in reviewing our work and providing feedback that will help us improve the quality and clarity of the paper. Below, we provide point-by-point responses to each of the raised concerns.
>
> ---
>
> Question1: [Please consider adding Terran and Zerg.]
>
> Response1: [Thank you for the valuable suggestion. Actually, the development of the other two races is already underway. We fully agree that including all three StarCraft races—Protoss, Terran, and Zerg—would further enhance the generality and usability of the platform, but these works involves significant engineering complexity and effort. We apologize for the current limitation. The other two races will be added to the environment in the future version of our platform.
>
> (1) Importantly, the platform was architected with extensibility. During the development of Protoss, we deliberately designed the framework in a way that closely mirrors the structure and mechanics of Terran and Zerg. This allows us to port and extend the core functionalities efficiently to the remaining races, and we are actively working on this extension. The platform is designed to evolve, and support for all races will be gradually added in future releases.
>
> (2) On the other hand, even with only the Protoss race currently implemented, the platform already supports a wide range of research capabilities. The existing features are sufficient for evaluating large language models’ planning, decision-making, and control abilities in complex environments. We believe it offers meaningful scientific value as-is, and we hope it can already serve as a strong foundation for future research.]
>
> ---
>
> Question2: [Please provide quantitative analyses for hallucination]
>
> Response2: [We fully agree that hallucination is a crucial aspect to analyze. However, we note that currently there is no widely accepted quantitative metric for measuring hallucination in such sequential decision-making settings, especially in complex, partially observable environments like StarCraft II.
>
> In our work, we observed two major types of hallucinations that commonly arise in agent behavior:
>
> (1) Factual hallucinations, such as misidentifying enemy units or incorrectly assessing the current game state; and
>
> (2) Strategic hallucinations, such as giving unreasonable build orders (e.g., only producing a single type of military units such as Zealot) or raising severely mistimed attacks.
>
> While difficult to quantify directly, these hallucinations ultimately manifest in degraded performance and suboptimal decision-making. As such, we use winning rate as indirect index to quantitative reflect the cumulative impact of hallucinations on gameplay outcomes.
> To aid understanding of the details of the hallucination, we have added a set of examples in Appendix E, which presents examples of both micro-level and macro-level erroneous decisions made by the agent. These examples help clarify how hallucinations lead to ineffective strategies, poor execution, or failure to respond to key events.
>
> We thank you again for pointing out this important direction, and we hope our added analysis provides meaningful insight.]
>
> ---
>
> Question3: [Please evaluate more open-source LLMs e.g., Qwen, Mistral, etc.]
>
> Response3: [Thank you for the valuable suggestion. We appreciate the importance of evaluating a broader range of open-source large language models (LLMs) such as Qwen and Mistral to enhance the credibility and generality of our platform's benchmarks.
> Due to the nature of the long sequential of interactions in StarCraft -II games, adding comprehensive winning rate evaluations of Qwen, Mistral or other models is time-intensive. We will try to add the evaluation of these models before the final  submission deadline.
>
> Since our work focuses on architectural design, the platform is built to support any large language model in a model-agnostic manner. We have already tested a wide range of open-source models including [GPT-3.5, GPT-4o, GPT-4o-mini, Claude, Llama3.1 with different sizes, GLM4, reasoning model GPT-o1-mini,etc.]. However, due to time constraints, we were unable to cover all mainstream models—there are simply too many, and they are evolving at a very rapid pace.
>
> We fully agree that incorporating more open-source models would strengthen the evaluation, and we will make our best effort to include evaluations of Qwen and Mistral before the final submission deadline. If time constraints prevent this, we will include them in a follow-up arXiv version and future updates to the platform.
>
> Finally, we would like to clarify that the primary goal of our work is to develop a general, open, and extensible StarCraft II platform tailored for LLM integration and benchmarking. The experiments serve mainly to demonstrate this compatibility and highlight common failure modes. Since our platform is model-agnostic and supports any LLM with a standard API, we believe it can be a valuable resource for continued research across a wide variety of open-source and proprietary models.]
>
> ---
>
> Additionally, we noticed that you pointed out several issues in the Weaknesses section that were not explicitly raised as formal questions in the Questions to Authors section. We would like to briefly address these points here, in the hope that our responses help clarify your concerns.
>
> ---
>
> Weakness2: [This work does not test the LLMs that are fine-tuned using RL techiniques like PPO, GRPO.]
>
> Response: [The main reason we have not tested reinforcement learning (RL)-fine-tuned LLMs such as those trained with PPO or GRPO is that, to the best of our knowledge, no existing LLMs have been fine-tuned using RL techniques in a full StarCraft II environment with an complete action space.
>
> Prior environments such as TextStarCraft2 and SwarmBrain are built with highly simplified settings. These platforms use narrow action spaces which significantly limit the complexity of the decision-making and reduce the generality of learned strategies and there is no wide-accepted LLM reinforcement work on these former StarCraft-II environments.
>
> Considering that our main goal is to provide an LLM decision making environment instead of a new training technique, we leave it to future researchers on how to efficiently train large models to complete sequence decision-making tasks in complex scenarios. Hoping that our environment will be helpful for future researches.]
>
> ---
>
> Weakness3: [Latency and token cost are not reported.]
>
> Response: [Thank you for pointing this out. We agree that latency and token cost are important aspects to consider when evaluating the practicality of using LLMs in real-time environments such as StarCraft II.
>
> In our setup, both latency and token consumption vary significantly depending on the task, game situation, and the agent. Here, we compute the token cost and average time spent on observation processing and action recognition in StarCraft-II complete game, categorized by agent type.
>
> | Agent Type | Obs Size (Tokens) | Response Size (Tokens, GPT-3.5) | Env Latency on Obs Processing (s) | Env Latency on Action Recognition (s) | LLM Response Latency (s, GPT-3.5) |
> |-------------------|----------|----------|----------|----------|---------|
> | Commander  | 1240   | 321    | 0.016  | 0.001  | 5.23    |
> | Developer      | 1587   | 320   | 0.076   | 0.001  | 5.51   |
> | Other             | 2269   | 363    | 0.019  | 0.001  | 5.58   |
>
> The length of input observation prompt typically varies significantly across different stages of the game (e.g., early vs. late game). As more units from both sides are observed, the amount of information increases, and the resulting rise in input token cost due to richer observations is inevitable. It is worth noting that the environment's latency in processing observation and text actions is significantly lower than that of LLM inference, ensuring that the environment's execution efficiency is fully sufficient for decision-making by LLMs.
>
> Considering that reporting latency and token cost is a very meaningful suggestion, we will include a summary of this analysis in the final version of the paper, either in the main text or in the appendix.]
>
> ---
>
> Weakness5: [Minor issues: Spacebar is not properly used in Table 2.]
>
> Response: [Thank you for your careful review. We have checked the submitted version and confirmed the formatting issue in Table 2. It seems that the problem is not raised by Spacebar but the LaTeX's compilation of table formats. We will try to correct this issue in the final version of the paper to make the table looks better.]
>
> ---
>
> Once again, we thank you for your valuable feedback and hope that our clarifications and updates address your concerns.
>
> Sincerely,
>
> The Authors

---

> > ### Comment · Reviewer_s46V · 2025-08-06
> >
> > Thank you for the detailed response. It indeed answers some of my questions.

---

> > > ### Author Response · Authors · 2025-08-07
> > >
> > > We sincerely thank you for your time, effort, and constructive feedback in the review and comments. Your insights have helped us identify important aspects for clarification and improvement, and we greatly appreciate the thoughtful suggestions that contributed to refining the quality of our work.

---

### Comment · Area_Chair_cv17 · 2025-08-02
**Please Engage with Authors’ Responses During Rebuttal**

Dear Reviewers,

As we approach the rebuttal phase, I want to take a moment to remind you of the importance of carefully reviewing the authors’ responses. The rebuttal process is a valuable opportunity for authors to clarify misunderstandings, address concerns, and provide additional evidence or analysis to support their work.

When reviewing the authors’ responses, please:

1. Read their replies thoroughly to ensure you understand their points.
2. Assess whether your concerns have been adequately addressed.

Your engagement in this process helps ensure a fair and constructive review outcome. Thank you for your time and dedication to maintaining the high standards of NeurIPS. Your thoughtful participation in this phase is greatly appreciated!

Best regards,
Area Chair

---

### Note · Authors · 2025-08-15

We sincerely thank the reviewers and the area chairs for your valuable time, constructive feedback, and thoughtful discussion throughout the review process. We have carefully considered all comments and addressed them in our rebuttal and discussion responses. Below, we summarize the most critical clarifications and conclusions for the ACs’ consideration.

---

1. Key Clarifications Provided During Rebuttal

During the rebuttal and discussion phase, we clarified the following critical issues:

(1) Wiki knowledge. We provided the path to the corresponding code and introduced the details of this wiki database and finally solved the reviewer's concern. This database provides complete unit information for all three SC2 races in JSON format, which will be helpful for future LLM+SC2 development.

(2) Comparison with RL. (a) Comparison with LLM+RL works. Since this is the first environment providing LLMs with the complete SC2 action space, there are no prior methods available for direct comparison; (b) Comparison with MARL. Currently, pretrained LLMs are weaker than in-environment trained MARL methods. The gap in data leads to LLMs’ current decision-making level lower than that of MARL.

(3) Performance, e.g., latency of the core modules. We introduced the latency of the two core components in our environment, which is roughly three orders of magnitude lower than that of the latency of LLM itself (1~20ms vs 5s). This ensures environmental efficiency.

2. Responses to Unsolved Discussion Points

Thanks to the reviewers’ responsible review and discussion, no new concerns were raised near the end of the discussion phase, and all questions and weaknesses mentioned by the reviewers have been answered and clarified.

3. Contribution and Significance

Our main contributions can be concluded as follows: We proposed the first LLM StarCraft-II environment that provide access to the complete pysc2 action space, with structured game Wiki knowledge, and evaluations for baseline performance across multiple models and scenarios.

We hope the environment will enable the LLMs to gradually approach the MARL-level decision quality in the future, while retaining the robustness, interactivity, and interpretability.

---

We believe the final paper will present a clear, well-supported, and impactful contribution, thanks to the reviewers’ suggestions. We appreciate the ACs’ and reviewers’ efforts in evaluating our work, and we hope this summary helps in the decision-making process.

---

### Decision · Program_Chairs · 2025-09-17

**Decision:**

Accept (poster)

**Comment:**

## Summary.
This work introduces LLM-PySC2, a StarCraft II environment tailored for large language models (LLMs), providing the full PySC2 action space, multimodal observations, and an asynchronous query architecture for multi-agent communication. The paper evaluates LLMs in macro-decision and micro-operation scenarios, highlighting their potential and limitations, such as hallucinations and inefficient collaboration.


### Strengths:
- Reviewers commend LLM-PySC2 as the first StarCraft II environment supporting the complete PySC2 action space for LLMs, providing a valuable platform for studying LLM-based decision-making in complex strategy games. The inclusion of multimodal observations (e.g., RGB screen and feature data) and a structured Wiki knowledge database is seen as a significant advancement for benchmarking multi-modal LLMs.
- Reviewers note that the environment’s setup is well-designed and reasonable, with detailed descriptions of the framework, action space, and multi-agent system.
- The paper evaluates LLMs across diverse scenarios, including macro-decision and micro-operation tasks, demonstrating their potential and limitations in complex decision-making.
- LLM-PySC2 fosters further research into LLM-based decision-making, particularly for StarCraft II, aligning with current trends in multi-modal and game AI research.

### Weaknesses:
- Reviewers criticize the lack of comparisons with traditional reinforcement learning (RL) methods, such as AlphaStar, or other LLM architectures.
- Reviewers note the absence of ablation studies to evaluate the contributions of individual components (e.g., multimodal observations, asynchronous query architecture).
- The Protoss race is considered, omitting Terran and Zerg, which limits the environment’s universality. Additionally, reviewer highlights that RL-fine-tuned LLMs (e.g., via PPO or GRPO) were not tested.
- The qualitative nature of hallucination analysis lacks quantitative evidence, and reviewer suggests improving the presentation of hallucination examples for clarity.
- The use of outdated LLMs and suggests testing newer models like GPT-o3/o4-mini or Gemini-2.5-Pro, noting that the poor performance of reasoning models (e.g., GPT-o1) in unseen scenarios contradicts other works.
- It is hard to assess novelty due to limited evaluation of failure modes enabled by the full action space.

**After rebuttal,** some concerns of reviewers are addressed. Overall, all reviewers are positive about the paper. Considering that this work lacks of enough insightful empirical findings or algorithm novelty, I'd like to recommend accept but with low confidence.